# Probabilistic Weather Forecasting with Hierarchical Graph Neural Networks

**Joel Oskarsson**
Linköping University
`joel.oskarsson@liu.se`

**Tomas Landelius**
Swedish Meteorological and
Hydrological Institute
`tomas.landelius@smhi.se`

**Marc Peter Deisenroth**
University College London
`m.deisenroth@ucl.ac.uk`

**Fredrik Lindsten**
Linköping University
`fredrik.lindsten@liu.se`

## Abstract

In recent years, machine learning has established itself as a powerful tool for high-resolution weather forecasting. While most current machine learning models focus on deterministic forecasts, accurately capturing the uncertainty in the chaotic weather system calls for probabilistic modeling. We propose a probabilistic weather forecasting model called Graph-EFM, combining a flexible latent-variable formulation with the successful graph-based forecasting framework. The use of a hierarchical graph construction allows for efficient sampling of spatially coherent forecasts. Requiring only a single forward pass per time step, Graph-EFM allows for fast generation of arbitrarily large ensembles. We experiment with the model on both global and limited area forecasting. Ensemble forecasts from Graph-EFM achieve equivalent or lower errors than comparable deterministic models, with the added benefit of accurately capturing forecast uncertainty.

## 1 Introduction

Forecasting the dynamics of Earth's atmosphere is a scientific problem of utmost importance. Society is dependent on fast and informative weather forecasts for planning in areas such as transportation and agriculture and for balancing the energy system [3]. Especially important is the use of forecasts to issue warnings for extreme weather events [1]. Recent advances in Machine-Learning-based Weather Prediction (MLWP) have enabled models that produce accurate forecasts in a fraction of the time of traditional physics-based systems [37, 4, 23]. So far these developments have largely been focused on deterministic modeling. However, forecasting only one likely weather scenario ignores the many uncertainties in predicting future weather.

Weather is a chaotic system, resulting in high forecast uncertainty [52]. This uncertainty comes from both imperfect representations of initial states and inaccurate descriptions of the function mapping from one time step to the next [26]. Accurately modeling this uncertainty significantly increases the value of weather forecasts. Such uncertainty can be communicated to end-users to improve decision making or be used in downstream products, for example to compute a distribution over solar power generation. Capturing the full forecast uncertainty requires us to predict not just a single likely state trajectory, but a collection of possible future weather states. Due to the complexity and dimensionality of the weather system the feasible way to achieve this is by generating samples from a modeled distribution. Such *ensemble forecasting* is today performed using physics-based methods, where a number of *ensemble members* are simulated as samples from this distribution. The computational cost of this is however massive, often limiting the spatial resolution or size of the ensemble [3].

38th Conference on Neural Information Processing Systems (NeurIPS 2024).

MLWP is a promising approach for addressing this limitation and enabling large ensemble forecasts. However, for the ensemble to add value the machine learning model needs to accurately represent the distribution. Initial attempts at MLWP ensemble forecasting either rely on ad-hoc initial state perturbations [10, 37, 4] or have not been scaled to spatial resolutions of interest [19]. Also diffusion models [18] have been applied to the problem, but sampling forecasts from these is computationally expensive and can be prohibitively slow [39]. We propose a Graph-based Ensemble Forecasting Model (Graph-EFM), enabling efficient sampling of ensemble members with only one forward-pass per time step. The method builds on graph-based MLWP [20, 23], which is a flexible framework that can be adapted to different geometries and state grid representations [24]. By combining a latent-variable formulation with a hierarchical Graph Neural Network (GNN) the distribution is modeled in a lower-dimensional space and sampled forecasts are spatially coherent.

MLWP models are typically trained for and evaluated on global weather forecasting [40, 23, 4]. Another common forecasting setup in practice is the use of Limited Area Models (LAMs) to produce high-resolution regional forecasts [11]. Such LAMs are for example used by local weather services in order to provide forecasts tailored to the geographical properties and societal needs of the region [38, 44, 7, 32]. These high-resolution models are also invaluable to various industrial sectors, including energy forecasters, who rely on precise weather predictions to manage supply and demand. This motivates research into also constructing MLWP LAMs, which brings new challenges related to the high resolution and boundary conditions of the limited area. In this work we experiment not just with global forecasting, but consider also how probabilistic LAMs can be trained to produce forecasts for the Nordic region.

**Our main contributions are:** 1) We develop a hierarchical GNN framework for both deterministic and probabilistic MLWP. The hierarchical construction encourages spatially coherent fields in forecasts. 2) We use this framework to define the probabilistic weather forecasting model Graph-EFM, capable of efficient sampling of arbitrarily large ensemble forecasts. 3) We develop a training method targeting both forecast quality and ensemble calibration. 4) We experiment with both global forecasting on 1.5° resolution and a novel limited-area modeling task at 10 km resolution.

## 2   Related Work

**Deterministic MLWP**   Multiple machine learning methods have been successfully applied to large-scale weather forecasting. These include graph-based models [20, 23, 24], transformers [4, 8, 10, 33, 25, 34] and neural operators [37, 5]. While large neural network models learn weather dynamics purely from data, there are also parallel developments in building hybrid physics-MLWP models [22, 50].

**Ensembles from perturbations**   Most existing methods for MLWP ensemble forecasting follow closely the physics-based methods, where initial states and model parameters are *perturbed* to create ensemble diversity. A number of MLWP works create ensembles by ad-hoc perturbing initial states with random noise [10, 37, 4, 16, 6]. More informed perturbations have been re-used from physics based ensembles [39, 6] and created based on model-informed singular vectors [43]. Others try to perturb the forecast model itself, rolling out ensemble members using different neural network parameters [51, 43]. Such multi-model approaches require training, or at least fine-tuning, a pre-defined number of MLWP models. Graubner et al. [16] use the SWAG method [30] to allow for constructing multi-model ensembles of arbitrary size.

**Generative modeling**   Probabilistic machine learning approaches aim to directly learn generative models producing ensemble members. Similar to our approach, the SwinVRNN model [19] uses a latent variable formulation, but combined with a Swin Transformer architecture [29]. SwinVRNN is developed for global forecasting at 5° resolution and scales poorly to higher spatial resolutions. Also building on the graph-based framework, Price et al. [39] train a diffusion model [18, 46] to sample each time step. Their Gencast model produces ensemble forecasts of 0.25° global data with 12 h time steps. Diffusion models produce realistic-looking samples, but typically require solving an ordinary differential equation involving multiple passes through the neural network to sample each time step. For GenCast, this results in a sampling time of 8 minutes for a single 15 day forecast on a TPUv5 device [39]. Other works use diffusion models to increase the size of physics-based ensembles [27] or stochastically downscale deterministic forecasts [9, 31].

**Hierarchical GNNs** Motivated by capturing multiple spatial scales, hierarchical GNNs have been used for modeling general partial differential equations [14, 28]. The overall hierarchical framework shares much of its structure with the popular U-Net architecture [41] for computer vision tasks, but extended to a general graph setting.

# 3 Background

## 3.1 Problem Definition

The weather forecasting problem can be summarized as mapping from a set of initial states $X^{-1:0} = (X^{-1}, X^0)$ to the sequence of future states $X^{1:T} = (X^1, \ldots, X^T)$. A table of notation is provided in appendix A. Each weather state $X^t \in \mathbb{R}^{N \times d_x}$ here contains $d_x$ variables modeled at $N$ different locations. Geospatial data is often represented as regular grids, in which case these locations correspond to the grid cells. The $d_x$ variables can include both atmospheric variables, modeled at multiple vertical levels, and surface variables. As is common in MLWP we assume the initial states to consist of two time steps, which allows for capturing first-order state dynamics. To produce a forecast, a set of forcing inputs $F^{1:T}$ are also available. These contain known quantities, such as the time of day. There are also static features associated with the grid cells, such as the orography, which we here consider part of the forcing.

Many variables impact the chaotic weather system, all of which are not fully captured in initial states represented on finite grids. This induces forecast uncertainty, which we view as a distribution $p(X^{1:T}|X^{-1:0}, F^{1:T})$. In deterministic forecasting we seek a model that minimizes the Mean Squared Error (MSE) to the future weather states [23, 33, 34]. This is equivalent to modeling only the mean of the distribution. In probabilistic forecasting we instead aim to model the full distribution. Note that we here specifically model the *conditional* distribution $p(X^{1:T}|X^{-1:0}, F^{1:T})$, rather than $p(X^{1:T}|F^{1:T})$. Hence we do not marginalize over uncertainty in initial states.

## 3.2 Graph-based Weather Forecasting

Graph-based MLWP models use an autoregressive mapping $\hat{X}^t = f(X^{t-2:t-1}, F^t)$ consisting of a sequence of GNNs [20, 23, 24]. Starting from the initial states, this mapping can be iteratively applied to roll out a full forecast $X^{1:T}$. Central to the graph-based framework is the idea of mapping from the original $N$ grid locations to a *mesh graph* $\mathcal{G}_M = (\mathcal{V}_M, \mathcal{E}_M)$. In the graph-context we refer to the grid locations as a set $\mathcal{V}_G$ of *grid nodes*. By choosing $|\mathcal{V}_M| < |\mathcal{V}_G| = N$ it becomes efficient to perform the majority of computations on the mesh. Such a mesh graph can also be tailored to the forecasting setting, for example to respect the spherical geometry in global forecasting [20]. The mapping $f$ realizes a single-step prediction by passing $X^{t-2:t-1}$ and $F^t$ through a series of GNN layers. In sequence, these layers: 1) map grid inputs to representations on the mesh graph; 2) perform a number of processing steps on the mesh; 3) map back to the grid to produce the prediction for $X^t$. Steps 1 and 3 use bipartite graphs $\mathcal{G}_{G2M} = (\mathcal{V}_G \cup \mathcal{V}_M, \mathcal{E}_{G2M})$ and $\mathcal{G}_{M2G} = (\mathcal{V}_G \cup \mathcal{V}_M, \mathcal{E}_{M2G})$ with edges connecting the grid and mesh nodes. The GNN layers in each step compute updates for node representations $H \in \mathbb{R}^{|\mathcal{V}| \times d_z}$ and edge representations $E \in \mathbb{R}^{|\mathcal{E}| \times d_z}$ in the graphs. For simplicity all representation vectors have dimensionality $d_z$.

**Interaction Networks** The specific GNN layers used in previous works are *Interaction Networks* [2, 23]. The layers in these networks pass messages from a set of sender nodes along directed graph edges to a set of receiver nodes. Based on these messages the edge and receiver node representations are then updated. For a graph $\mathcal{G} = (\mathcal{V}, \mathcal{E})$ let $\boldsymbol{e}_{\alpha \to \beta} \in \mathbb{R}^{d_z}$ be the row of $E$ corresponding to the edge $(\alpha, \beta) \in \mathcal{E}$. Let $H^S$ be the matrix with rows containing sender node representations and $H^R$ the corresponding matrix for receiver nodes. Interaction Networks then implement the representation update $H^R, E \leftarrow \text{GNN}(\mathcal{G}, H^S, E, H^R)$ as

$$\tilde{\boldsymbol{e}}_{\alpha \to \beta} \leftarrow \text{MLP}\big(\boldsymbol{e}_{\alpha \to \beta}, H_\alpha^S, H_\beta^R\big) \tag{1a}$$

$$\boldsymbol{e}_{\alpha \to \beta} \leftarrow \boldsymbol{e}_{\alpha \to \beta} + \tilde{\boldsymbol{e}}_{\alpha \to \beta} \qquad H_\beta^R \leftarrow H_\beta^R + \text{MLP}\Big(H_\beta^R, \textstyle\sum_{\alpha \in \text{Ne}(\beta)} \tilde{\boldsymbol{e}}_{\alpha \to \beta}\Big) \tag{1b}$$

where $\text{Ne}(\beta) = \{\alpha : (\alpha, \beta) \in \mathcal{E}\}$ are the incoming neighbors of node $\beta$. Parameters in Multi-Layer Perceptrons (MLPs) are shared across nodes and edges in the graph, but not between GNN layers.

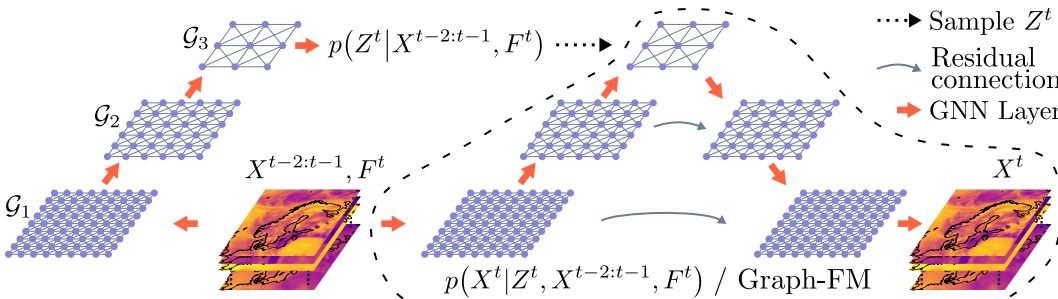

Figure 1: Overview of our Graph-EFM model, with example data and graphs for a Limited Area Model. The corresponding overview for the global setting is given in fig. 6 in appendix C.

**Global mesh graphs** Keisler [20] proposed to construct a mesh graph for global MLWP as an icosahedral grid covering the globe. This approach was extended in the GraphCast model [23] by introducing a multi-scale mesh graph with edges of varying length. Such multi-scale edges are capable of propagating information and capturing statistical dependencies both locally and over long distances in the graph. The multi-scale mesh graph is created by sequentially splitting the faces of an icosahedron into a sequence of graphs $\mathcal{G}_L, \ldots, \mathcal{G}_1$ with node sets satisfying $\mathcal{V}_L \subset \cdots \subset \mathcal{V}_1$ by construction. The original icosahedron $\mathcal{G}_L$ has the longest edges $\mathcal{E}_L$, stretching far across the globe, whereas the final graph $\mathcal{G}_1$ has short edges $\mathcal{E}_1$ only connecting nodes locally. The final multi-scale mesh graph is constructed as $\mathcal{G}_{\mathrm{MS}} = (\mathcal{V}_1, \mathcal{E}_L \cup \cdots \cup \mathcal{E}_1)$, taking the nodes from the final graph but connecting these using edges of all different lengths [23].

## 4 Weather Forecasting with Hierarchical Graph Neural Networks

Two great challenges in weather forecasting is to accurately capture processes unfolding over different spatial scales and modeling the uncertainty in the chaotic system [52]. To tackle these challenges, we propose to construct a hierarchical mesh graph, working with different length scales at each level in the hierarchy. We use a sequence $\mathcal{G}_1, \ldots, \mathcal{G}_L$ of graphs as the different levels in the hierarchy, additionally adding connections between the nodes of adjacent levels. This construction is also highly suitable as a basis for building probabilistic forecasting models, as discussed below. Figure 1 shows an overview of the hierarchical mesh used in our model. See figs. 12 and 14 in the appendix for illustrations of how this differs from the multi-scale graph.

There are multiple benefits to such a hierarchical mesh construction for MLWP. By keeping the graphs at different levels separate, we can define GNN layers with independent parametrizations at each level. This adds flexibility by allowing the model to learn different representation updates for edges of different spatial scales. A hierarchical mesh graph also offers a natural, spatially-aware dimensionality reduction, as the state in the grid is encoded into a few nodes at the top level. Such a representation can capture the general structure of each weather state, with finer details added as this is propagated down through the hierarchy. We leverage this property to construct a probabilistic model by imposing a distribution over these lower-dimensional representations at the top level. This allows for efficiently drawing spatially coherent samples from the distribution of future weather states.

### 4.1 Hierarchical Graph

Our hierarchical mesh graph consists of $L$ graph levels $\mathcal{G}_1, \ldots, \mathcal{G}_L$ with $\mathcal{G}_l = (\mathcal{V}_l, \mathcal{E}_l)$. Only level 1 of the hierarchy is connected to the grid, so we re-define $\mathcal{G}_{\mathrm{G2M}} = (\mathcal{V}_G \cup \mathcal{V}_1, \mathcal{E}_{\mathrm{G2M}})$ and $\mathcal{G}_{\mathrm{M2G}} = (\mathcal{V}_G \cup \mathcal{V}_1, \mathcal{E}_{\mathrm{M2G}})$. The number of nodes $|\mathcal{V}_l|$ decreases with the level $l$. The smallest set of nodes are found at the top level $L$.

To pass information between the levels of the hierarchy we introduce additional graphs connecting the different levels. Let $\mathcal{G}_{l,l+1} = (\mathcal{V}_l \cup \mathcal{V}_{l+1}, \mathcal{E}_{l,l+1})$ be a graph containing directed edges from mesh level $l$ to level $l+1$. We make use of a graph sequence $\mathcal{G}_{1,2}, \ldots, \mathcal{G}_{L-1,L}$ to propagate information up through the hierarchy and similarly a sequence $\mathcal{G}_{L,L-1}, \ldots, \mathcal{G}_{2,1}$ in the downward direction. The exact layout of nodes and edges at and in-between levels are design choices that should be tailored to the specific forecasting setting. Examples for global and limited-area forecasting are given in section 5.

## 4.2 Graph-FM: Deterministic Forecasting

The hierarchical graph allows for defining GNN layers both on and in-between the different levels. By sequentially updating node and edge representations at different levels in the hierarchy, information can be propagated up from the grid to the different levels. As these levels have edges of different lengths, the processing at each level happens on different spatial scales. Note that this differs from the multi-scale graph approach, where information processing over all different spatial scales happen in the same GNN layer [23]. As a step towards our probabilistic model, we define an alternative deterministic Graph-based Forecasting Model *Graph-FM*[1], operating on the hierarchical graph.

In Graph-FM one processing step on the mesh graph is defined as a complete sweep through the hierarchy. GNNs are applied sequentially to the inter-level and intra-level graphs in the order $\mathcal{G}_1, \mathcal{G}_{1,2}, \mathcal{G}_2, \ldots, \mathcal{G}_{L-1,L}, \mathcal{G}_L$, updating edge and node representations at the different levels. Processing steps going up the hierarchy are alternated with similar steps going down from level $L$ to 1. The single step mapping $f$ consists of multiple such sweeps up and down (see appendix C.2).

## 4.3 Graph-EFM: Probabilistic Forecasting

To capture the uncertainty in the chaotic weather system we next aim to construct a probabilistic model from the ground up to capture the full distribution $p\big(X^{1:T}\big|X^{-1:0}, F^{1:T}\big)$. We start by assuming the weather system to satisfy a second-order Markov assumption, decomposing

$$p\big(X^{1:T}\big|X^{-1:0}, F^{1:T}\big) = \prod_{t=1}^{T} p\big(X^t\big|X^{t-2:t-1}, F^t\big). \qquad (2)$$

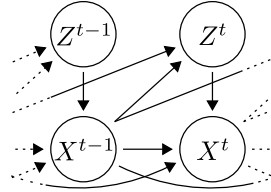

Figure 2: Graphical model for eq. (3).

Factoring the distribution over time steps allows us to work with forecasts of varying length. Specifying the model for single-step prediction avoids having to learn separate parameters for different lead times. Next, we seek a flexible, but computationally efficient parametrization for the distribution $p\big(X^t\big|X^{t-2:t-1}, F^t\big)$. This can be achieved by introducing a latent random variable $Z^t$, and letting

$$p\big(X^t\big|X^{t-2:t-1}, F^t\big) = \int p\big(X^t|Z^t, X^{t-2:t-1}, F^t\big) p\big(Z^t\big|X^{t-2:t-1}, F^t\big) dZ^t. \qquad (3)$$

Here the stochasticity in $Z^t$ should capture the uncertainty over $X^t$ at each time step. The corresponding graphical model is shown in fig. 2. We impose a spatial structure over the latent variable by letting $Z^t$ be $|\mathcal{V}_L| \times d_z$ matrix-valued, with each row a $d_z$-dimensional vector associated with one node in the top level $\mathcal{G}_L$ of the mesh graph.

The single-step model consists of two components, a latent map $p\big(Z^t\big|X^{t-2:t-1}, F^t\big)$ and predictor $p\big(X^t|Z^t, X^{t-2:t-1}, F^t\big)$. The latent map is parametrized using GNNs, mapping the conditioning variables to parameters of a Gaussian distribution. We consider the predictor to be concentrated around its mean, and realize $p\big(X^t|Z^t, X^{t-2:t-1}, F^t\big)$ as a deterministic mapping of a similar form as Graph-FM. By sampling $Z^t$ and passing this through the predictor we can draw a sample of $X^t$ from eq. (3). This sample can then be conditioned on at the next time step, continuing this sampling process to roll out a forecast following eq. (2). This forecast constitutes one ensemble member, and the process can be repeated to sample an ensemble of arbitrary size. We call our Graph-based Ensemble Forecasting Model *Graph-EFM*. Full details about the model are given in appendix C.

**Latent map**   We let the latent map be an isotropic Gaussian

$$p\big(Z^t\big|X^{t-2:t-1}, F^t\big) = \prod_{\alpha \in \mathcal{V}_L} \mathcal{N}\big(Z_\alpha^t\big|\mu_Z\big(X^{t-2:t-1}, F^t\big)_\alpha, I\big) \qquad (4)$$

with the mean as a function of the conditioning variables. The variance is fixed, imposing a fixed scale for the learned latent space. The mean function $\mu_Z$ consists of a sequence of GNNs. These take the inputs at the grid, propagate representations up through the hierarchical mesh graph, and finally predicts the mean of $Z_\alpha^t$ at each node $\alpha$ at level $L$. In appendix L.2 we verify empirically the importance of using the latent map over a static distribution for $Z^t$.

---

[1]The deterministic Graph-FM model was first proposed in a preliminary version of this work [35], but there only for the LAM setting under the name *Hi-LAM*.

**Predictor** The predictor is a deterministic mapping

$$\hat{X}^t = g\big(Z^t, X^{t-2:t-1}, F^t\big) = X^{t-1} + \tilde{g}\big(Z^t, X^{t-2:t-1}, F^t\big). \tag{5}$$

With the small time steps used in MLWP, $X^t$ does not change dramatically in a single step. We thus follow the common practice of including a skip connection to the previous state [23, 5, 19]. The predictor takes both inputs $X^{t-2:t-1}, F^t$ at the grid and $Z^t$ at the top of the mesh graph. To incorporate both we design $g$ similar to Graph-FM, performing sweeps up and down through the mesh hierarchy. At the top of the hierarchy $Z^t$ is added to node representations $H^L$ through the residual connections in the GNN layers. A sampled value of $Z^t$ then affects the prediction $\hat{X}^t$ through the downward sweep. While multiple such sweeps are possible, we found one to be sufficient in practice.

**Spatial dependencies** We want each sample of $X^t$ to contain spatially coherent atmospheric fields. One approach would be to impose spatial dependencies in the joint distribution over $Z^t$. However, learning and sampling from such a distribution typically comes with computational challenges [19]. Instead, we impose spatial dependencies by integrating the latent variable formulation with the hierarchical graph. We argue that as the independent components of $Z^t$ are propagated down through the mesh graph, gradually increasing the spatial resolution, spatial dependencies are introduced by the model in the GNN layers. The hierarchical graph is key to this property, as the stochasticity in $Z^t$ is necessarily spread out over the forecast region, rather than only affecting the output locally.

## 4.4 Training Objective

Deterministic forecasting models can be straightforwardly trained by minimizing a weighted MSE [23] or Negative Log-Likelihood (NLL) loss [8] for rolled out forecasts. To train Graph-EFM we instead leverage the fact that the single-step model has a structure similar to a (conditional) Variational AutoEncoder (VAE) [21, 45], allowing us to use a variational objective. We introduce a variational approximation $q\big(Z^t\big|X^{t-2:t-1}, X^t, F^t\big)$ at each time step, approximating the true posterior $p\big(Z^t\big|X^{t-2:t-1}, X^t, F^t\big)$ over $Z^t$. This variational distribution is parametrized in a similar way as the latent map, with GNN layers mapping to a Gaussian over $Z^t$. Note however that $q$ also depends on $X^t$, since it approximates the posterior. Using $q$, we can then define

$$\mathcal{L}_{\text{Var}}\big(X^{t-2:t-1}, X^t, F^t\big) = \lambda_{\text{KL}} D_{\text{KL}}\big(q\big(Z^t\big|X^{t-2:t-1}, X^t, F^t\big)\big\|p\big(Z^t\big|X^{t-2:t-1}, F^t\big)\big)$$
$$-\mathbb{E}_{q(Z^t|X^{t-2:t-1}, X^t, F^t)}\Big[\sum_{\alpha\in\mathcal{V}_G}\sum_{j=1}^{d_x}\log\mathcal{N}\Big(X_{\alpha,j}^t\Big|g\big(Z^t, X^{t-2:t-1}, F^t\big)_{\alpha,j}, \sigma_{\alpha,j}^2\Big)\Big] \tag{6}$$

which is equal to the (negative) Evidence Lower Bound (ELBO) when the weighting is $\lambda_{\text{KL}} = 1$. While the predictor $g$ is a deterministic mapping, we introduce a Gaussian likelihood in eq. (6) to get a well-defined learning problem. This setup corresponds to the common practice in VAEs of assuming Gaussian observation noise, but not adding this to samples from the model [42]. The standard deviation $\sigma_{\alpha,j}$ can either be a second output from the predictor or manually chosen (see appendix D for details). As with deterministic models [23, 20, 8, 34], we found it crucial to fine-tune on rolled out forecasts of multiple time steps. This improves stability and performance for longer lead times. In the final fine-tuning we include also a Continuous Ranked Probability Score (CRPS) loss term $\mathcal{L}_{\text{CRPS}}$ [15, 22]. The full objective function is then $\mathcal{L} = \mathcal{L}_{\text{Var}} + \lambda_{\text{CRPS}}\mathcal{L}_{\text{CRPS}}$, with $\lambda_{\text{CRPS}}$ a weighting hyperparameter. Including this CRPS loss improves the calibration of ensemble forecasts.

## 4.5 Improved GNN Layers: Propagation Networks

In Graph-EFM there is a large amount of information that needs to be propagated between the grid and $Z^t$. However, the Interaction Network GNNs are biased towards keeping old representations of receiver nodes, rather than updating this with new information from incoming edges. Note in eq. (1) that if the MLPs are initialized to give outputs close to 0, there will be no change to $e_{\alpha\to\beta}$ and $H_\beta^R$.

In practice the model has a hard time learning to propagate useful information up from the grid to $Z^t$. Even when trained purely as an auto-encoder ($\lambda_{\text{KL}} = 0$), $Z^t$ easily ends up being ignored. To remedy this we propose an alternative GNN formulation that we call *Propagation Network*, defined by

$$\tilde{e}_{\alpha\to\beta} \leftarrow H_\alpha^S + \text{MLP}\big(e_{\alpha\to\beta}, H_\alpha^S, H_\beta^R\big) \qquad e_{\alpha\to\beta} \leftarrow e_{\alpha\to\beta} + \tilde{e}_{\alpha\to\beta} \tag{7a}$$

$$\tilde{H}_\beta^R \leftarrow \frac{1}{|\text{Ne}(\beta)|}\sum_{\alpha\in\text{Ne}(\beta)}\tilde{e}_{\alpha\to\beta} \qquad H_\beta^R \leftarrow \tilde{H}_\beta^R + \text{MLP}\Big(H_\beta^R, \tilde{H}_\beta^R\Big). \tag{7b}$$

For MLPs initialized with outputs close to 0, Propagation Networks reduce to averaging the values of neighboring nodes. This encourages the propagation of information from $H^S$ to $H^R$ by construction. Propagation Networks were found to perform better also in the deterministic model (see comparison in appendix L.1), so we employ these in both Graph-FM and Graph-EFM.

## 5 Experiments

To evaluate our models we conduct experiments on both global and limited area forecasting. The models are implemented[2] in PyTorch and trained on 8 A100 80 GB GPUs in a data-parallel configuration. Training takes 700–1400 total GPU-hours for the global models, and around half of that for the limited area models. The computational demands prevent us from re-training multiple models for statistical analysis. Once trained, sampling from Graph-EFM is highly efficient. Using batched sampling on a single GPU, 80 ensemble members are produced in 200 s (2.5 s per member) for global forecasting.

**Metrics**    We measure the skill of deterministic models by Root Mean Squared Error (**RMSE**). For probabilistic models we compute the RMSE for the ensemble mean. Good skill in terms of RMSE is however not enough for ensemble forecasts, where we want to capture the full distribution. For these we also assess the ensemble calibration by computing the Spread-Skill-Ratio (**SpSkR**). Calibrated uncertainty corresponds to SpSkR $\approx 1$ [13]. We additionally use **CRPS** to measure how well the marginal distributions of the model matches the data. For deterministic models the CRPS reduces to Mean Absolute Error (MAE). Complete definitions of all metrics are given in appendix E.

**Models**    Achieving a fair comparisons of the actual machine learning methodology in MLWP is challenging due to models using different spatial resolution, variables and initial states. We here train an illustrative set of models on the same data and with comparable training setups. Our full **Graph-EFM** model is compared to: 1) **Graph-EFM (ms)**, a version of Graph-EFM using a multiscale mesh graph instead of the hierarchical one. 2) **Graph-FM**, our deterministic model using the hierarchical graph. 3) **GraphCast***, a reimplementation of GraphCast [23], adapted and trained on our datasets. 4) **GraphCast*+SWAG**, a multi-model ensemble created by applying Stochastic Weight Averaging Gaussian (SWAG) [30] to GraphCast*. Inspired by Graubner et al. [16], this represents a simple way to augment a deterministic model to perform ensemble forecasting. Further details about the baseline models are given in appendix C.5. For ensemble models we sample 80 members for the global experiments and 100 members for limited area forecasting. In appendix L.3 we investigate the impact of ensemble size on the evaluation. We find that improvements in metric values quickly saturate when increasing the ensemble size. This shows that sampling even more members would have negligible impact on the results of our experiments.

### 5.1 Global Forecasting with ERA5

**Data and graphs**    We experiment on global weather forecasting up to 10 days with 6 h time steps. The dataset used for training and evaluation is a 1.5° version of the global ERA5 reanalysis[3] [17], provided through the WeatherBench 2 benchmark [40]. The models forecast $d_x = 83$ different variables in total, including both surface-level variables and atmospheric variables at 13 different pressure levels. We use the years 1959–2017 for training, 2018–2019 for validation and 2020 as a test set. Forecasts are always started from initial conditions taken directly from ERA5, both during training and evaluation. For global forecasting we use the graph generation process from GraphCast [23]. The multi-scale graph $\mathcal{G}_{\mathrm{MS}}$ is created by refining the icosahedron 4 times. The hierarchical graph contains 4 levels of such icosahedral grids. More details on the global experiments are given in appendix H.

**Results**    As the models forecast many different variables we present only a selection of results in the main paper. Metric values for geopotential (z500) and 2 m temperature (2t) are listed in table 1 and results for mean sea level pressure (msl) plotted in fig. 3. Line plots for all metrics and a large number of variables are given in appendix J.1. In the appendix we also show comparisons to additional models from the literature, trained on different data, as well as the physics-based IFS-ENS model [12]. The

---

[2]Our code is available at `https://github.com/mllam/neural-lam/tree/prob_model_global` (global forecasting) and `https://github.com/mllam/neural-lam/tree/prob_model_lam` (LAM).

[3]Provided by the Copernicus Climate Change Service under the ECMWF Copernicus License.

Table 1: Selection of results for global forecasting, including geopotential at 500 hPa (`z500`) and 2 m temperature (`2t`). For RMSE and CRPS lower values are better, and SpSkR should be close to 1 for a calibrated ensemble. The best metric values are marked with **bold** and second best underlined.

| Variable | Model | Lead time 5 days | | | Lead time 10 days | | |
|---|---|---|---|---|---|---|---|
| | | RMSE | CRPS | SpSkR | RMSE | CRPS | SpSkR |
| `z500` | GraphCast* | 387 | 236 | - | 808 | 498 | - |
| | Graph-FM | **363** | 223 | - | 825 | 510 | - |
| | GraphCast*+SWAG | 437 | 269 | 0.07 | 960 | 590 | 0.12 |
| | Graph-EFM (ms) | 472 | 211 | 0.77 | 756 | 333 | 0.83 |
| | Graph-EFM | 399 | **169** | **1.18** | **695** | **299** | **1.15** |
| `2t` | GraphCast* | 1.65 | 1.00 | - | 2.82 | 1.69 | - |
| | Graph-FM | **1.57** | 0.94 | - | 2.82 | 1.66 | - |
| | GraphCast*+SWAG | 2.03 | 1.20 | 0.06 | 3.58 | 2.04 | 0.13 |
| | Graph-EFM (ms) | 1.76 | 0.77 | 0.75 | 2.55 | 1.09 | 0.82 |
| | Graph-EFM | 1.64 | **0.71** | **0.98** | **2.32** | **1.00** | **0.99** |

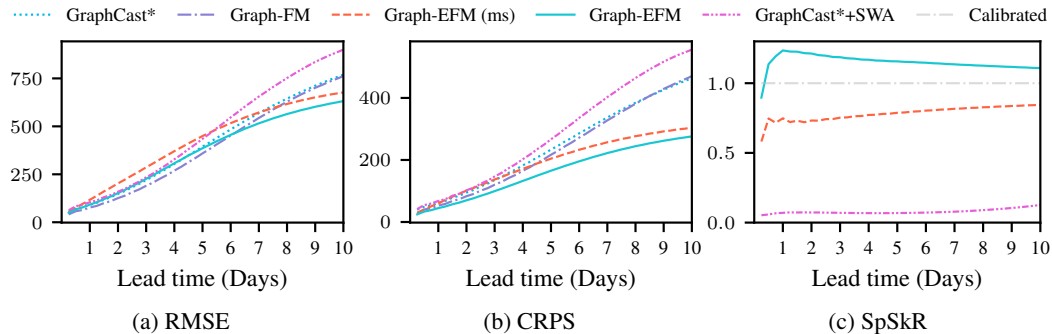

(a) RMSE  (b) CRPS  (c) SpSkR

Figure 3: Results for global forecasting of mean sea level pressure (`msl`) at all lead times.

ensemble mean from Graph-EFM often shows improvements in RMSE over the deterministic models, especially for longer lead times. Across the ensemble models, Graph-EFM achieves lower CRPS values, better capturing the distribution of the weather data. Without any perturbations to initial states Graph-EFM reaches a SpSkR close to 1. We note that GraphCast*+SWAG does not produce useful ensemble forecasts, as these are poorly calibrated and in general do not lead to improved forecast errors. Figure 4 shows an example forecast from Graph-EFM for specific humidity (`q700`) at 10 days lead time. Examples for other variables are given in appendix J.2.

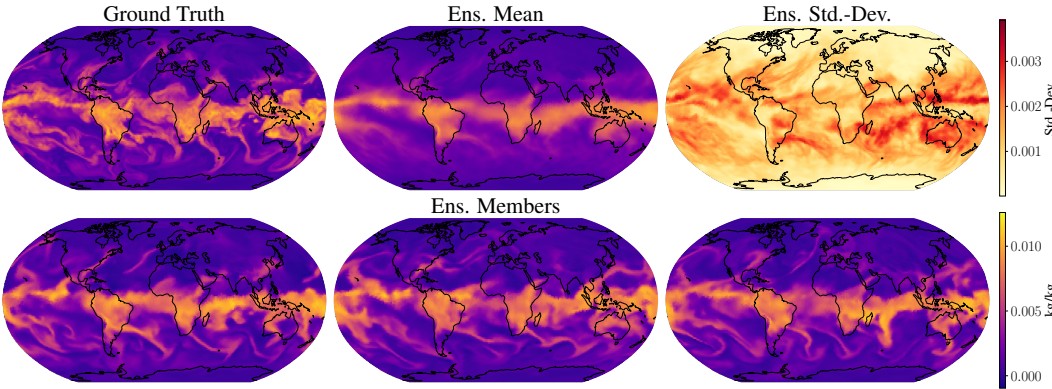

Figure 4: Example Graph-EFM ensemble forecast for specific humidity at 700 hPa (`q700`), for lead time 10 days. The bottom row shows 3 ensemble members, randomly chosen out of the 80.

Table 2: Selection of results for LAM forecasting, including geopotential at 500 hPa (`z500`) and integrated column of water vapor (`wvint`).

| Variable | Model | Lead time 24 h | | | Lead time 57 h | | |
|---|---|---|---|---|---|---|---|
| | | RMSE | CRPS | SpSkR | RMSE | CRPS | SpSkR |
| `z500` | GraphCast* | **153** | 108 | - | **201** | 138 | - |
| | Graph-FM | 230 | 162 | - | 354 | 238 | - |
| | GraphCast*+SWAG | 219 | 136 | 0.08 | 376 | 206 | 0.10 |
| | Graph-EFM (ms) | 400 | 261 | 0.22 | 711 | 470 | 0.23 |
| | Graph-EFM | 172 | **91** | **0.84** | 219 | **115** | **0.75** |
| `wvint` | GraphCast* | **1.51** | 1.01 | - | **2.06** | 1.32 | - |
| | Graph-FM | 1.64 | 1.08 | - | 2.48 | 1.58 | - |
| | GraphCast*+SWAG | 1.78 | 1.17 | 0.05 | 2.34 | 1.50 | 0.05 |
| | Graph-EFM (ms) | 2.39 | 1.43 | 0.16 | 3.51 | 2.12 | 0.13 |
| | Graph-EFM | 1.61 | **0.79** | **0.57** | 2.08 | **1.00** | **0.53** |

**Extreme weather case study**   An important use case for ensemble forecasting is modeling extreme weather events. While higher resolutions than 1.5° are generally desirable for accurately capturing such extremes, we conduct one case study on using Graph-EFM for forecasting hurricane Laura. The full case study with visualized forecasts is available in appendix F. For this example we show that there exists ensemble members accurately predicting the landfall location of the hurricane at 7 days lead time, while the deterministic models still show no sign of the hurricane in the region. Closer to the landfall event the ensemble forecast from Graph-EFM indicates uncertainties associated with the landfall location and wind intensity. This demonstrates the added value of a probabilistic forecasting model.

## 5.2   Limited Area Modeling with MEPS Data

In LAMs weather forecasts are produced for a bounded region of the globe. LAM forecasting allows for higher resolution modeling and regionally tailored model configurations [11], properties that can be inherited by MLWP models by training on LAM data. To model weather over a limited domain, boundary conditions need to be taken into account. In physics-based LAMs these are typically given by a global forecast [38, 44, 7, 32]. We adapt a similar approach for MLWP LAMs, by taking boundary conditions as additional forcing along the boundary of the forecast area. The problem of LAM forecasting is thus about simulating physics not just based on the initial state, but also consistent with these boundary inputs. In the models we introduce $N_b$ additional grid nodes along the area boundary, for the boundary forcing $B^t \in \mathbb{R}^{N_b \times d_x}$. Boundary forcing $B^t$ is always fed together with $X^t$ to the model. Grid nodes on the boundary and within the area are treated identically by the GNN layers. We perform this adaptation to all models in our experiment.

**Data and graphs**   We experiment with a dataset containing 6069 forecasts from the MetCoOp Ensemble Prediction System (MEPS) LAM. Training on forecasts, the goal is here to learn a fast surrogate model for MEPS. We use forecasts started during April 2021 – Jun 2022 for training and validation, and forecasts from July 2022 – March 2023 as a test set. The data is laid out in a $238 \times 268$ grid with spatial resolution 10 km, covering the Nordic region. This dataset contains in total $d_x = 17$ weather variables, some repeated on multiple vertical levels. Forecasts are rolled out with 3 h time steps up to lead time 57 h. In this experiment we take also the boundary forcing directly from the MEPS dataset. We define the boundary as the outermost 10 grid positions. Using the same dataset for the area and boundary allows us to investigate the modeling choices in a controlled experimental setup. In an operational scenario the boundary forcing would instead come from a re-gridded global forecast. In the LAM setting we define our graphs as regular quadrilateral meshes covering the MEPS forecasting area, but with far fewer nodes than the original grid. The graph hierarchy $\mathcal{G}_1, \ldots, \mathcal{G}_L$ is created by constructing such meshes at different resolutions. By placing each node in $\mathcal{G}_l$ at the center of $3 \times 3$ nodes in $\mathcal{G}_{l-1}$, we can merge 4 such graph levels to create $\mathcal{G}_{MS}$. In the hierarchical graph we instead introduce edges from each node in $\mathcal{G}_l$ to the $3 \times 3$ nodes in the level below. More details about the MEPS data and experiment can be found in appendix I.

**Results**   A selection of metrics are shown in table 2 and full results given in appendix K. At these shorter lead times there is no clear benefit of probabilistic modeling in terms of RMSE. Still,

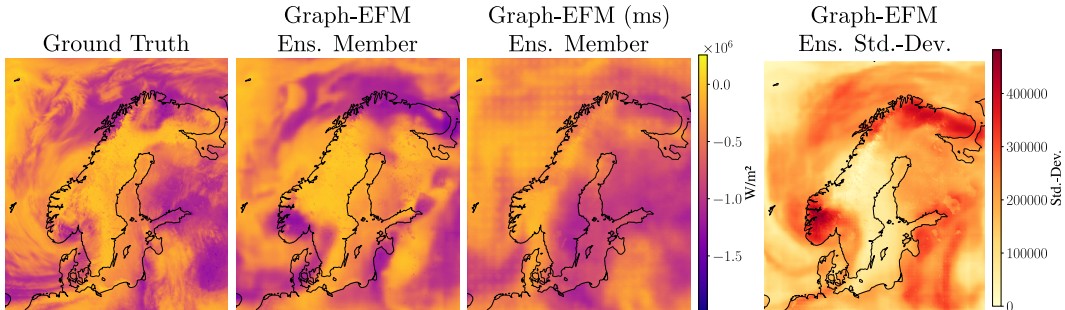

Figure 5: Example forecasts for net solar longwave radiation (`nlwrs`) at lead time 57 h.

as exemplified by the standard-deviation plotted in fig. 5, probabilistic modeling provides useful information about the forecast uncertainty. Comparing the ensemble members in fig. 5 highlights the improved spatial coherency of the hierarchical graph in Graph-EFM. In contrast, the Graph-EFM (ms) forecast looks patchy and lacks physically intuitive features. There are also clear visual artifacts, that can be traced to the multi-scale graph structure. We discuss this more in-depth in appendix G. In the LAM setting all models are under-dispersed, with SpSkR $< 1$. One explanation for this is that the boundary forcing constrains the space of plausible forecasts, hindering the ensemble spread.

## 6    Discussion

In this paper we have explored MLWP ensemble weather forecasting using graph-based latent variable models. Our Graph-EFM model is capable of efficiently producing accurate ensemble forecasts. This paves the way for large-scale MLWP ensemble forecasting both in operational use and research settings. In appendix B we further discuss the societal impact of this research. With this work we hope to emphasize that MLWP models are not just deterministic mappings, but parametrize distributions of weather states. It follows that ensemble forecasting should not be achieved by perturbing models, but by directly modeling the distribution of interest.

**Limitations**    The training process comes with some complications in terms of choosing a training schedule and hyperparameters $\lambda_{\text{KL}}$ and $\lambda_{\text{CRPS}}$. While the CRPS fine-tuning is an important training step, we have found that choosing a too high $\lambda_{\text{CRPS}}$ can introduce visual artifacts, especially for the Graph-EFM (ms) model (see appendix G). While Graph-EFM produces diverse and physically plausible ensemble members, the forecasts still suffer from some of the blurriness common to deterministic models [23, 40]. We here trade off some of the visual fidelity achieved for example by diffusion models [39] for more efficient sampling of ensemble members.

**Future work**    Interesting avenues for future work include learning probabilistic weather models based on other types of autoencoders [48, 49], or by directly optimizing scoring rules [36, 22]. Another approach for achieving efficient ensemble forecasting is to explore techniques for speeding up diffusion model sampling [47].

## Acknowledgments and Disclosure of Funding

This research is financially supported by the Swedish Research Council via the project *Handling Uncertainty in Machine Learning Systems* (contract number: 2020-04122), the Wallenberg AI, Autonomous Systems and Software Program (WASP) funded by the Knut and Alice Wallenberg Foundation, the Excellence Center at Linköping–Lund in Information Technology (ELLIIT), and the project *OWGRE*, funded by partners of the ERA-Net Smart Energy Systems and Mission Innovation through the Joint Call 2020. As such, this project has received funding from the European Union's Horizon 2020 research and innovation programme under grant agreement no. 883973. Our computations were enabled by the Berzelius resource at the National Supercomputer Centre, provided by the Knut and Alice Wallenberg Foundation.

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

## Appendix Table of Contents

## A  Table of Notation

Notation used throughout the paper is listed in table 3.

## B  Societal Impact

**Extreme weather**    Due to climate change the prevalence and severity of extreme weather events is expected to increase substantially, endangering both property and human life [56]. These events are also getting harder to predict and the cost of damages per event has increased nearly 77% over the past five decades [62]. In order to detect extreme events, there is a need for ensemble forecasting to better capture the full distribution of possible weather states. While ensemble forecasting historically has been limited by computational costs [3], efficient MLWP ensemble models have the potential to vastly improve our ability to model extreme weather. In appendix F we present a case study showing how our Graph-EFM could be used for forecasting Hurricane Laura. More extensive evaluation of the abilities of MLWP models to capture extreme events is a complex, but important issue [54, 55]. We view these capabilities as one of the main motivations for further developments of MLWP ensemble forecasting models.

**Forecast failures**    There will always be cases where weather forecasting systems fail, and produce inaccurate predictions for future weather. Depending on the weather event, such forecast errors can have disastrous consequences. In these cases the lack of interpretability of black-box MLWP systems can be a problem, making it hard to understand why the model was wrong. Traditional physics-based systems can also be tough to interpret due to their complexity, but at their core are physical equations understood by researchers. With the rapid progress of MLWP, it is likely that we will soon see a landscape where the physical models take the back seat to a plethora of skillful MLWP forecasting systems. In such a scenario the question of how to investigate forecast failures becomes pressing. It seems desirable to be able to fall back on physical models for understanding impactful events poorly

Table 3: Table of notation

| | |
|---|---|
| $X^t$ | Weather state at time step $t$ |
| $\hat{X}^t$ | Predicted weather state at time step $t$ |
| $X^{t,(m)}$ | Ground truth weather state at time step $t$ in sample $m$ from dataset |
| $\hat{X}^{t,(m)}$ | Predicted weather state at time step $t$ in sample $m$ from dataset |
| $\hat{X}^{t,(m),(k)}$ | Prediction of ensemble member $k$ at time step $t$ in sample $m$ from dataset |
| $F^t$ | Forcing inputs at time step $t$ |
| $Z^t$ | Latent random variable for time step $t$ |
| $B^t$ | Boundary forcing input at time step $t$, for limited area modeling |
| $H$ | Matrix with node representation vectors as rows |
| $E$ | Matrix with edge representation vectors as rows |
| $\boldsymbol{e}_{\alpha \to \beta}$ | Representation vector for the edge going from node $\alpha$ to node $\beta$ |
| $f$ | Autoregressive mapping predicting the next state in deterministic models |
| $g$ | Predictor part of Graph-EFM, predicting the next state conditioned on a sample of $Z^t$ |
| $\tilde{g}$ | Predictor output before skip connection |
| $\mu_Z$ | Mean function of latent map, mapping to the mean of $Z^t$ |
| $T$ | Length of one forecast, in discrete time steps |
| $N$ | Number of grid nodes (grid cells) |
| $N_b$ | Number of grid nodes in the boundary region, for limited area modeling |
| $L$ | Number of levels in hierarchical graph |
| $S$ | Number of samples (forecasts) in dataset |
| $K$ | Ensemble size, number of members in ensemble forecast |
| $d_x$ | Weather state dimensionality (total number of variables) in each grid node |
| $d_z$ | Latent dimensionality, dimensionality of each node or edge representation vector |
| $\mathcal{G}_M$ | Mesh graph |
| $\mathcal{V}_M$ | Nodes of mesh graph |
| $\mathcal{E}_M$ | Edges between the nodes of the mesh graph |
| $\mathcal{V}_G$ | Grid nodes, corresponding to grid cells in the data |
| $\mathcal{G}_{\text{G2M}}$ | Graph connecting grid nodes to the mesh |
| $\mathcal{E}_{\text{G2M}}$ | Edges of $\mathcal{G}_{\text{G2M}}$, going from grid nodes to mesh nodes |
| $\mathcal{G}_{\text{M2G}}$ | Graph connecting the mesh to grid nodes |
| $\mathcal{E}_{\text{M2G}}$ | Edges of $\mathcal{G}_{\text{M2G}}$, going from mesh nodes to grid nodes |
| $\mathcal{G}_{\text{MS}}$ | Multi-scale graph |
| $\mathcal{G}_l$ | Intra-level graph at level $l$ in hierarchical mesh graph |
| $\mathcal{V}_l$ | Nodes at level $l$ of hierarchical mesh graph |
| $\mathcal{E}_l$ | Edges of $\mathcal{G}_l$, intra-level edges at level $l$ in hierarchical mesh graph |
| $\mathcal{G}_{l,l+1}$ | Inter-level graph with edges from level $l$ to level $l+1$ in hierarchical mesh graph |
| $\mathcal{E}_{l,l+1}$ | Edges of $\mathcal{G}_{l,l+1}$, from level $l$ to level $l+1$ in hierarchical mesh graph |
| $\text{Ne}(\beta)$ | Incoming neighbors of node $\beta$, $\{\alpha : (\alpha, \beta) \in \mathcal{E}\}$ |
| $w_\alpha$ | Area weighting for node $\alpha$, proportional to the area of corresponding grid cell |
| $\lambda_{\text{KL}}$ | Weighting for KL-term in variational training objective |
| $\lambda_{\text{CRPS}}$ | Weighting for CRPS term in fine-tuning objective of Graph-EFM |
| $\sigma$ | Standard deviation used in NLL or variational training objective |

---

**Algorithm 1** Single-step prediction $f$ for graph-based MLWP

---

1: $H^G \leftarrow \mathrm{MLP}(X^{t-2:t-1}, F^t)$        ▷ Embedd grid inputs to $d_z$-dimensional vectors
2: $H^M, \cdot \leftarrow \mathrm{GNN}(\mathcal{G}_{\mathrm{G2M}}, H^G, E^{\mathrm{G2M}}, H^M)$        ▷ Map grid representation to mesh
3: **for all** mesh processing steps **do**
4:      $H^M, E^{\mathrm{M2M}} \leftarrow \mathrm{GNN}(\mathcal{G}_M, H^M, E^{\mathrm{M2M}}, H^M)$        ▷ Update mesh representations
5: **end for**
6: $H^G \leftarrow H^G + \mathrm{MLP}(H^G)$
7: $H^G, \cdot \leftarrow \mathrm{GNN}(\mathcal{G}_{\mathrm{M2G}}, H^M, E^{\mathrm{M2G}}, H^G)$        ▷ Map mesh representation back to grid
8: **if** outputs $\sigma$ **then**
9:      $[Y, U] \leftarrow \mathrm{MLP}(H^G)$
10:      **return** $X^{t-1} + Y, \mathrm{Softplus}(U)$        ▷ Return prediction of $X^t$ and $\sigma$ for loss
11: **else**
12:      **return** $X^{t-1} + \mathrm{MLP}(H^G)$        ▷ Return prediction of $X^t$
13: **end if**

---

forecast by MLWP. However, to allow this we can not do away with all infrastructure and expertise related to physical modeling. These are important considerations as weather forecasting moves into an era of operational MLWP.

**Renewable energy**    The production of renewable energy, such as solar and wind power, can be highly volatile [60]. This creates challenges for including these sources in the larger energy system. Accurate weather forecasts, translated into forecasts of energy production, fill an important role as enablers of these energy sources by making their output predictable. Detailed probability estimates from ensemble forecasting can additionally allow for improved cost-loss decision making in these systems.

**The energy footprint of weather forecasting**    Traditional weather forecasting systems utilize massive computing clusters [11], resulting in a substantial energy footprint of the forecasting process. In comparison, MLWP models are highly energy efficient, even when taking into account their initial training process. The total energy required to train the large global MLWP model FourCastNet is comparable to running a 10 day forecast with 50 ensemble members using a traditional forecasting system [37]. Producing a forecast using the same model uses four orders of magnitude less energy than a physics-based model. A similar reduction in energy footprint is to be expected for our models, if applied at the same scale. For ensemble forecasting the total energy saving is even greater, as it is multiplied by the number of ensemble members. However, one has to beware of rebound effects, where efficiency improvements result in more extensive use of resources. If ensemble forecasting becomes 1000 times more energy efficient and we run 1000 times as many ensemble members there is no energy saved in the end.

## C    Model Details

We here give more details about the different models discussed in the main paper.

### C.1   Deterministic graph-based models [23, 20]

Deterministic graph-based MLWP models represent the single-step prediction function $\hat{X}^t = f(X^{t-2:t-1}, F^t)$ as a sequence of MLPs and GNN layers. Algorithm 1 describes the full prediction process. Note that for some of the GNN layers the edge representations are not updated, as these edges are not used in later steps of the process. We denote this by $H, \cdot \leftarrow \mathrm{GNN}(\ldots)$ in algorithm 1. MLPs always act on the last dimension of its concatenated inputs. Mesh node and edge representations are initialized by embedding related static features using additional MLPs. These static features contain information about node and edge positions (see appendices H.1 and I.1). The number of processing steps to run on the mesh is a hyperparameter. Each such step contains one GNN layer, and these GNNs do not share parameters. The mapping $f$ can optionally output also standard deviations $\sigma$, which we use when training with NLL loss (see appendix D.1). We restrict $\sigma$ to be positive by applying the Softplus function.

**Algorithm 2** Single-step prediction $f$ in Graph-FM

1: $H^G \leftarrow \text{MLP}(X^{t-2:t-1}, F^t)$      ▷ Embedd grid inputs to $d_z$-dimensional vectors
2: $H^1, \cdot \leftarrow \text{GNN}(\mathcal{G}_{\text{G2M}}, H^G, E^{\text{G2M}}, H^1)$      ▷ Map from the grid ...
3: **for** $l = 2, \ldots, L$ **do**      ▷ ... and up through the whole mesh hierarchy
4:     $H^l, E^{l-1,l} \leftarrow \text{GNN}(\mathcal{G}_{l-1,l}, H^{l-1}, E^{l-1,l}, H^l)$
5: **end for**
6: **for** number of mesh processing steps / 2 **do**      ▷ Each iteration contains two steps, down and up
7:     $H^L, E^L \leftarrow \text{GNN}(\mathcal{G}_L, H^L, E^L, H^L)$
8:     **for** $l = (L-1), \ldots, 1$ **do**      ▷ Downward pass of sweep
9:       $H^l, E^{l+1,l} \leftarrow \text{GNN}(\mathcal{G}_{l+1,l}, H^{l+1}, E^{l+1,l}, H^l)$
10:       $H^l, E^l \leftarrow \text{GNN}(\mathcal{G}_l, H^l, E^l, H^l)$
11:     **end for**
12:     $H^1, E^1 \leftarrow \text{GNN}(\mathcal{G}_1, H^1, E^1, H^1)$
13:     **for** $l = 2, \ldots, L$ **do**      ▷ Upward pass of sweep
14:       $H^l, E^{l-1,l} \leftarrow \text{GNN}(\mathcal{G}_{l-1,l}, H^{l-1}, E^{l-1,l}, H^l)$
15:       $H^l, E^l \leftarrow \text{GNN}(\mathcal{G}_l, H^l, E^l, H^l)$
16:     **end for**
17: **end for**
18: **for** $l = (L-1), \ldots, 1$ **do**
19:     $H^l, \cdot \leftarrow \text{GNN}(\mathcal{G}_{l+1,l}, H^{l+1}, E^{l+1,l}, H^l)$      ▷ Map down through the mesh hierarchy ...
20: **end for**
21: $H^G \leftarrow H^G + \text{MLP}(H^G)$
22: $H^G, \cdot \leftarrow \text{GNN}(\mathcal{G}_{\text{M2G}}, H^1, E^{\text{M2G}}, H^G)$      ▷ ... and finally to the grid
23: **if** outputs $\sigma$ **then**
24:     $[Y, U] \leftarrow \text{MLP}(H^G)$
25:     **return** $X^{t-1} + Y, \text{Softplus}(U)$      ▷ Return prediction of $X^t$ and $\sigma$ for loss
26: **else**
27:     **return** $X^{t-1} + \text{MLP}(H^G)$      ▷ Return prediction of $X^t$
28: **end if**

## C.2 Graph-FM

In the hierarchical graph there are node representations associated with every level and edge representations associated with each subset of edges. We let $H^l$ be the representations associated with nodes $\mathcal{V}_l$ at level $l$ in the mesh hierarchy. Similarly, let $E^l$ contain representations of intra-level edges $\mathcal{E}_l$, $E^{l,l+1}$ representations of upwards edges $\mathcal{E}_{l,l+1}$, and $E^{l,l-1}$ representations of downward edges $\mathcal{E}_{l,l-1}$. As in the non-hierarchical case, all representations associated with the mesh graph are initialized by MLPs applied to static features. Independent MLPs are used for the different levels of the hierarchy.

Forecasting using Graph-FM follows a similar structure as previous works, but replaces the mappings between the grid and mesh to encompass all levels and changes the processing steps into sweeps through the hierarchy. Recall that for the hierarchical graph we re-define $\mathcal{G}_{\text{G2M}} = (\mathcal{V}_G \cup \mathcal{V}_1, \mathcal{E}_{\text{G2M}})$ and $\mathcal{G}_{\text{M2G}} = (\mathcal{V}_G \cup \mathcal{V}_1, \mathcal{E}_{\text{M2G}})$. The full prediction mapping $f$ for Graph-FM is described in algorithm 2. We define one processing step in Graph-FM as one update to all node and edge representations in the graph, keeping a similar interpretation as processing steps in non-hierarchical graphs. This means that one sweep down and up (lines 7 to 16 in algorithm 2) is counted as two processing steps. It follows that only even numbers of processing steps are reasonable in Graph-FM. Note also here that GNNs only share parameters across nodes and edges in the specific graph they operate on. There is no parameter sharing across processing steps nor levels in the hierarchy. We include Propagation Networks in Graph-FM, but not all parts of the model can be expected to benefit from this change. For some steps of algorithm 2 we want to keep the inductive bias of Interaction Networks to retain information. Specifically, we use Propagation Networks for the mappings directly between the grid and lowest mesh level (lines 2 and 22) and for the upward processing steps (line 14).

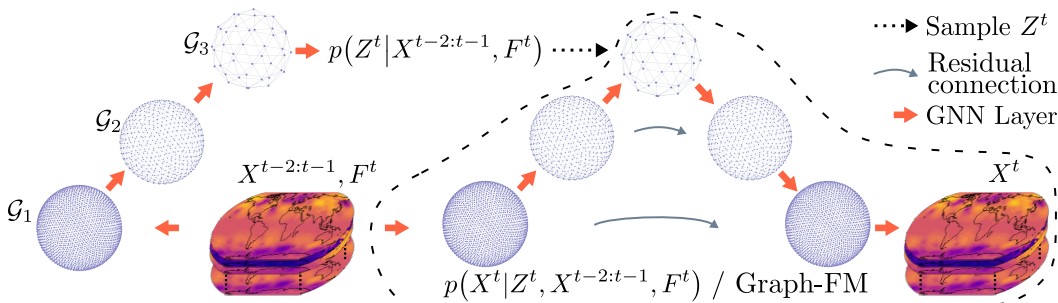

Figure 6: Overview of our Graph-EFM model, with example data and graphs for global forecasting. The corresponding overview for the LAM setting is given in fig. 1. Note that the predictor part of the model has the same structure as one sweep through the hierarchy in Graph-FM.

---

**Algorithm 3** Sampling of ensemble member from $p\big(X^{1:T}\big|X^{-1:0}, F^{1:T}\big)$

---

1: $\hat{X}^{-1:0} \leftarrow X^{-1:0}$
2: **for** $t = 1, \dots, T$ **do**
3:      $Z^t \sim p\Big(Z^t\Big|\hat{X}^{t-2:t-1}, F^t\Big)$               ▷ Sample $Z^t$ from latent map
4:      $\hat{X}^t \leftarrow \hat{X}^{t-1} + g\Big(Z^t, \hat{X}^{t-2:t-1}, F^t\Big)$       ▷ Compute next state using predictor
5: **end for**
6: **return** $\hat{X}^{1:T}$

---

### C.3 Graph-EFM

An overview of the Graph-EFM model with global graphs is shown in fig. 6. At each tim step Graph-EFM takes as input $X^{t-2:t-1}, F^t$ and produces a sample of $X^t$. By feeding the sample from the model back at the next time step, an ensemble member can be rolled out. This process is described in algorithm 3, where we let $\hat{X}^t$ be an initial state for $t < 1$ and then assign it sampled predictions for the actual forecast time steps $1, \dots, T$.

When creating an ensemble of size $K$ from deterministic initial conditions some care has to be taken as to how many members to sample. If we at each time $t$ would sample $K$ values of $X^t$ for each realization of $X^{t-2:t-1}$ we would end up with $K^T$ members at time $T$. Instead, we sample $K$ realizations of $X^1$ given the initial conditions and after that only one realization of $X^t$ for each of the members. This is equivalent to doing $K$ independent runs of algorithm 3. Note that since we can always draw new samples of $Z^t$ we are never restricted in the ensemble size.

**Latent map**    The latent map is defined as

$$p\big(Z^t\big|X^{t-2:t-1}, F^t\big) = \prod_{\alpha \in \mathcal{V}_L} \mathcal{N}\big(Z^t_\alpha\big|\mu_Z\big(X^{t-2:t-1}, F^t\big)_\alpha, I\big) \tag{8}$$

where the mean function $\mu_Z$ is realized as a sequence of GNN layers mapping up through the hierarchy. The exact process is described in algorithm 4. Since only one upward pass is performed we do not update any of the edge representations. The final MLP at line 8 maps from node representations at mesh level $L$ to the mean of $p\big(Z^t\big|X^{t-2:t-1}, F^t\big)$. In the latent map we use exclusively Propagation Networks, encouraging the model to encode useful information from the grid into $Z^t$.

**Predictor**    The predictor $p\big(X^t\big|Z^t, X^{t-2:t-1}, F^t\big)$ is chosen as a dirac measure, with all probability mass concentrated in one point. It thus takes the form of a deterministic mapping

$$\hat{X}^t = g\big(Z^t, X^{t-2:t-1}, F^t\big) = X^{t-1} + \tilde{g}\big(Z^t, X^{t-2:t-1}, F^t\big). \tag{9}$$

This choice emphasizes that all randomness in $p\big(X^t\big|X^{t-2:t-1}, F^t\big)$ should come from the latent variable $Z^t$. An alternative would be to parametrize the predictor as a diagonal Gaussian. We found this inferior in practice, as that would require also sampling from this Gaussian when rolling out

---

**Algorithm 4** Latent map mean function $\mu_Z$

---

1: $H^G \leftarrow \text{MLP}(X^{t-2:t-1}, F^t)$        ▷ Embedd grid inputs to $d_z$-dimensional vectors
2: $H^1, \cdot \leftarrow \text{GNN}(\mathcal{G}_{\text{G2M}}, H^G, E^{\text{G2M}}, H^1)$
3: $H^1, \cdot \leftarrow \text{GNN}(\mathcal{G}_1, H^1, E^1, H^1)$
4: **for** $l = 2, \ldots, L$ **do**        ▷ Propagate up the hierarchy
5:      $H^l, \cdot \leftarrow \text{GNN}(\mathcal{G}_{l-1,l}, H^{l-1}, E^{l-1,l}, H^l)$
6:      $H^l, \cdot \leftarrow \text{GNN}(\mathcal{G}_l, H^l, E^l, H^l)$
7: **end for**
8: **return** $\text{MLP}(H^L)$        ▷ Return mean of $p(Z^t | X^{t-2:t-1}, F^t)$

---

---

**Algorithm 5** Predictor function $g$

---

1: $H^G \leftarrow \text{MLP}(X^{t-2:t-1}, F^t)$        ▷ Embedd grid inputs to $d_z$-dimensional vectors
2: $H^1, \cdot \leftarrow \text{GNN}(\mathcal{G}_{\text{G2M}}, H^G, E^{\text{G2M}}, H^1)$
3: **for** $l = 1, \ldots, L-1$ **do**        ▷ Upward pass of sweep
4:      $H^l, E^l \leftarrow \text{GNN}(\mathcal{G}_l, H^l, E^l, H^l)$
5:      **if** $l = L-1$ **then**
6:          $H^L, \cdot \leftarrow \text{GNN}(\mathcal{G}_{L-1,L}, H^{L-1}, E^{L-1,L}, Z^t)$        ▷ Incorporate $Z^t$ through GNN
7:      **else**
8:          $H^{l+1}, \cdot \leftarrow \text{GNN}(\mathcal{G}_{l,l+1}, H^l, E^{l,l+1}, H^{l+1})$
9:      **end if**
10: **end for**
11: $H^L, \cdot \leftarrow \text{GNN}(\mathcal{G}_L, H^L, E^L, H^L)$
12: **for** $l = (L-1), \ldots, 1$ **do**        ▷ Downward pass of sweep
13:      $H^l, \cdot \leftarrow \text{GNN}(\mathcal{G}_{l+1,l}, H^{l+1}, E^{l+1,l}, H^l)$
14:      $H^l, \cdot \leftarrow \text{GNN}(\mathcal{G}_l, H^l, E^l, H^l)$
15: **end for**
16: $H^G \leftarrow H^G + \text{MLP}(H^G)$
17: $H^G, \cdot \leftarrow \text{GNN}(\mathcal{G}_{\text{M2G}}, H^1, E^{\text{M2G}}, H^G)$
18: **if** outputs $\sigma$ **then**
19:      $[Y, U] \leftarrow \text{MLP}(H^G)$
20:      **return** $X^{t-1} + Y, \text{Softplus}(U)$        ▷ Return prediction of $X^t$ and $\sigma$ for loss
21: **else**
22:      **return** $X^{t-1} + \text{MLP}(H^G)$        ▷ Return prediction of $X^t$
23: **end if**

---

forecasts. This sampling would simply entail adding independent Gaussian noise to the forecasts, which only degrades their quality. We still re-introduce the Gaussian form in the variational objective (eq. (6)), as some choice of likelihood is necessary for a well-defined learning problem. The interpretation of this is that the forecast is rolled out as a latent process with all randomness coming from $Z^t$. Observation noise is then assumed added to this forecast independently at each time step. We use the latent, noise-free forecast as our prediction. Connecting back to the VAE analogue of our model, this setup corresponds to the common practice in the VAE literature of using a Gaussian likelihood but treating the mean as the generated sample [42].

The exact form of the predictor function is described in algorithm 5. It has a similar form as Graph-FM, doing sweeps up and down the mesh hierarchy. We found it sufficient to perform one sweep up and down in the predictor, but it can easily be generalized to multiple sweeps as in Graph-FM. The predictor function optionally returns also the $\sigma$ to be used in our variational objective. In the predictor we use Interaction Networks in the upward direction and Propagation Networks in the downward direction. This is to guarantee that randomness and useful information encoded in $Z^t$ at the top level reaches all the way down to the grid. Note that on line 6 $Z^t$ is combined with $H^{L-1}$ through the Interaction Network layer mapping from level $L-1$ to $L$. For the details of how this combination happens, see eq. (1), where $Z^t$ takes the role of the receiver node representations $H^R$. Note that edge and node representations updated on line 4 in the upward sweep then re-appear in the downward pass. This constitutes residual connections between the upward and downward passes, as illustrated in fig. 6.

---

**Algorithm 6** Parameter mappings $\mu_q, \sigma_q$ of variational approximation

---

1: $H^G \leftarrow \mathrm{MLP}(X^{t-2:t-1}, X^t, F^t)$      ▷ Embedd grid inputs (including $X^t$) to $d_z$-dim. vectors
2: $H^1, \cdot \leftarrow \mathrm{GNN}(\mathcal{G}_{\mathrm{G2M}}, H^G, E^{\mathrm{G2M}}, H^1)$
3: $H^1, \cdot \leftarrow \mathrm{GNN}(\mathcal{G}_1, H^1, E^1, H^1)$
4: **for** $l = 2, \ldots, L$ **do**      ▷ Propagate up the hierarchy
5:      $H^l, \cdot \leftarrow \mathrm{GNN}(\mathcal{G}_{l-1,l}, H^{l-1}, E^{l-1,l}, H^l)$
6:      $H^l, \cdot \leftarrow \mathrm{GNN}(\mathcal{G}_l, H^l, E^l, H^l)$
7: **end for**
8: $[Y, U] \leftarrow \mathrm{MLP}(H^L)$
9: **return** $Y, \mathrm{Softplus}(U)$      ▷ Return mean and standard deviation of $q(Z^t | X^{t-2:t-1}, X^t, F^t)$

---

**Variational approximation**    Defining the ELBO for the model requires a variational approximation $q(Z^t | X^{t-2:t-1}, X^t, F^t)$ at each time step, approximating the true posterior $p(Z^t | X^{t-2:t-1}, X^t, F^t)$ over $Z^t$. Due to the autoregressive structure of the model, it is sufficient to condition $q$ on states at time steps $t-2$, $t-1$ and $t$. This conditioning removes all dependence between $Z^t$ and other variables (see fig. 2), simplifying the parametrization and training of $q$. We choose the variational approximation to have a similar Gaussian form as the latent map

$$q(Z^t | X^{t-2:t-1}, X^t, F^t) =$$
$$\prod_{\alpha \in \mathcal{V}_L} \prod_{j=1}^{d_z} \mathcal{N}\left(Z_{\alpha,j}^t \middle| \mu_q(X^{t-2:t-1}, X^t, F^t)_{\alpha,j}, \sigma_q(X^{t-2:t-1}, X^t, F^t)_{\alpha,j}^2\right). \tag{10}$$

For added flexibility we also model the variance of $q$. Here both $\mu_q$ and $\sigma_q$ are functions mapping from the grid inputs to $|\mathcal{V}_L| \times d_z$ matrices. These functions are implemented jointly as described in algorithm 6. This mapping is similar to the latent map, with the differences being that $X^t$ is taken as an input and both mean and standard deviation returned. Similarly to the latent map, all GNNs used in the variational approximation are Propagation Networks.

## C.4    MLP parametrization

Following Lam et al. [23], all MLPs have one hidden layer with Swish activation functions [59]. With the exception of MLPs outputting predictions or distribution parameters, we apply LayerNorm [53] to all other MLP outputs.

While MLPs in different GNN layers never share parameters, we use some parameter sharing across the embedding MLPs in the different components of Graph-EFM. Specifically, the latent map and predictor share parameters for the MLP embedding the grid input (line 1 in algorithm 4 and line 1 in algorithm 5). The MLPs that embedd static graph features, for initializing graph representation vectors, are also shared across all components of the Graph-EFM model.

## C.5    Baseline models

**GraphCast\***    GraphCast\* is a reimplementation of the GraphCast model [23] in our codebase. This was done to allow for a fair comparison by using the same data and multi-scale graphs as other models in our experiments.

**Graph-EFM (ms)**    The Graph-EFM (ms) model is a version of Graph-EFM, but using a multi-scale mesh graph $\mathcal{G}_{\mathrm{MS}}$ [23], instead of the hierarchical one. We replace the sweeps up and down the hierarchy with multiple processing steps performed on the same mesh graph, similar to the deterministic case described in algorithm 1. In the predictor of this model, the latent $Z^t$ is integrated already in the mapping from the grid to the mesh as

$$H^M, \cdot \leftarrow \mathrm{GNN}(\mathcal{G}_{\mathrm{G2M}}, H^G, E^{\mathrm{G2M}}, Z^t). \tag{11}$$

Note that as $Z^t$ here is associated with the nodes of $\mathcal{G}_{\mathrm{MS}}$, it has shape $|\mathcal{V}_1| \times d_z$ as opposed to $|\mathcal{V}_L| \times d_z$ for Graph-EFM. As $\mathcal{V}_1$ has many more nodes, the total dimensionality of $Z^t$ is higher for Graph-EFM (ms). This is a consequence of the multi-scale graph offering no further dimensionality reduction beyond mapping from the grid to the mesh.

**GraphCast\*+SWAG** One simple way to create an MLWP ensemble is by re-training multiple deterministic models from random initializations. Training many large weather models from scratch is however infeasible in practice. It also limits the number of ensemble members to the number of models trained. Inspired by Graubner et al. [16], we create a multi-model ensemble baseline using SWAG [30]. SWAG is a technique for approximate Bayesian inference based on running Stochastic Gradient Descent (SGD) training with a constant high learning rate. At regular intervals throughout this optimization process the model parameters are saved. The full SWAG process includes estimating a high-dimensional Gaussian over these parameters and drawing samples to create the ensemble. This gets around the limitation of needing to decide the maximum ensemble size when training the model. As we never require more ensemble members than saved during the SGD training we skip estimating this Gaussian and resampling. Instead, we directly use a subset of the saved model parameters to create our ensemble members. Graubner et al. [16] combine SWAG with initial state perturbations, but as we model the conditional distribution $p\left(X^{1:T}|X^{-1:0}, F^{1:T}\right)$ we do not include any such perturbations to the initial states.

To create our GraphCast\*+SWAG ensemble we start from the trained GraphCast\* model and run SGD training for $T = 8$ unrolled steps and with a fixed learning rate ($10^{-3}$ for the global experiment and $5 \times 10^{-5}$ for MEPS). Even higher learning rates led to numerical issues or the model training diverging. Model parameters are then saved every 100 gradient descent steps. During evaluation these parameters are loaded one after another and used to produce the forecast for each ensemble member. The error and spread of the GraphCast\*+SWAG ensemble is highly dependent on the correlation of consecutively saved model parameters. Given the poor spread of GraphCast\*+SWAG we tested also saving parameters with 1000 steps in-between, but this led to similar results.

## D  Training Objectives

We here give detailed definitions of the objectives used to train our models.

### D.1  Deterministic models

**Weighted MSE** Deterministic forecasting models can be trained by minimizing a weighted MSE [23] of rolled-out forecasts

$$\mathcal{L}_{\text{WMSE}} = \frac{1}{TN} \sum_{t=1}^{T} \sum_{\alpha \in \mathcal{V}_G} \sum_{j=1}^{d_x} w_\alpha \omega_j \lambda_j \left( \hat{X}_{\alpha,j}^t - X_{\alpha,j}^t \right)^2 \tag{12}$$

where we recall that $N = |\mathcal{V}_G|$ and:

- $w_\alpha$ is the normalized area weighting for grid cell $\alpha$. These are computed as described by Rasp et al. [40].

- $\lambda_j$ is the inverse variance of time differences for variable $j$. These can be computed directly from iterating over the data.

- $\omega_j$ is a weight associated with the vertical level of variable $j$. For global experiments we use the same weights $\omega_j$ as Lam et al. [23].

**Negative Log-Likelihood loss** For the MEPS data however, choosing weights $\omega_j$ is challenging due to the many different variables and their irregular vertical levels. In such cases, an alternative approach is to use an NLL loss, where the model itself determines the trade-offs between different variables [8]. As the MEPS area has grid cells of approximately equal size we can also simply set $w_\alpha = 1$. For the MEPS experiments we thus use

$$\mathcal{L}_{\text{NLL}} = -\frac{1}{TN} \sum_{t=1}^{T} \sum_{\alpha \in \mathcal{V}_G} \sum_{j=1}^{d_x} w_\alpha \log \mathcal{N}\left( X_{\alpha,j}^t \Big| \hat{X}_{\alpha,j}^t, \sigma_{\alpha,j}^2 \right). \tag{13}$$

The standard deviation $\sigma_{\alpha,j}$ is here a second output from the mapping $f$. When unrolling forecasts we do not sample from this Gaussian, but feed the mean $\hat{X}^t$ to the next time steps. Note that eq. (13) decomposes into weighted squared error terms and log-determinant terms preventing too large $\sigma$

outputs. The weighted MSE in eq. (12) can therefore also be viewed as a special case of the NLL loss, up to an additive constant. This relationship corresponds to eq. (13) with constant variances given by

$$\sigma_{\alpha,j}^2 = \frac{1}{2\omega_j\lambda_j}. \tag{14}$$

## D.2 Probabilistic model

For the probabilistic model we can not directly apply the training objectives above, as they only seek to match the first moments of the model distribution to the data. To train the probabilistic model we instead leverage the fact that the single-step model has a structure similar to a (conditional) VAE [21, 45]. We can therefore optimize a variational objective, the ELBO, to match the distribution of the data.

**Variational objective**  For a single time step, our variational objective is

$$\tilde{\mathcal{L}}_{\text{Var}}\big(X^{t-2:t-1}, X^t, F^t\big) = \lambda_{\text{KL}} D_{\text{KL}}\big(q\big(Z^t\big|X^{t-2:t-1}, X^t, F^t\big)\big\|p\big(Z^t\big|X^{t-2:t-1}, F^t\big)\big)$$
$$-\mathbb{E}_{q(Z^t|X^{t-2:t-1}, X^t, F^t)}\Big[\sum_{\alpha\in\mathcal{V}_G}\sum_{j=1}^{d_x} w_\alpha \log\mathcal{N}\Big(X_{\alpha,j}^t\Big|g\big(Z^t, X^{t-2:t-1}, F^t\big)_{\alpha,j}, \sigma_{\alpha,j}^2\Big)\Big]. \tag{15}$$

We include a weight $\lambda_{\text{KL}}$ in front of the KL-regularizer to allow for some tuning between the two parts of the objective function [61]. In practice we found this useful to make sure the model does not collapse to pure deterministic predictions. Note that when $\lambda_{\text{KL}} = 1$ this is exactly equal to the (negative) ELBO. The NLL from eq. (13) here shows up again in the second term. Similarly to the deterministic models, $\sigma_{\alpha,j}$ can either be output by the model (the predictor in this case) or chosen as a constant. For global experiments we choose $\sigma_{\alpha,j}$ as in eq. (14), reducing the term to a weighted MSE loss. For the MEPS data we again let each $\sigma_{\alpha,j}$ be output by the model. Equation (15) can in practice be computed efficiently by using a single-sample Monte Carlo estimate for the expectation and the reparametrization trick [21]. The KL-divergence is available in closed form as both distributions are Gaussian. As with deterministic models [23, 20, 8, 34], it is crucial to fine-tune on rolled out forecasts of multiple time steps. This induces stability and improves performance for longer lead times. We thus define our fine-tuning objective as

$$\mathcal{L}_{\text{Var}} = \sum_{t=1}^T \tilde{\mathcal{L}}_{\text{Var}}\Big(\hat{X}^{t-2:t-1}, X^t, F^t\Big) \tag{16}$$

where $\hat{X}^t$ is an initial state from the dataset for $t < 1$ and otherwise sampled using the variatonal approximation for $Z^t$. In eq. (16) $X^t$ is always from the training data.

**CRPS fine-tuning**  In practice we found that fine-tuning the model with only $\mathcal{L}_{\text{Var}}$ tends to result in underdispersed ensemble forecasts, underestimating the variance of the distribution. To remedy this, we include an additional fine-tuning objective based on the CRPS [15, 63, 22]. The CRPS measures how well a univariate distribution matches the observed data and is defined as

$$\text{CRPS}(p(x), y) = \mathbb{E}[|x - y|] - \frac{1}{2}\mathbb{E}[|x - x'|] \tag{17}$$

where $x, x'$ are independent copies of a random variables with distribution $p(x)$ and $y$ the observed data. The CRPS is minimized only when the predicted distribution matches that of the data [15], making it suitable for calibrating our ensemble forecasting model. We define our CRPS fine-tuning loss as an unbiased two-sample estimator, summed over all dimensions of the predictive distribution

$$\mathcal{L}_{\text{CRPS}} = \sum_{t=1}^T \sum_{\alpha\in\mathcal{V}_G}\sum_{j=1}^{d_x} w_\alpha \frac{1}{2}\Big(\Big|\hat{X}_{\alpha,j}^t - X_{\alpha,j}^t\Big| + \Big|\check{X}_{\alpha,j}^t - X_{\alpha,j}^t\Big| - \Big|\hat{X}_{\alpha,j}^t - \check{X}_{\alpha,j}^t\Big|\Big) \tag{18}$$

with $\hat{X}^t$ and $\check{X}^t$ coming from two independent ensemble members sampled from the model.

Note that the CRPS objective targets only the marginal distributions, while spatio-temporally coherent samples requires matching the joint distribution of the data. To avoid degenerate solutions we weight $\mathcal{L}_{\text{CRPS}}$ by $\lambda_{\text{CRPS}}$ and combine it with the variational objective for the final fine-tuning loss $\mathcal{L} = \mathcal{L}_{\text{Var}} + \lambda_{\text{CRPS}}\mathcal{L}_{\text{CRPS}}$. Computing $\mathcal{L}$ requires rolling out three forecasts, one for $\mathcal{L}_{\text{Var}}$ (using samples from $q$) and two for $\mathcal{L}_{\text{CRPS}}$ (using samples from the latent map). This can have a substantial

memory cost. It is however sufficient to include the CRPS term only in the final steps of the training process, making this less of an issue in practice.

Empirically we found that including the CRPS objective improves both ensemble calibration and forecast accuracy. The accuracy improvement can be attributed to the MAE terms in eq. (18). Note that in $\mathcal{L}_{\text{Var}}$ forecasts are rolled out by sampling $Z^t$ from the variational approximation, while in $\mathcal{L}_{\text{CRPS}}$ these samples are from the latent map. This matches how forecasting is carried out at test time. As a consequence the CRPS loss has an additional effect of bridging the gap between the variational training and the final forecasting.

# E   Metric Definitions

We here give full definitions for the metrics used to evaluate our models.

## E.1   Deterministic Metrics

We define evaluation metrics based on set of $S$ forecasts, each a sequence $\hat{X}^{1,(m)}, \ldots, \hat{X}^{T,(m)}$ of predicted weather states. Metrics are computed per lead-time and variable. For deterministic forecasting we define the RMSE of variable $j$ at lead time $t$ as

$$\text{RMSE}_{t,j} = \sqrt{\frac{1}{SN} \sum_{m=1}^{S} \sum_{\alpha \in \mathcal{V}_G} w_\alpha \left( \hat{X}_{\alpha,j}^{t,(m)} - X_{\alpha,j}^{t,(m)} \right)^2}. \tag{19}$$

In all metrics we include area weighting $w_\alpha$ to handle the fact that the grid cells for the global data has different area. These weights are computed as described in Rasp et al. [40]. For the MEPS data these are all set to $w_\alpha = 1$. Note that the square root is applied after sample averaging, following standard convention and the WeatherBench 2 benchmark [40].

## E.2   Probabilistic Metrics

In the probabilistic setting we evaluate an ensemble forecast of $K$ members. Let $\hat{X}^{t,(m),(k)}$ be the prediction of ensemble member $k$. For ensemble forecasts we compute the RMSE of the ensemble mean as

$$\text{RMSE}_{t,j} = \sqrt{\frac{1}{SN} \sum_{m=1}^{S} \sum_{\alpha \in \mathcal{V}_G} w_\alpha \left( \bar{X}_{\alpha,j}^{t,(m)} - X_{\alpha,j}^{t,(m)} \right)^2} \tag{20}$$

$$\bar{X}^{t,(m)} = \frac{1}{K} \sum_{k=1}^{K} \hat{X}^{t,(m),(k)}. \tag{21}$$

To measure the calibration of uncertainty expressed by the ensemble we use the (bias-corrected) Spread-Skill-Ratio

$$\text{SpSkR}_{t,j} = \sqrt{\frac{K+1}{K}} \frac{\text{Spread}_{t,j}}{\text{RMSE}_{t,j}} \tag{22a}$$

$$\text{Spread}_{t,j} = \sqrt{\frac{1}{SKN} \sum_{m=1}^{S} \sum_{k=1}^{K} \sum_{\alpha \in \mathcal{V}_G} w_\alpha \left( \bar{X}_{\alpha,j}^{t,(m)} - \hat{X}_{\alpha,j}^{t,(m),(k)} \right)^2}. \tag{22b}$$

If the ensemble members represent realistic forecasts, and observe exchangeability with the ground truth, $\text{SpSkR}_{t,j} \approx 1$ [13].

As a probabilistic metric we also use the CRPS [15], here computed as a finite sample estimate [63] over all the ensemble members

$$\text{CRPS}_{t,j} = \frac{1}{SN} \sum_{m=1}^{S} \sum_{\alpha \in \mathcal{V}_G} w_\alpha \left( \frac{1}{K} \sum_{k=1}^{K} \left| \hat{X}_{\alpha,j}^{t,(m),(k)} - X_{\alpha,j}^{t,(m)} \right| \right.$$
$$\left. - \frac{1}{2K(K-1)} \sum_{k=1}^{K} \sum_{k'=1}^{K} \left| \hat{X}_{\alpha,j}^{t,(m),(k)} - \hat{X}_{\alpha,j}^{t,(m),(k')} \right| \right). \tag{23}$$

Note that this is a spatial average over marginal CRPS values and does not take any covariance structure of the data or model distributions into account. To avoid the quadratic sum in eq. (23) we in practice use the idea of Zamo and Naveau [63] to compute this with linear memory by sorting the ensemble members. For deterministic models the CRPS reduces to MAE. See this for example from eq. (23) by letting all ensemble members be the same predicted value.

## F   Extreme Weather Case Study: Hurricane Laura

One key motivations for probabilistic weather forecasting is to allow for better predictions of extreme events and improved estimates of uncertainties associated with these. To exemplify this we include here a case study looking at forecasts for Hurricane Laura, using the global models.

In August of 2020 Laura developed in the Atlantic Ocean and reached hurricane levels in the Gulf of Mexico, eventually making landfall in Louisiana and causing major damages [58]. We study here forecasts for 2020–08–27T12 UTC, which is 6 hours after the hurricane hit land. All forecasts are for this exact time point, but initialized a varying number of days before. We here run 50 member ensemble forecasts using Graph-EFM and deterministic forecasts using the GraphCast* and Graph-FM models. Figure 7 shows 10 m wind speeds in ERA5 and the forecasts. For Graph-EFM we plot both forecasts from randomly sampled ensemble members and a cherry-picked best member that was deemed to most closely match ERA5. Note that the 1.5° resolution that we work with makes determining similarity or exact positions somewhat challenging. The resolution also puts some limitations on how well the most extreme wind speeds can be captured.

We see in fig. 7 that at 7 days lead time there is great uncertainty, and the deterministic models do not show the hurricane at all. In the ensemble forecast from Graph-EFM there exists however already members indicating the possibility of a hurricane making landfall a week ahead (see for example the cherry-picked "Best member"). While the ensemble includes many possible scenarios, a total of 7 members show the development of a hurricane in the area. Having information of such possible scenarios a long time ahead allows for planning and readying disaster response efforts that might be needed. Note that discovering these scenarios is only possible through an ensemble forecast, as the deterministic models do not indicate such an event. At 5 days lead time all models are indicating the development of a hurricane. While the deterministic models do a good job here, they are indicating a landfall location slightly too far eastward. The ensemble members from Graph-EFM however show a range of different positions for the hurricane, indicating the uncertainty in the landfall location. At 3 days ahead 42 out of 50 ensemble members show the hurricane making landfall, but still with some uncertainty about the exact location. At 1 day lead time all models give an accurate forecast of the position of the hurricane. Apart from position, it is also interesting to consider how the models capture the intensity in terms of wind speeds. At 1 day ahead the deterministic models are somewhat underestimating the wind speed. The Graph-EFM ensemble shows a range of possible values, indicating the uncertainty in the exact wind intensity. Overall this study exemplifies how ensemble forecasts from a machine learning model can be used to discover possible extreme weather scenarios at long lead times and uncover the uncertainties associated with them.

## G   Spatial Coherency of Forecasts

Apart from giving a low forecast error, it is also important for forecasts to look realistic. Specifically, we want forecasts to be *physical*, containing features that are possible under laws of atmospheric physics. This is necessary for forecasts to be interpretable by meteorologists and builds trust in the MLWP model. One key property for realistic forecasts is that of spatial coherency. Forecasts should take realistic values not just locally, in each grid cell, but show larger-scale features that are consistent over the forecasting area. This is important since the weather system contains spatial dependencies on multiple different scales. Note that in the probabilistic setting this connects closely to capturing the correct joint distribution spatially, rather than just the marginal distributions in each grid cell. As the metrics commonly used to evaluate MLWP models only consider the marginal distributions, it is important to also inspect sample forecasts to gauge the spatial coherency.

This modeling aspect is especially relevant for probabilistic models, since being able to visualize and interpret individual ensemble members is an important capability of ensemble forecasting. Other uncertainty quantification techniques can be applied to estimate forecast uncertainty [6], but these

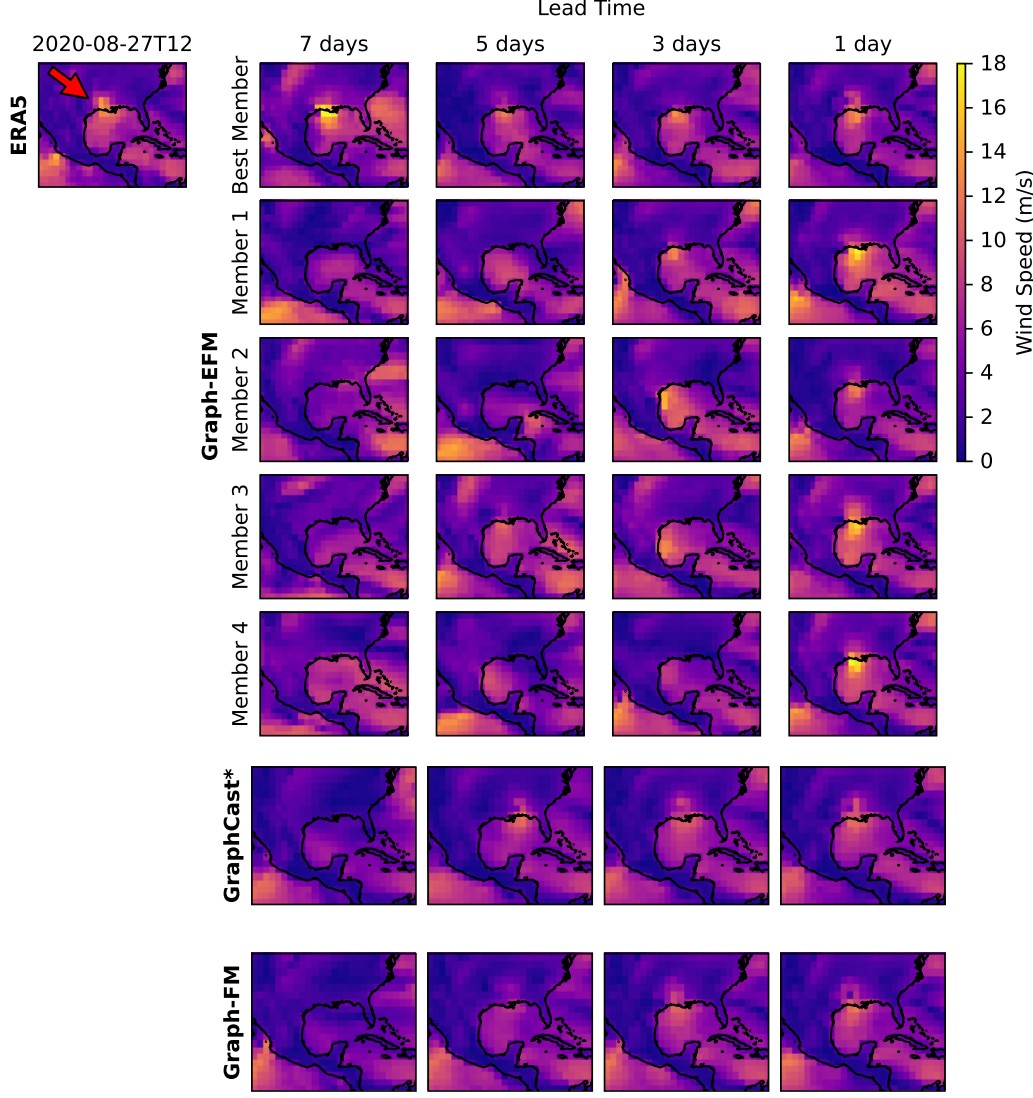

Figure 7: Example forecasts of 10 m wind speeds during Hurricane Laura, all for 2020–08–27T12 UTC. The first column shows the wind speeds in ERA5, with the red arrow indicating the location of the hurricane. Remaining columns show model forecasts for the specific time, initialized at different lead times ahead. For Graph-EFM we plot both 4 randomly sampled ensemble members and one cherry-picked member that was deemed to best match ERA5.

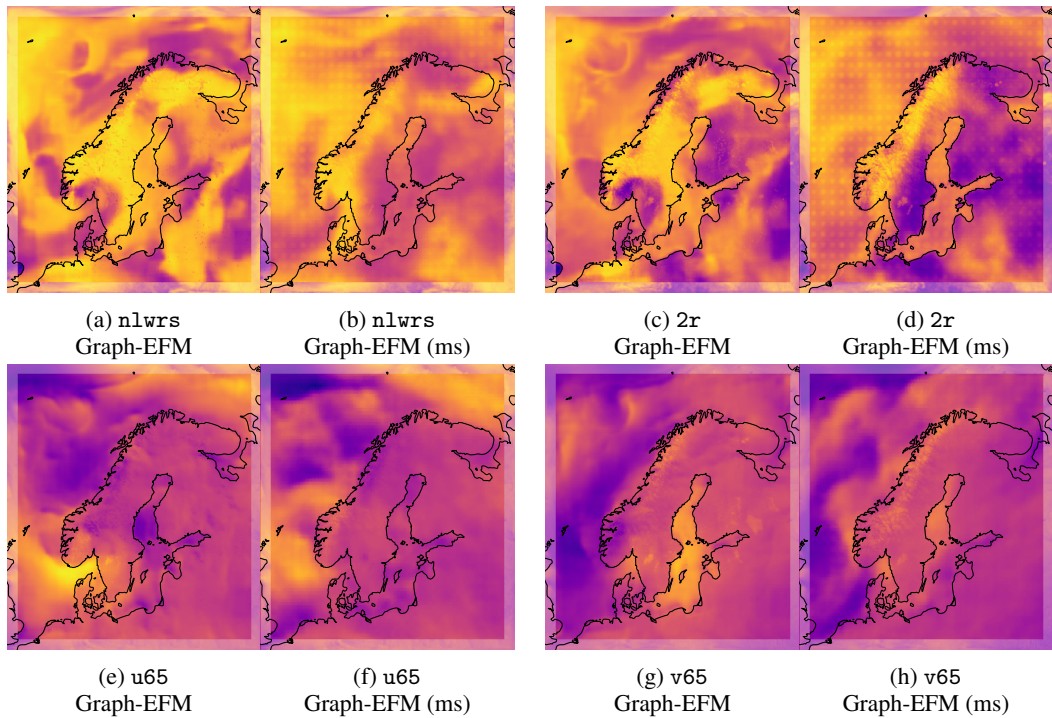

|     |     |     |     |
|-----|-----|-----|-----|
| (a) `nlwrs` Graph-EFM | (b) `nlwrs` Graph-EFM (ms) | (c) `2r` Graph-EFM | (d) `2r` Graph-EFM (ms) |
| (e) `u65` Graph-EFM | (f) `u65` Graph-EFM (ms) | (g) `v65` Graph-EFM | (h) `v65` Graph-EFM (ms) |

Figure 8: Example forecasts from Graph-EFM and Graph-EFM (ms) trained on the MEPS data. All forecasts are from single ensemble members, plotted for lead time 57 h.

generally do not generate samples of possible future weather states. Having samples to inspect can build trust and understanding of the model. If one member is predicting an extreme event, it is possible to inspect this forecast and see what sequence of events in the atmosphere could lead to this outcome.

For the final Graph-EFM model forecasts are generally spatially coherent and represent physically plausible scenarios. Examples of this is shown in appendices J.2 and K.2. In this appendix we contrast this with some of the shortcomings of baseline models. We also discuss some of the challenges that we have encountered with achieving spatially coherent forecasts and ways to tackle these.

### G.1   Limited Area Modeling with MEPS Data

In the MEPS experiment the benefits of the hierarchical graph structure become the most clear when visually inspecting forecasts. Figure 8 shows a comparison between a few forecasts from Graph-EFM and Graph-EFM (ms). In the Graph-EFM (ms) model samples tend to be more patchy and poorly reproduce spatial features. We connect this to the fact that the randomness in $Z^t$ is more local, as it is associated with the nodes in $\mathcal{G}_{MS}$ which directly connect to the grid. While the randomness introduced from sampling $Z^t$ can spread to the full forecast area using the multi-scale graph, this is something the model has to explicitly learn. For matching marginal distributions in grid cells it can be enough for Graph-EFM (ms) to just have the randomness in each mesh node impact the connected grid nodes locally. We can contrast this to Graph-EFM, where $Z^t$ is associated with nodes at the top of the hierarchical graph. For this randomness to affect the prediction it must necessarily be propagated down the hierarchy, which disperses it spatially by construction.

We have observed that the CRPS fine-tuning (see appendix D.2) can be central to the performance of Graph-EFM. This aligns the model distribution with the distribution of the data. However, since this objective only encourages matching of the marginal distributions it is not sufficient for making the model capture the full joint distribution. This has to be achieved by constraints to the model or through additional parts of the training objective. We have observed that the weighting $\lambda_{CRPS}$ used for the CRPS term can have a large impact on the spatial coherency of forecasts. When $\lambda_{CRPS}$ is chosen too high the models trade off the ability to generate meaningful large-scale patterns for capturing

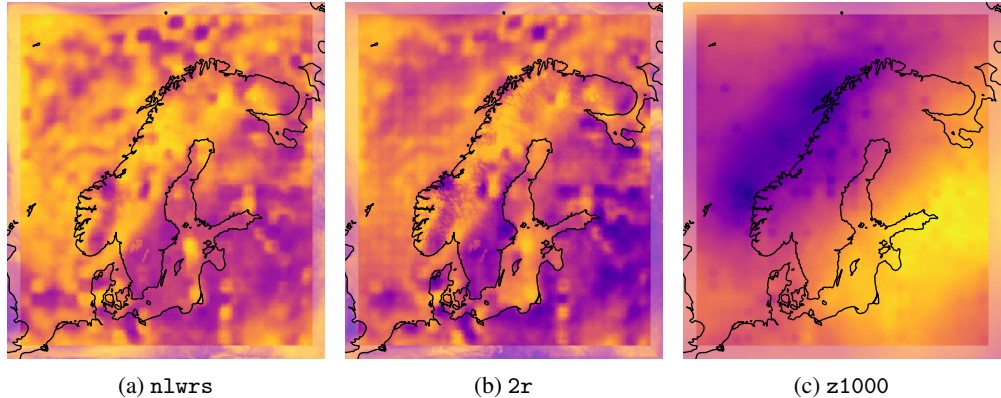

(a) `nlwrs`               (b) `2r`               (c) `z1000`

Figure 9: Examples of artifacts appearing in Graph-EFM (ms) forecasts when trained with too high $\lambda_{\text{CRPS}}$ on the MEPS data. The checkerboard-like pattern can be related to the layout of the multi-scale graph used.

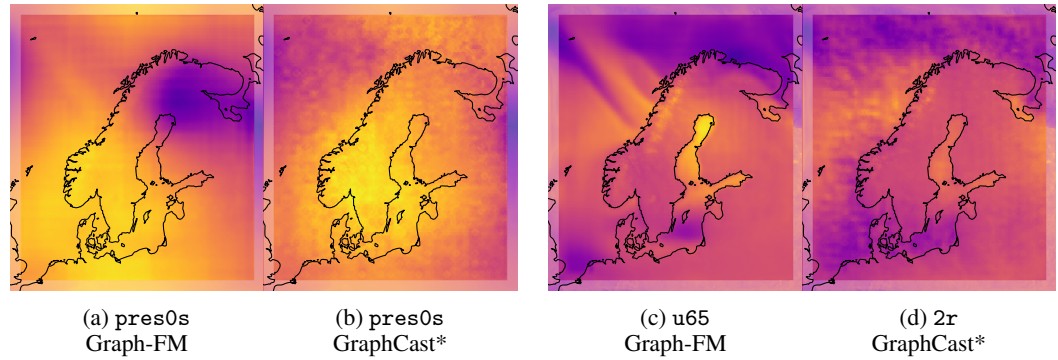

(a) `pres0s`     (b) `pres0s`          (c) `u65`     (d) `2r`
Graph-FM     GraphCast*          Graph-FM     GraphCast*

Figure 10: Examples of artifacts in smaller versions of the deterministic models. The circular artifacts in GraphCast* become much more prevalent when the model capacity is limited. These models use 4 processing steps, $d_z = 64$ and are trained using NLL loss.

local variations. This is especially severe for the Graph-EFM (ms) model, forcing us to use lower $\lambda_{\text{CRPS}}$ values in order to still get physical-looking forecasts. Figure 9 shows an example of the type of local noise-patterns that can appear when training Graph-EFM (ms) with a too high $\lambda_{\text{CRPS}}$.

Another problem observed with the multi-scale graph $\mathcal{G}_{\text{MS}}$ in the LAM setting is the appearance of circular artifacts over the forecasting area. These appear in samples from Graph-EFM (ms) (see figs. 8b and 8d), but can also be found in deterministic forecasts from GraphCast*. These artifacts can be traced to the heterogeneous structure of the multi-scale graph. Due to how $\mathcal{G}_{\text{MS}}$ is constructed, mesh nodes can have different number of neighbors. The artifact positions match the positions of mesh nodes with many neighbors. Note that the hierarchical graph does not have this issue, as $\mathcal{G}_1$ (that connects to the grid) has a more uniform structure. We noticed that these artifacts are even more prevalent for smaller models, as shown in fig. 10. This points to the fact that with increased capacity, the GraphCast* model can to some extent learn to compensate for this problem.

## G.2 Global Forecasting with ERA5

We have found it less challenging to achieve spatially coherent forecasts in the global setting. Both Graph-EFM and Graph-EFM (ms) generally produce samples with realistic physical features. Early in the training process we observe some hexagonal patterns for longer lead times (also noted by Keisler [20]), but these disappear as we train the models on longer rollouts.

One challenge that we encountered with the global probabilistic models was the appearance of small areas of instabilities when rolling out the models to longer lead times. In fig. 11 we show some examples of this in forecasts from preliminary models. These unphysical artifacts typically covered

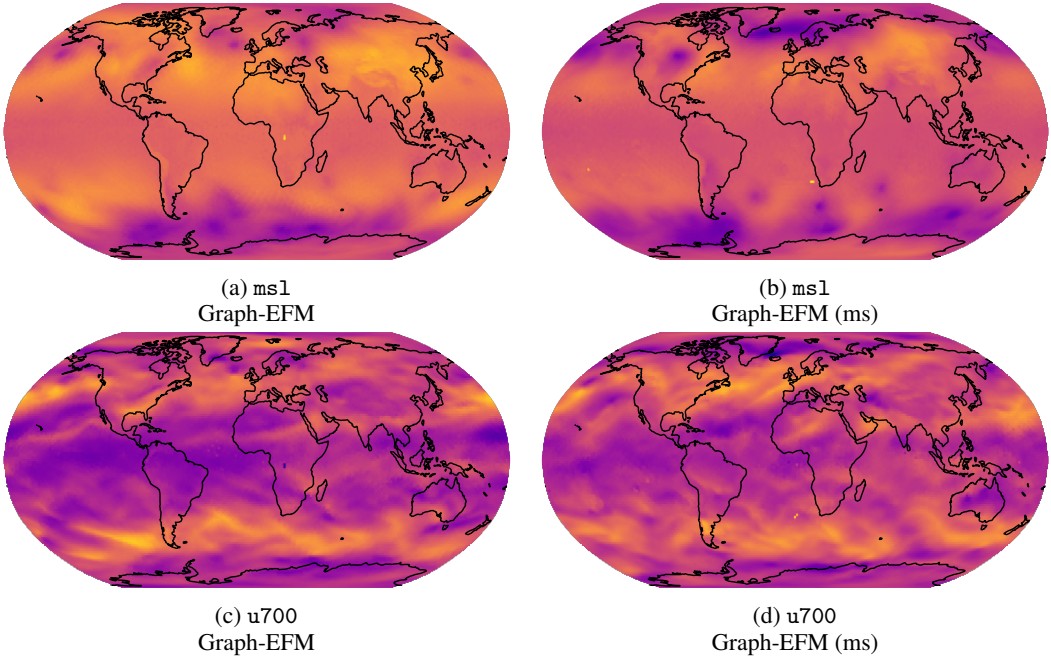

(a) `msl`
Graph-EFM

(b) `msl`
Graph-EFM (ms)

(c) `u700`
Graph-EFM

(d) `u700`
Graph-EFM (ms)

Figure 11: Examples of artifacts in preliminary version of the global models unrolled to 10 days lead time. Small patches of deviating values appear and impact multiple variables. While these were more prevalent in Graph-EFM (ms), we did also observe this issue in Graph-EFM.

only a few grid-cells, but are of course undesirable in a forecasting model. To remedy this for Graph-EFM we found it sufficient to train on longer rollouts. The stabilizing effect of unrolling up to $T = 12$ time steps during training did suppress these issues. This was however not enough to solve the problem for Graph-EFM (ms), for which we additionally needed to further lower $\lambda_{\text{CRPS}}$.

## H    Experiment Details: Global Forecasting with ERA5

In this appendix we give further detail about the graphs, data and experimental setup used in our global forecasting experiment.

### H.1    Graph Construction

**Multi-scale graph**    The global multi-scale graph is created in the same way as in GraphCast [23], by recursively splitting the faces of an icosahedron. As we work with data on a coarser resolution, we perform only 4 steps of such splitting, resulting in the graphs $\mathcal{G}_5, \ldots, \mathcal{G}_1$. These are all then merged to create $\mathcal{G}_{\text{MS}}$, used by GraphCast* and Graph-EFM (ms). By splitting 4 times we end up with a ratio of $\frac{N}{|\mathcal{V}_M|} \approx 11$ between grid nodes and mesh nodes. This can be compared with the 6 splitting steps of GraphCast, resulting in a ratio $\frac{N}{|\mathcal{V}_M|} \approx 25$ for their 0.25°data. We initially experimented also with splitting only 3 times (resulting in $\frac{N}{|\mathcal{V}_M|} \approx 45$), but models using these graphs showed inferior performance.

**Hierarchical graph**    As the original icosahedron $\mathcal{G}_5$ contains very few nodes, we do not use this for the hierarchical mesh graph, but rather construct the hierarchy using $\mathcal{G}_4, \ldots, \mathcal{G}_1$. Edges $\mathcal{E}_{l,l+1}$ between levels are constructed by connecting each mesh node at level $l$ with nodes at level $l + 1$ within a distance of $1.1$ times the edge length in $\mathcal{G}_l$. This guarantees that each node at level $l$ has 1 or 2 connection to the level above (see fig. 12g). Downward edges $\mathcal{E}_{l+1,l}$ are created by simply flipping the edge directions of $\mathcal{E}_{l,l+1}$. We visualize the global mesh graphs in fig. 12 and list the exact number of nodes and edges in table 4.

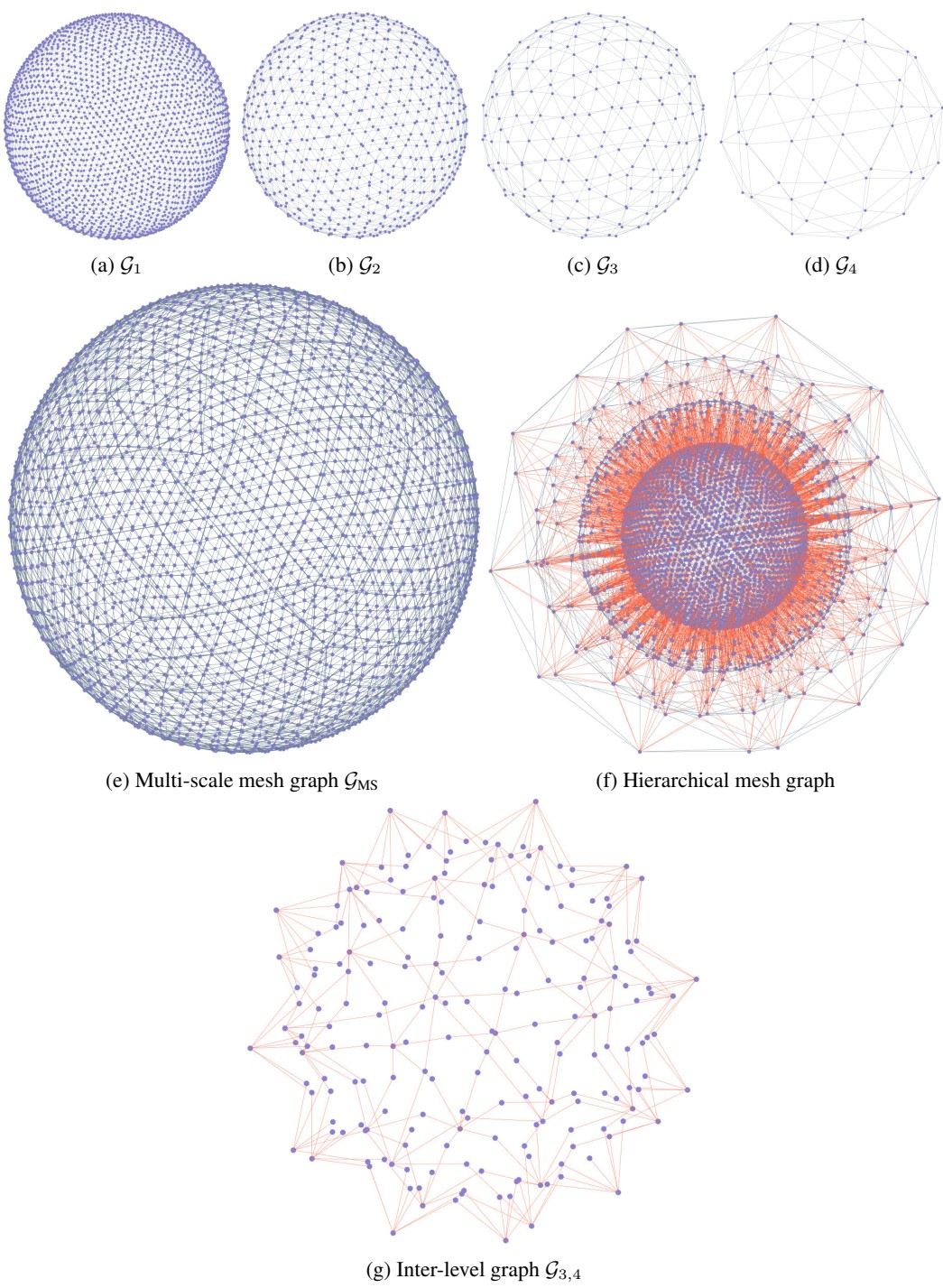

(a) $\mathcal{G}_1$      (b) $\mathcal{G}_2$      (c) $\mathcal{G}_3$      (d) $\mathcal{G}_4$

(e) Multi-scale mesh graph $\mathcal{G}_{\mathrm{MS}}$      (f) Hierarchical mesh graph

(g) Inter-level graph $\mathcal{G}_{3,4}$

Figure 12: Mesh graphs used in the global experiment. Note that the vertical positioning (away from earth's surface) is purely for visualization purposes.

Table 4: Global graph statistics.

| Graph | | Nodes | Edges |
|---|---|---|---|
| Hierarchical | $\mathcal{G}_1$ | 2562 | 15 360 |
| | $\mathcal{G}_2$ | 642 | 3840 |
| | $\mathcal{G}_3$ | 162 | 960 |
| | $\mathcal{G}_4$ | 42 | 240 |
| | $\mathcal{G}_{1,2}/\mathcal{G}_{2,1}$ | - | 4482 |
| | $\mathcal{G}_{2,3}/\mathcal{G}_{3,2}$ | - | 1122 |
| | $\mathcal{G}_{3,4}/\mathcal{G}_{4,3}$ | - | 282 |
| | Total | 3408 | 32 172 |
| $\mathcal{G}_{\mathrm{MS}}$ | | 2562 | 20 460 |
| $\mathcal{G}_{\mathrm{G2M}}$ | | - | 46 158 |
| $\mathcal{G}_{\mathrm{M2G}}$ | | - | 87 120 |
| Grid | | 29 040 | - |

**Connecting the grid and mesh**  Edges connecting the grid and mesh graph are also constructed following Lam et al. [23]. The edges $\mathcal{E}_{\mathrm{G2M}}$ are created by connecting each grid node to mesh nodes within a distance of 0.6 times the edge length in $\mathcal{G}_1$. Edges $\mathcal{E}_{\mathrm{M2G}}$ are constructed by for each grid node finding the triangle in $\mathcal{G}_1$ containing it, and adding three edges from the corner mesh nodes of that triangle. Note that the edge sets $\mathcal{E}_{\mathrm{G2M}}$ and $\mathcal{E}_{\mathrm{M2G}}$ are identical for the multi-scale and hierarchical mesh graphs, since $\mathcal{G}_{\mathrm{MS}}$ and $\mathcal{G}_1$ contain the same nodes. Since the grid nodes are laid out in a latitude-longitude grid and the mesh is icosahedral all mesh nodes will not be connected to the same number of grid nodes. Specifically, mesh nodes close to the poles will have many more grid connections than mesh nodes around the equator.

**Static features**  All mesh nodes are associated with static features related to their position, specifically $\cos(\text{Latitude})$, $\sin(\text{Longitude})$ and $\cos(\text{Longitude})$. Edges (in the mesh, $\mathcal{E}_{\mathrm{G2M}}$ and $\mathcal{E}_{\mathrm{M2G}}$) have static features containing the edge length and vector difference between its endpoints (see Lam et al. [23] for details). Edge features are normalized by the length of the longest edge.

## H.2 Dataset Details

We use a version of ERA5 [17] re-gridded to 1.5° latitude-longitude gridding, provided through the WeatherBench 2 benchmark [40]. Data from the period 1959-01-01T12 to 2017-12-31T12 is used for training, 2017-12-31T18 to 2019-12-31T12 for validation and 2019-12-31T18 to 2021-01-10T18 for testing. These exact time stamps guarantee that there is no overlap between the subsets. For evaluation we consider forecasts initialized at times 00 and 12 UTC each day of 2020, following WeatherBench 2. We perform 10 day forecasts using 6 h time steps. During training we also start forecasts at times 06 and 18 UTC.

The exact set of forecast variables, forcing and static fields is listed in table 5. We use the same inputs as Lam et al. [23]. The forcing is windowed over three consecutive time steps, meaning that each $F^t$ contains forcing from times $t-1$, $t$ and $t+1$ as well as the the static fields. For training we rescale the values of each variable to zero mean and unit variance.

## H.3 Model and Training Configurations

We train all models using the AdamW optimizer [57] and utilize BFloat16 mixed precision to save GPU memory. The training costs for the models makes extensive hyperparameter tuning unfeasible. We choose hyperparameters based on initial experimentation with smaller models. For Graph-EFM the important weightings $\lambda_{\mathrm{KL}}$ and $\lambda_{\mathrm{CRPS}}$ in $\mathcal{L}$ can be chosen based on monitoring the model behavior during training. This was done by plotting example forecasts throughout the training process and monitoring metrics on the validation set. The weight $\lambda_{\mathrm{KL}}$ was tuned to prevent the model from collapsing to deterministic predictions, and to still achieve useful predictions when $Z^t$ was sampled

Table 5: Variables, forcing and static fields from ERA5 used in our global forecasting experiments.
[†] kg of water vapour per kg of air.

|  | Abbreviation | Unit | Vertical Levels |
|---|---|---|---|
| **Variables** | | | |
| Geopotential | z | $m^2/s^2$ | 50 hPa, 100 hPa, 150 hPa, 200 hPa, |
| Specific humidity | q | kg/kg[†] | 250 hPa, 300 hPa, 400 hPa, |
| Temperature | t | K | 500 hPa, 600 hPa, 700 hPa, |
| $u$-component of wind | u | m/s | 850 hPa, 925 hPa, 1000 hPa |
| $v$-component of wind | v | m/s | |
| Vertical velocity | w | Pa/s | |
| Temperature | t | K | 2 m above surface |
| $u$-component of wind | u | m/s | 10 m above surface |
| $v$-component of wind | v | m/s | 10 m above surface |
| Mean sea level pressure | msl | Pa | Sea level |
| Total precipitation (Acc. over 6 h) | tp | m | Surface |
| **Forcing** | | | |
| Top of atm. solar radiation flux | toa | $W/m^2$ | Top of atmosphere |
| Sine-encoded time of day | sin_tod | - | - |
| Cosine-encoded time of day | cos_tod | - | - |
| Sine-encoded time of year | sin_toy | - | - |
| Cosine-encoded time of year | cos_toy | - | - |
| **Static Fields** | | | |
| Land-sea mask | water | $[0, 1]$ | Surface |
| Surface geopotential (topography) | topography | $m^2/s^2$ | Surface |
| $\cos(\text{Latitude})$ | cos_lat | - | - |
| $\sin(\text{Longitude})$ | sin_lon | - | - |
| $\cos(\text{Longitude})$ | cos_lon | - | - |

Table 6: Details of model architectures and training times for global forecasting.

| Model | $d_z$ | Processing steps | Parameters | Training time (GPU-hours) |
|---|---|---|---|---|
| GraphCast* | 256 | 8 | $5.2 \times 10^6$ | 716 |
| Graph-FM | 256 | 8 | $30.9 \times 10^6$ | 1040 |
| Graph-EFM (ms) | 256 | $\mu_Z$: 2, $g$: 4, $q$: 4 | $7.7 \times 10^6$ | 1372 |
| Graph-EFM | 256 | - | $16.3 \times 10^6$ | 1264 |

from the latent map. The CRPS weight $\lambda_{\text{CRPS}}$ was tuned to achieve a good ensemble spread while avoiding artifact issues, as discussed in appendix G.

The exact model configurations and training times for our global experiments are listed in table 6. Parameter counts for probabilistic models include parameters in the variational approximation, although this component does not play a role during forecasting. We report training times as hours of active computations on a single GPU. In practice we use 8 80GB NVIDIA A100 GPUs in parallel, meaning that the wall-clock time of our training is given by dividing the numbers in table 6 by 8. For the Graph-FM model one processing step on the mesh graph constitutes a full pass through the hierarchy (either from the grid up or from level $L$ down). In Graph-EFM (ms) multiple GNN processing steps operate on $\mathcal{G}_{\text{MS}}$. We list the number of such steps in table 6 separately for the latent map ($\mu_Z$), predictor ($g$) and variational approximation ($q$).

We train all models by a sequence of training step, starting from single-step prediction and then unrolling predictions $T$ steps. For the probabilistic models we first train in a pure auto-encoder setup with $\lambda_{\text{KL}} = 0$, encouraging $q$ to encode useful information in the distribution over $Z^t$. We find this useful for preventing the model from ignoring $Z^t$ and collapsing to purely deterministic forecasting.

Table 7: Training schedule for deterministic models (GraphCast* and Graph-FM) on global data.

| Epochs | Learning Rate | Unrolling $T$ |
|--------|---------------|---------------|
| 70 | $10^{-3}$ | 1 |
| 20 | $10^{-4}$ | 4 |
| 20 | $10^{-4}$ | 8 |

Table 8: Training schedule for Graph-EFM on global data. For Graph-EFM (ms) we use the same schedule but with different constants ($\lambda_{\mathrm{KL}} = 1$, $\lambda_{\mathrm{CRPS}} = 10^3$).

| Epochs | Learning Rate | Unrolling $T$ | $\lambda_{\mathrm{KL}}$ | $\lambda_{\mathrm{CRPS}}$ |
|--------|---------------|---------------|------|--------|
| 50 | $10^{-3}$ | 1 | 0 | 0 |
| 75 | $10^{-4}$ | 1 | 0.1 | 0 |
| 20 | $10^{-4}$ | 4 | 0.1 | 0 |
| 10 | $10^{-4}$ | 8 | 0.1 | 0 |
| 2 | $10^{-4}$ | 12 | 0.1 | $10^4$ |

The probabilistic models additionally include the fine-tuning using CRPS as a final training step. The full training schedule for the deterministic models is given in table 7 and for the probabilistic models in table 8. Note that we here define one epoch as initializing the model at each possible time in the training set (00, 06, 12 and 18 UTC in each day) such that all unrolled lead times are still within the training set period. The reason for probabilistic models being unrolled to a longer lead time $T$ is mainly to combat the artifacts discussed in appendix G.2.

# I    Experiment Details: Limited Area Modeling with MEPS Data

In this appendix we give further detail about the graphs, data and experimental setup used in our experiments with the MEPS data.

## I.1    Graph Construction

**Multi-scale graph**    In this limited-area setting we construct graphs as regular quadrilateral meshes covering the rectangular MEPS forecasting area. To create these we lay out mesh nodes in regular rows and columns over the area. Each node is then connected with bidirectional edges to its neighbors horizontally, vertically and diagonally (see fig. 13a). This results in all nodes (except those at the edge of the area) having 8 neighbors. The procedure is repeated at 4 different resolutions, tripling the distance between nodes at each resolution. This means that a node at resolution level $l$ is positioned at the center of a group of $3 \times 3$ nodes at resolution level $l - 1$, sharing its exact position with the center node of the group (illustrated in fig. 13b). To create the multi-scale mesh graph $\mathcal{G}_{\mathrm{MS}}$ we then merge the graphs at different resolutions, combining any nodes that sit at the same coordinates into one node. Note that this is possible due to how nodes align across the resolution levels.

**Hierarchical graph**    For the hierarchical model the graphs of different resolution are not merged, but used as the different levels of the hierarchy. We include only the 3 finest meshes $\mathcal{G}_1$, $\mathcal{G}_2$ and $\mathcal{G}_3$, as $\mathcal{G}_4$ contains only 9 nodes. Additional inter-level edge sets $\mathcal{E}_{l,l+1}$ and $\mathcal{E}_{l+1,l}$ are then created for all adjacent levels. Each set $\mathcal{E}_{l,l+1}$ of upwards edges is created by connecting each node on level $l$ with the closest node on level $l + 1$. This means that each node at levels $l > 1$ will have 9 incoming edges from the level below. The downward edges $\mathcal{E}_{l+1,l}$ are a copy of $\mathcal{E}_{l,l+1}$ with the direction of each edge flipped. The mesh graphs used for the MEPS experiment are visualized in fig. 14 and the number of nodes and edges in each graph listed in table 9.

**Connecting the grid and mesh**    To form $\mathcal{E}_{\mathrm{G2M}}$, each grid node is connected to mesh nodes closer than 0.67 times the distance between nodes in $\mathcal{G}_1$. All distances are here 2-dimensional euclidean, computed in the MEPS Lambert projection coordinates. The set $\mathcal{E}_{\mathrm{M2G}}$ is constructed by iterating over the grid nodes, and at each creating edges from the 4 closest mesh nodes to the node in the grid.

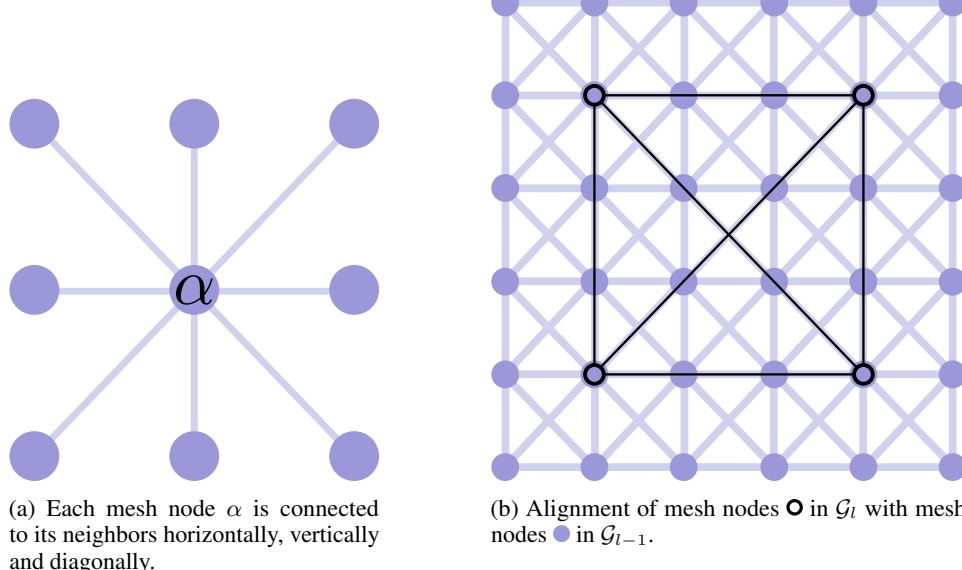

(a) Each mesh node $\alpha$ is connected to its neighbors horizontally, vertically and diagonally.

(b) Alignment of mesh nodes ○ in $\mathcal{G}_l$ with mesh nodes ● in $\mathcal{G}_{l-1}$.

Figure 13: Illustration of mesh graph construction in the LAM setting.

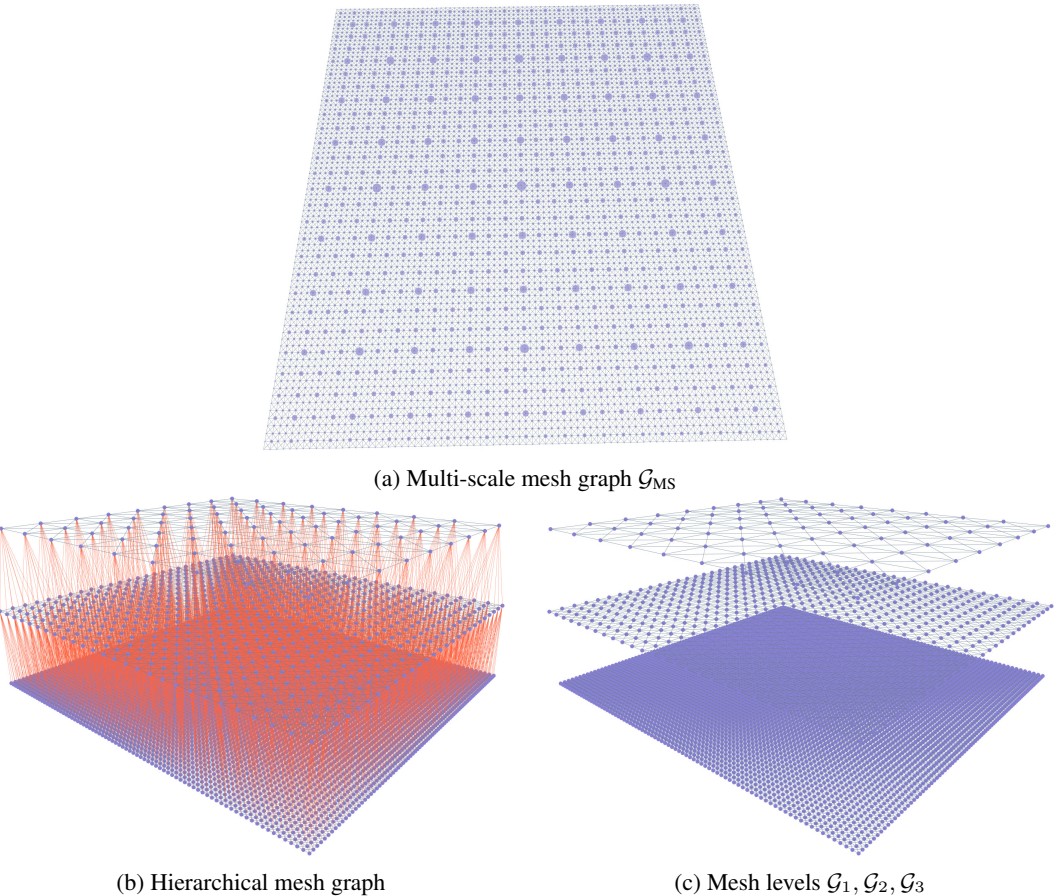

(a) Multi-scale mesh graph $\mathcal{G}_{\text{MS}}$

(b) Hierarchical mesh graph

(c) Mesh levels $\mathcal{G}_1, \mathcal{G}_2, \mathcal{G}_3$

Figure 14: Mesh graphs used in the MEPS experiment. Note that the vertical positioning and size of nodes is purely for visualization purposes.

Table 9: MEPS graph statistics.

| Graph | | Nodes | Edges |
|---|---|---|---|
| | $\mathcal{G}_1$ | 6561 | 51 520 |
| | $\mathcal{G}_2$ | 729 | 5512 |
| | $\mathcal{G}_3$ | 81 | 544 |
| Hierarchical | $\mathcal{G}_{1,2}/\mathcal{G}_{2,1}$ | - | 6561 |
| | $\mathcal{G}_{2,3}/\mathcal{G}_{3,2}$ | - | 729 |
| | Total | 7371 | 72 156 |
| $\mathcal{G}_{MS}$ | | 6561 | 57 616 |
| $\mathcal{G}_{G2M}$ | | - | 100 656 |
| $\mathcal{G}_{M2G}$ | | - | 255 136 |
| Grid | | 63 784 | - |

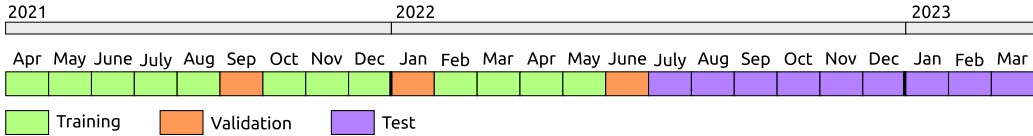

Figure 15: Overview of the period covered in the dataset and the training/validation/test split.

**Static features**   The static features associated with mesh nodes are their 2-dimensional coordinates in the MEPS Lambert projection, normalized by the maximum coordinate value. All edges have static features including the length of the edge and the vector difference between the source and target nodes (using the projection coordinates). The edge features are normalized by the length of the longest edge in the whole mesh graph.

## I.2   Dataset Details

**Dataset**   The MEPS dataset consists of archived forecasts from the operational MEPS system during the period April 2021 – March 2023. This period was chosen due to the system configuration being reasonably stable, preventing distributional shifts within the dataset. From the chosen period we extract the forecasts started at 00 and 12 UTC each day. At each initialization time there are 5 ensemble forecasts, started from slightly different initial conditions. This results in 10 forecasts per day (assuming no ensemble members are missing due to operational issues). When retrieving the data we additionally downsample the spatial resolution from the original 2.5 km to 10 km. This results in a dataset of 6069 forecasts of length 66 h with 1 h time steps. We split the forecasts into training, validation and test sets according to fig. 15. The specific validation months were chosen to reasonably cover the seasonal variations. Note that the dataset contains around 46 years of individual time steps. However, since there are obvious correlations between ensemble members and successively started forecasts the actual information content is far lower than in a 46 year reanalysis dataset.

**Variables and forcing**   At each grid cell we model 17 weather variables, including a broad range of different quantities and different vertical levels in the atmosphere. All variables, forcing and static fields are described in table 10. The particular choice of variables was motivated by a combination of meteorological relevance, data availability and striving for a diverse set of variables to evaluate the model on. We use the same type of windowing for the forcing and standardization of variables as in the global experiment. For solar radiation (`nlwrs` and `nswrs`) we consider the net flux at ground level, aggregated over the past 3 hours (since the last time step). Apart from the solar radiation all other variables are instantaneous. For the MEPS data we use the fraction of open water in the grid cell as a forcing input. We assume this to be constant over the forecast period and take the value from the time of the initial state. In the global experiment the land-sea mask is static, but treating this as forcing could be useful for taking into account seasonal fluctuations of the ice cover in the Nordic region. The boundary forcing $B^t$ consists of the same variables as listed in table 10. We include a static binary indicator variable describing if a node is in the boundary or forecast area.

Table 10: Variables, forcing and static fields in the MEPS dataset. [†]In the MEPS system 65 vertical model levels are defined from the ground to the top of the atmosphere [32]. The lowest MEPS level (Lvl65) sits at approximately 12.5 m above ground.

| | Abbreviation | Unit | Vertical Levels |
|---|---|---|---|
| **Variables** | | | |
| Atmospheric pressure | `pres` | Pa | Ground level (0g), Sea level (0s) |
| Net longwave solar radiation flux | `nlwrs` | W/m$^2$ | Surface |
| Net shortwave solar radiation flux | `nswrs` | W/m$^2$ | Surface |
| Relative humidity | `r` | $[0, 1]$ | 2 m, Lvl65[†] (65) |
| Temperature | `t` | K | 2 m, Lvl65[†] (65), 500 hPa, 850 hPa |
| $u$-component of wind | `u` | m/s | Lvl65[†] (65), 850 hPa |
| $v$-component of wind | `v` | m/s | Lvl65[†] (65), 850 hPa |
| Water vapor, integrated column | `wvint` | kg/m$^2$ | Full column |
| Geopotential | `z` | m$^2$/s$^2$ | 500 hPa, 1000 hPa |
| **Forcing** | | | |
| Top of atm. solar radiation flux | `toa` | W/m$^2$ | Top of atmosphere |
| Fraction of open water | `water` | $[0, 1]$ | Surface |
| Sine-encoded time of day | `sin_tod` | - | - |
| Cosine-encoded time of day | `cos_tod` | - | - |
| Sine-encoded time of year | `sin_toy` | - | - |
| Cosine-encoded time of year | `cos_toy` | - | - |
| **Static Fields** | | | |
| Surface geopotential (topography) | `topography` | m$^2$/s$^2$ | Surface |
| $x$-coordinate in MEPS projection | `x_coord` | - | - |
| $y$-coordinate in MEPS projection | `y_coord` | - | - |
| On-boundary binary indicator | `boundary` | 0/1 | - |

**Forecast steps and length**    The original data uses 1 h time steps, but our MLWP models predict in 3 h steps. Because of this we can extract 3 training samples from each forecast in the dataset (i.e. original time steps $\{1, 4, 7, \ldots\}$, $\{2, 5, 8, \ldots\}$ and $\{3, 6, 9, \ldots\}$). As we train on only $T \leq 8$ rollout steps, we also only need a subset of each such time series for each training iteration. To make maximum use of the data during training we randomly sample which time step to use as initial state for unrolling the model. The combination of 1) the 3 h time steps, 2) using the last two weather states as model inputs, 3) including forcing from multiple times as input, means that we reduce the effective length of our ground truth forecasts in pre-processing. This explains why we predict for 57 h rather than the original 66 h. Note however that nothing prevents us from unrolling the models to create forecasts for 66 h or beyond. Although this is possible, we do not have ground truth data to compare against past 57 h and therefore view such experiments to be of limited interest. There is still no reason to expect the model performance to become drastically worse specifically past 57 h, as all models are anyhow fine-tuned on shorter rollouts than this.

### I.3  Boundary Forcing

In all models we use boundary forcing $B^t$ in order to include information about the surrounding area. At each time step boundary forcing $B^{t-2:t-1}$ is passed to the model together with $X^{t-2:t-1}$, $F^t$, as described in fig. 16. There is a separate set of grid nodes used for the boundary input. These nodes are treated identically as grid nodes within the forecasting area by the GNN layers. As the boundary forcing is only a change to the model inputs this does not require any substantial re-design and we can adapt all models in our experiments to this LAM setting. Note that we use a boundary area that lays *inside* the original MEPS area, allowing us to use parts of the ground truth forecasts as boundary forcing. We specifically define the boundary as the 10 grid cells closest to the edge of the limited area. In the operational system the boundary area is defined along the edge *outside* the MEPS area. There is no major conceptual difference between these and one could easily re-define the different areas to match the operational MEPS system.

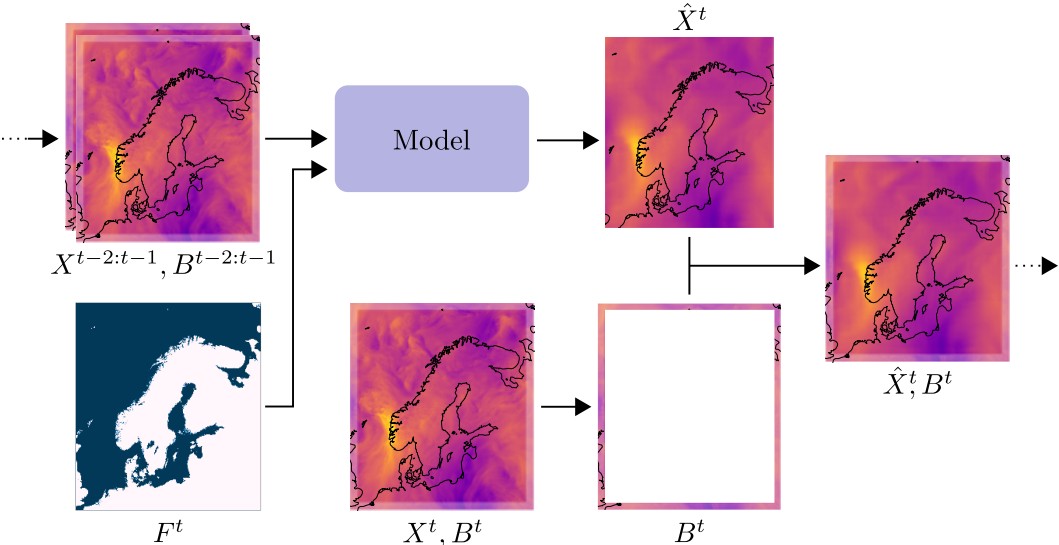

Figure 16: Schematic showing how we feed boundary forcing as input to the model in each autoregressive step.

Table 11: Details of model architectures and training times for LAM forecasting.

| Model | $d_z$ | Processing steps | Parameters | Training time (GPU-hours) |
|---|---|---|---|---|
| GraphCast* | 128 | 6 | $1.1 \times 10^6$ | 336 |
| Graph-FM | 128 | 6 | $4.5 \times 10^6$ | 432 |
| Graph-EFM (ms) | 128 | $\mu_Z$: 2, $g$: 4, $q$: 2 | $1.7 \times 10^6$ | 456 |
| Graph-EFM | 128 | - | $3.1 \times 10^6$ | 556 |

### I.4 Model and Training Configurations

For the MEPS experiment we again use the AdamW optimizer [57], but not mixed precision computations. Hyperparameter tuning follows a similar strategy as for the global experiment. The exact model configurations and training times for our MEPS experiments are listed in table 11. The training schedule for the deterministic models is given in table 12 and for the probabilistic models in table 13. These tables follow the same formats as tables 6 to 8, and we refer to the global experiment details in appendix H.3 for further explanations. In the MEPS case one epoch means training on one sub-sample (see description of sub-sampling in appendix I.2) of each forecast in the training set once.

## J Additional Results: Global Forecasting with ERA5

In this appendix we present additional results for global forecasting.

### J.1 Metrics

Comparisons between existing MLWP models are challenging, due to the many factors that impact forecast quality. While most global models proposed in the literature are trained on ERA5, the

Table 12: Training schedule for deterministic models (GraphCast* and Graph-FM) on MEPS data.

| Epochs | Learning Rate | Unrolling $T$ |
|---|---|---|
| 600 | $10^{-3}$ | 1 |
| 300 | $10^{-4}$ | 4 |
| 200 | $10^{-4}$ | 8 |

Table 13: Training schedule for Graph-EFM on MEPS data. For Graph-EFM (ms) we use the same schedule but with different constants ($\lambda_{\mathrm{KL}} = 100$, $\lambda_{\mathrm{CRPS}} = 10^4$, fine-tuning learning rate $10^{-4}$).

| Epochs | Learning Rate | Unrolling $T$ | $\lambda_{\mathrm{KL}}$ | $\lambda_{\mathrm{CRPS}}$ |
|---|---|---|---|---|
| 300 | $10^{-3}$ | 1 | 0 | 0 |
| 100 | $10^{-3}$ | 1 | 1 | 0 |
| 200 | $5 \times 10^{-4}$ | 4 | 1 | 0 |
| 50 | $5 \times 10^{-4}$ | 4 | 1 | $10^6$ |
| 100 | $5 \times 10^{-4}$ | 6 | 1 | $10^6$ |

exact choice of train/test split, spatial resolution and forecasted variables can vary (compare for example Keisler [20], Pathak et al. [37] and Hu et al. [19]). Models that operate on a higher spatial resolution and with more variables get more information in initial states, and should therefore be expected to produce better forecasts. When it comes to MLWP ensemble forecasting, an even greater challenge is that different initial conditions are used [39, 6, 16]. Given this situation, it is hard to disentangle the proposed machine learning methods from their surrounding design choices and make fair comparisons of model architectures.

We approach these complications by re-training a smaller set of models on the same data and experimental setup. Our focus is on graph-based, non-hybrid MLWP models, as this is the regime of Graph-FM and Graph-EFM. The baseline models are described in detail in appendix C.5. In this appendix we also include additional metrics taken directly from the WeatherBench 2 Benchmark [40]. These are for the original **GraphCast** model [23], **KeislerNet** [20] and the **IFS-ENS** operational ensemble from the European Centre for Medium-Range Weather Forecasts (ECMWF) [12]. The evaluation setup for these results match ours (evaluation against 2020 from ERA5, identical metrics computed on 1.5° data). However, as discussed above there are differences in the exact training data, variables and operating resolutions compared to our models.

Metric values from our global experiments are presented in figs. 17 to 19. We here showcase values for surface variables and atmospheric variables at pressure levels 500, 700 and 850 hPa. The results taken directly from WeatherBench 2 are shown in gray, to emphasize that any comparison with these comes with caveats. Note that results for all variables are not available for all models.

## J.2  Example Forecasts

In fig. 20 we showcase example forecasts from different models at lead time 10 days. Note that forecasting this far into the future is challenging, and sampled ensemble members show only one possible scenario predicted by the models.

Figure 21 shows example ensemble forecasts from Graph-EFM for surface variables, atmospheric variables at 700 hPa, z500 and t850. All forecasts are for 10 days lead time. For practical reasons we use only 20 members to estimate the ensemble mean and standard deviation in these plots.

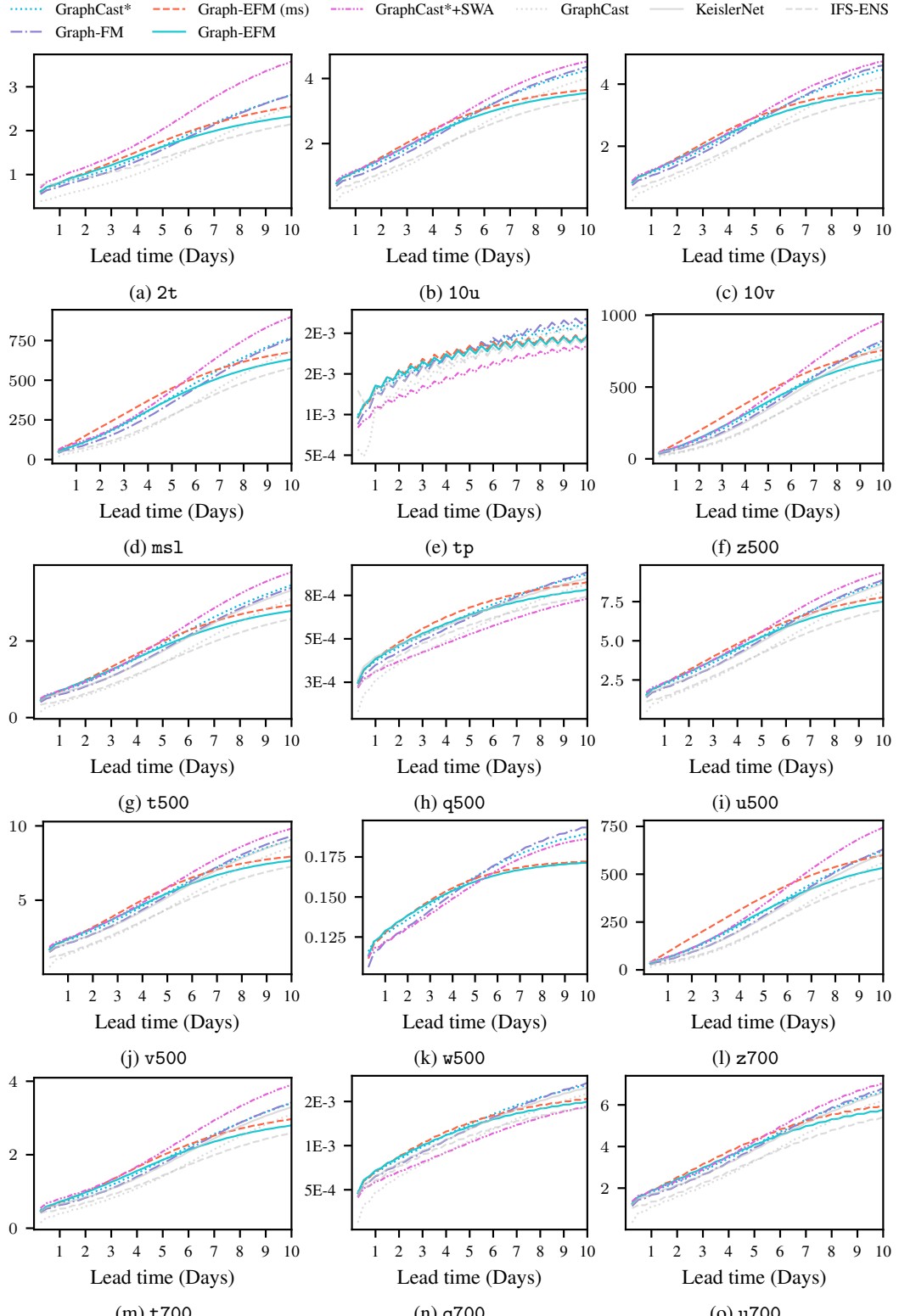

(a) `2t`  (b) `10u`  (c) `10v`

(d) `msl`  (e) `tp`  (f) `z500`

(g) `t500`  (h) `q500`  (i) `u500`

(j) `v500`  (k) `w500`  (l) `z700`

(m) `t700`  (n) `q700`  (o) `u700`

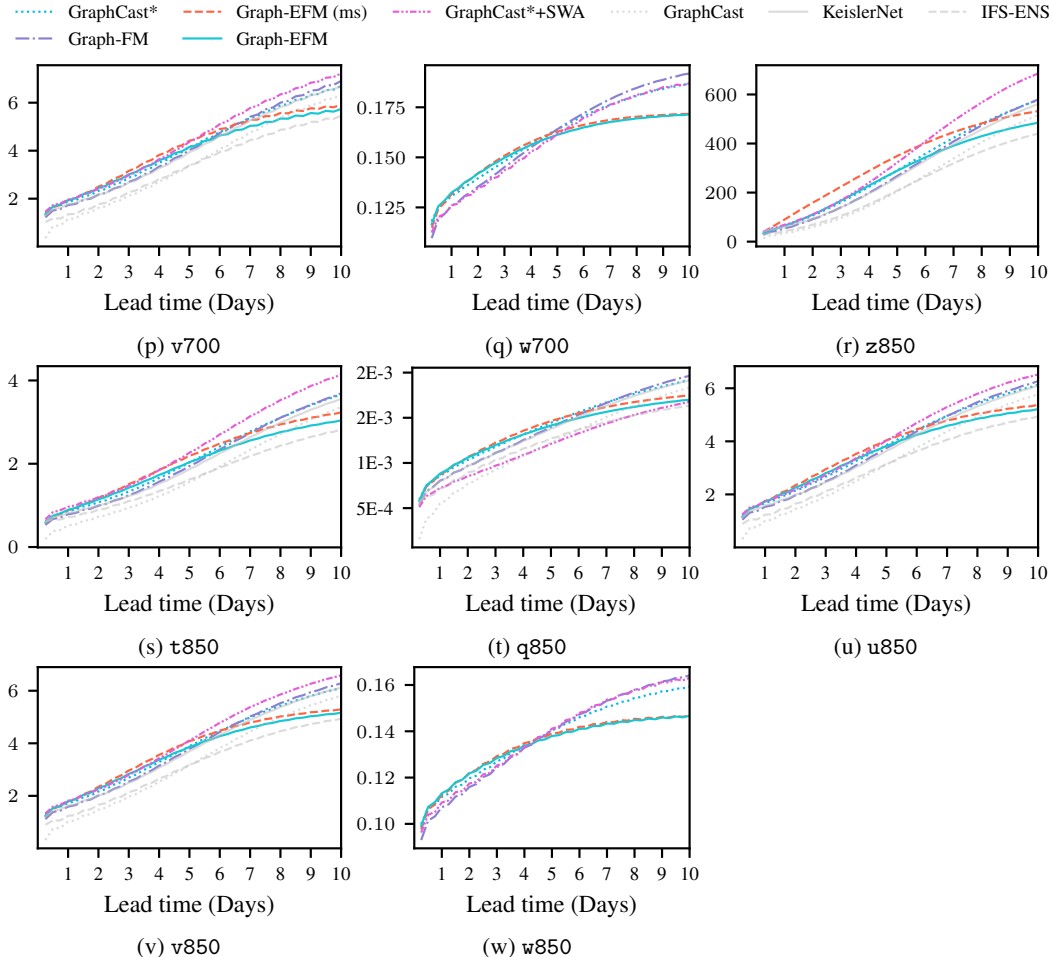

Figure 17: RMSE results for global experiment.

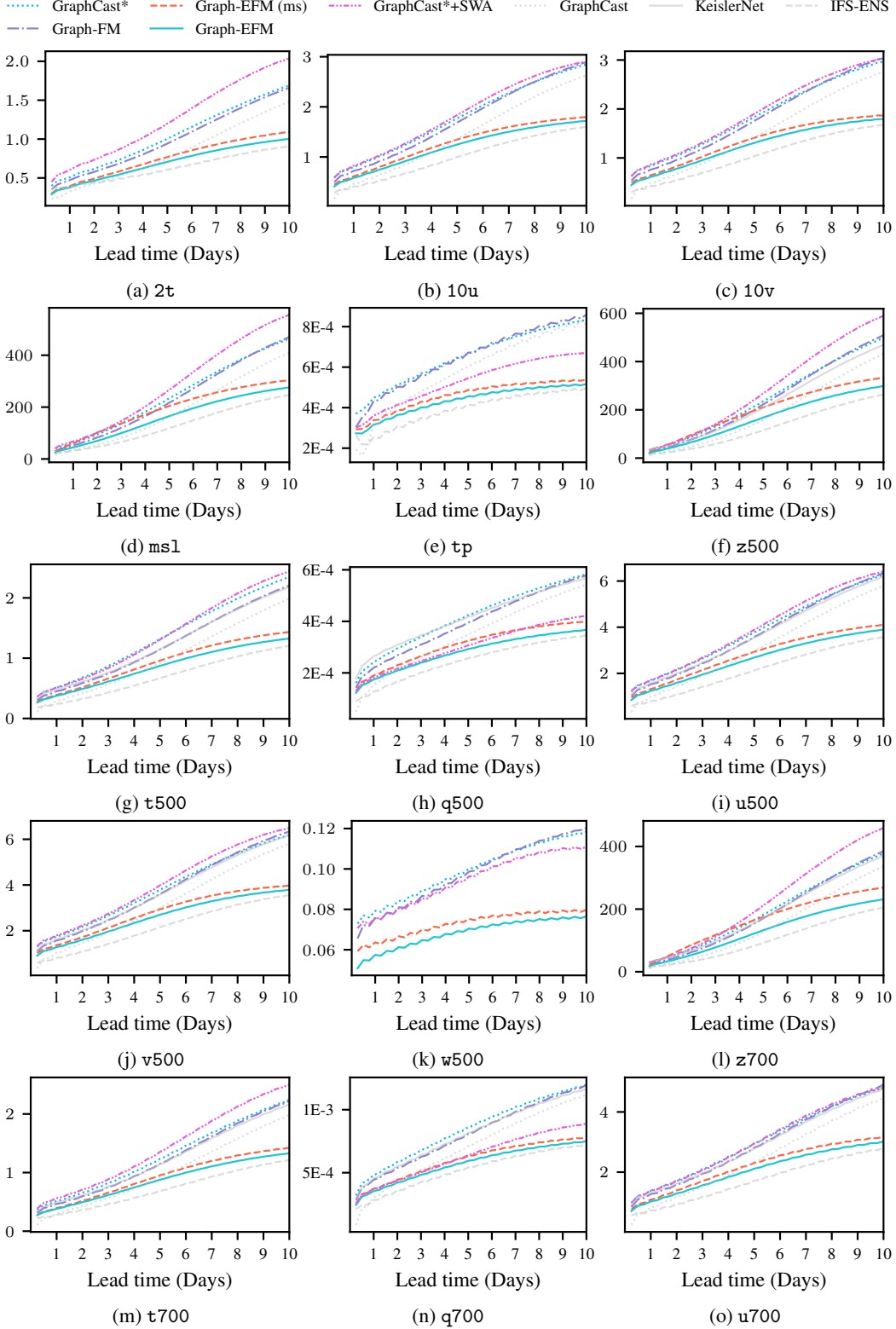

(a) `2t`

(b) `10u`

(c) `10v`

(d) `msl`

(e) `tp`

(f) `z500`

(g) `t500`

(h) `q500`

(i) `u500`

(j) `v500`

(k) `w500`

(l) `z700`

(m) `t700`

(n) `q700`

(o) `u700`

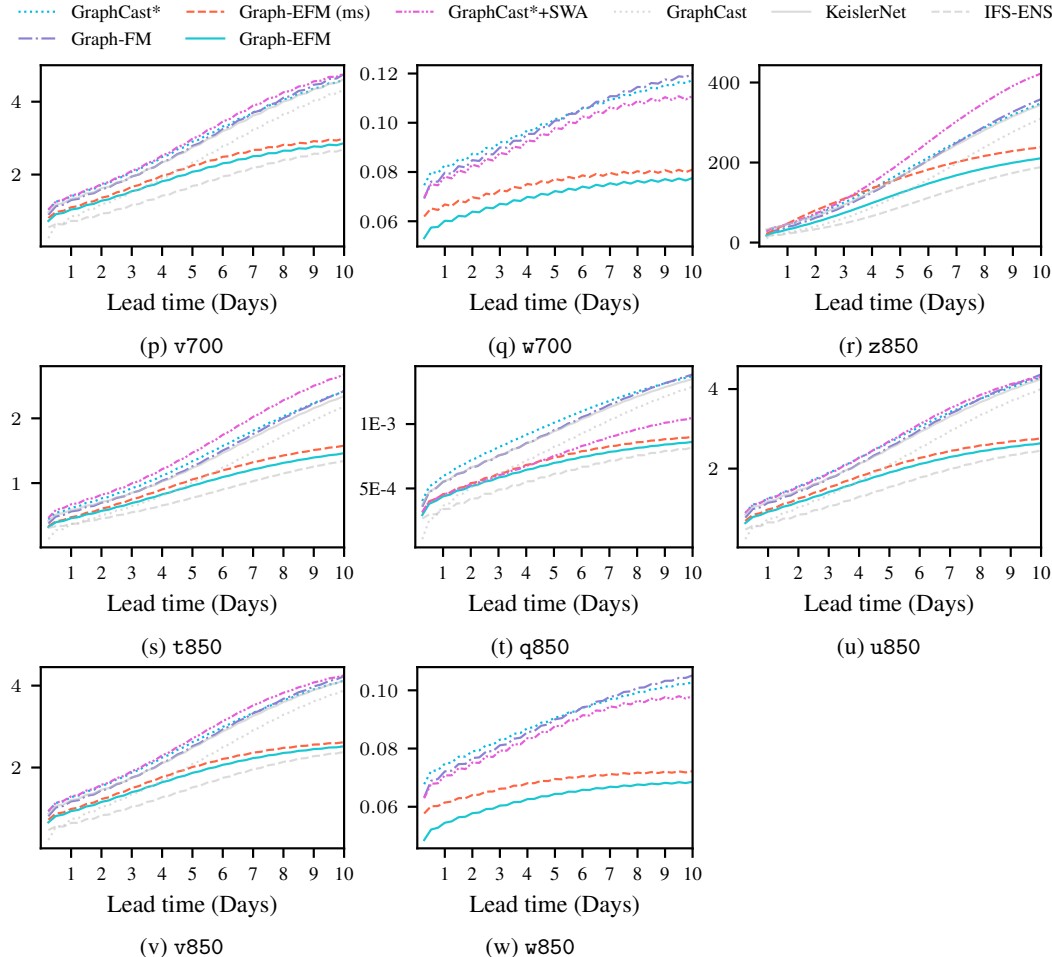

Figure 18: CRPS results for global experiment.

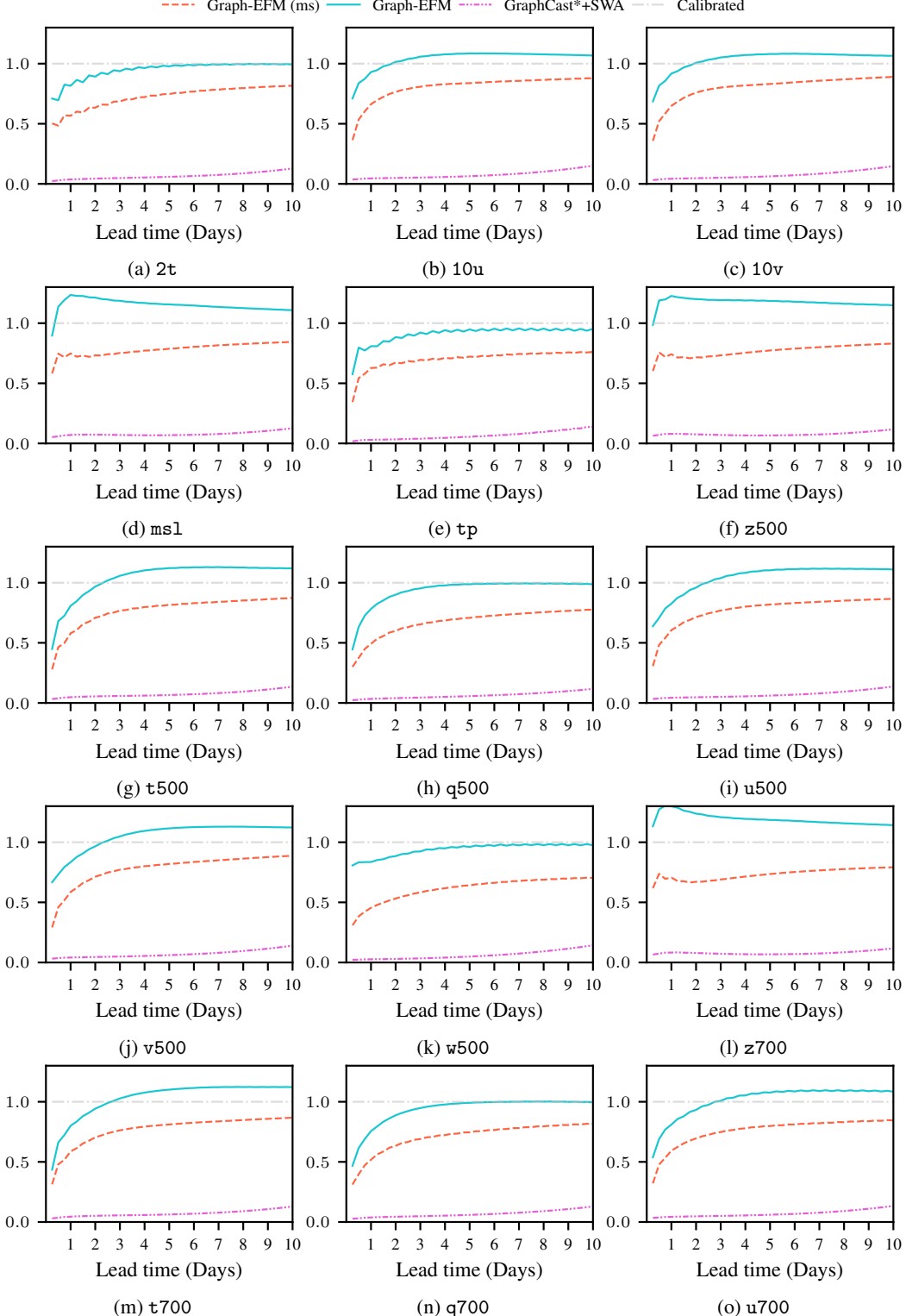

(a) `2t`  (b) `10u`  (c) `10v`

(d) `msl`  (e) `tp`  (f) `z500`

(g) `t500`  (h) `q500`  (i) `u500`

(j) `v500`  (k) `w500`  (l) `z700`

(m) `t700`  (n) `q700`  (o) `u700`

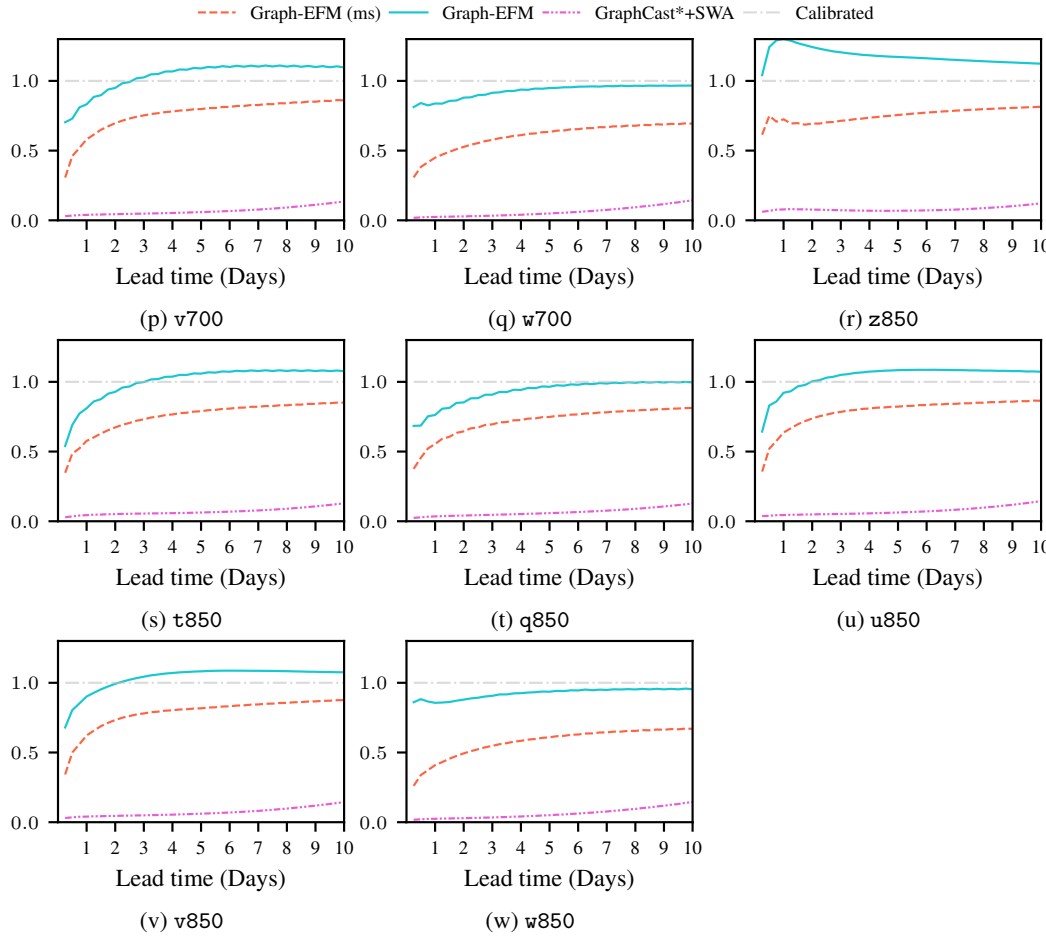

Figure 19: SpSkR results for global experiment.

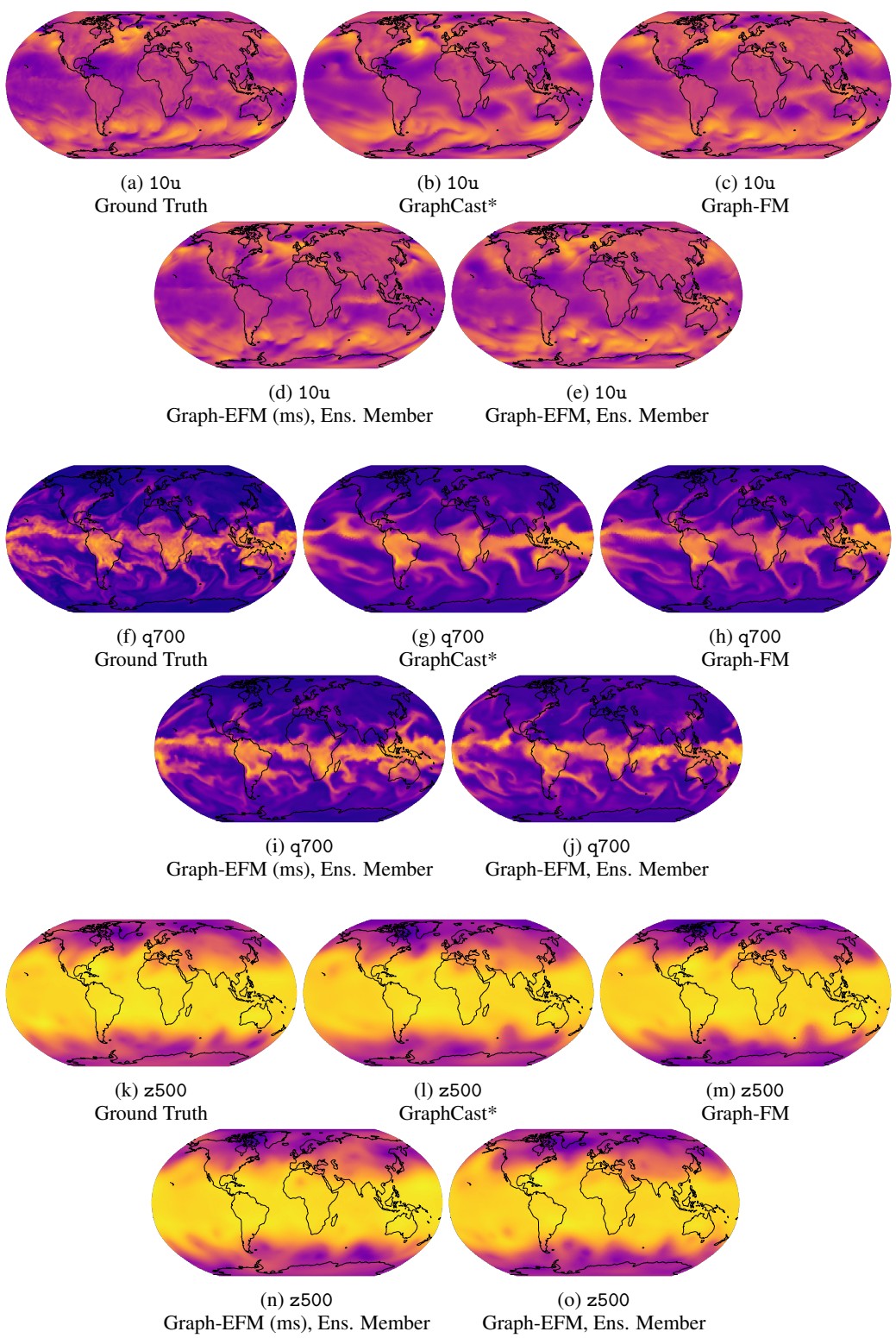

Figure 20: Comparison of global model forecasts at 10 days lead time for u-component of 10 m wind (10u), specific humidity at 700 hPa (q700) and geopotential at 500 hPa (z500). For probabilistic models we show sampled ensemble members.

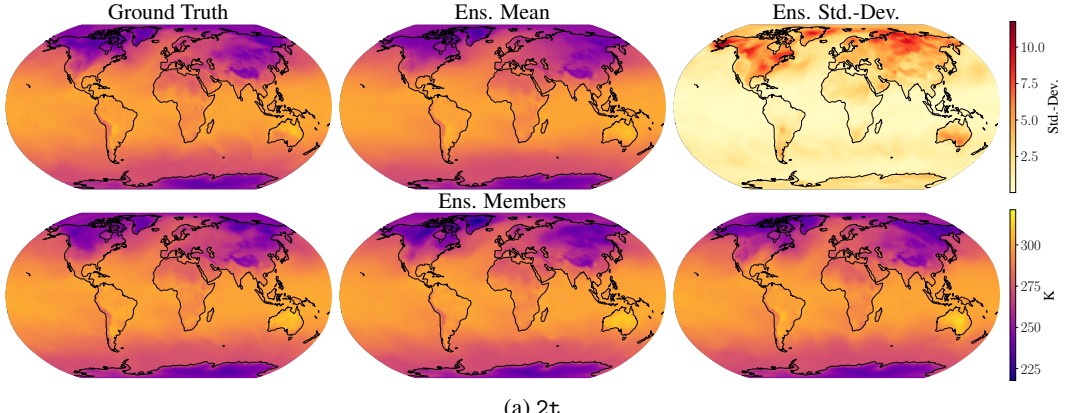

(a) 2t

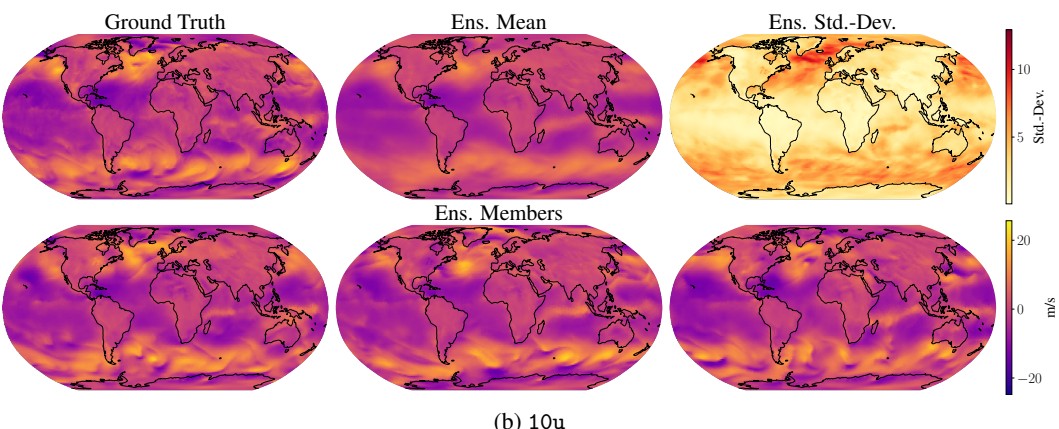

(b) 10u

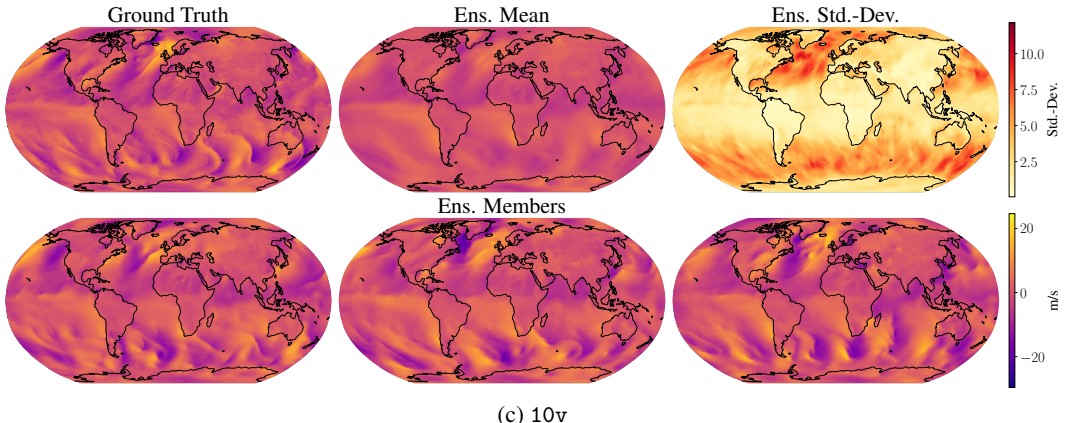

(c) 10v

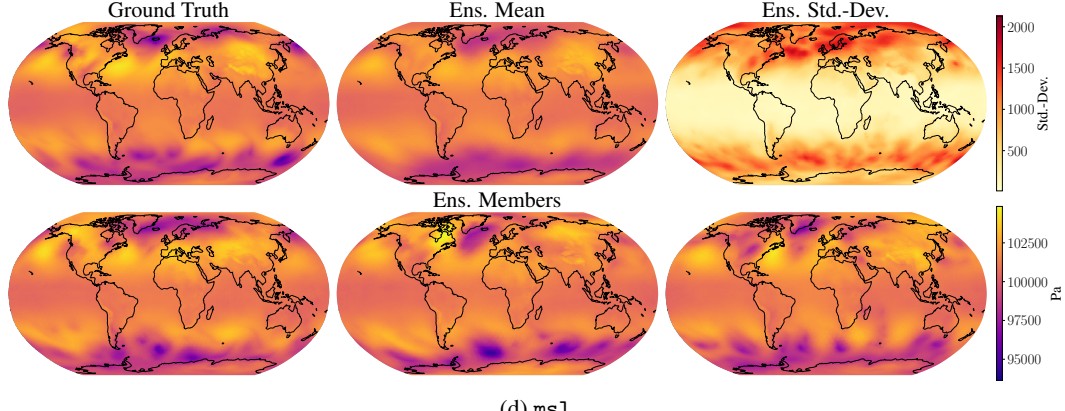

(d) `msl`

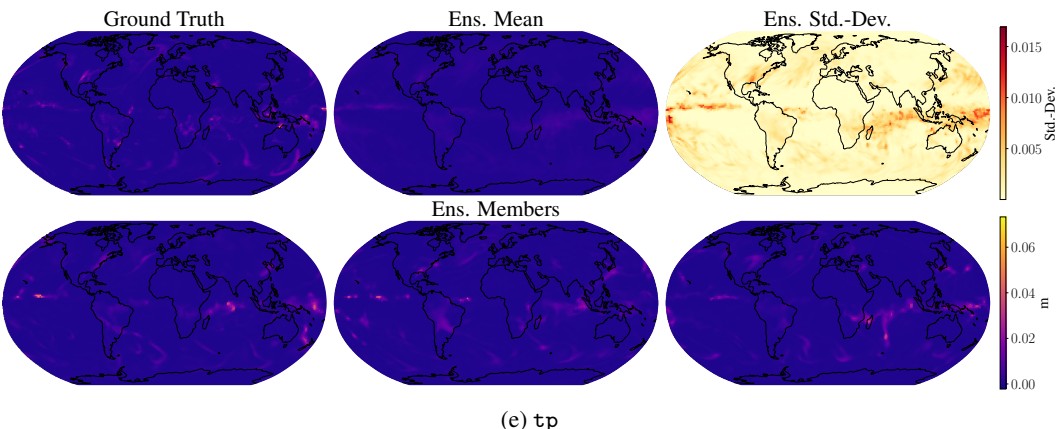

(e) `tp`

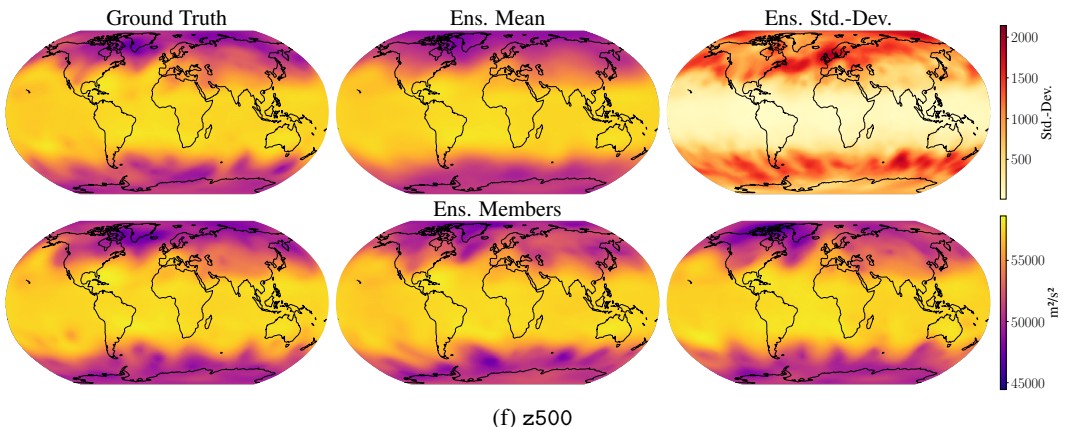

(f) `z500`

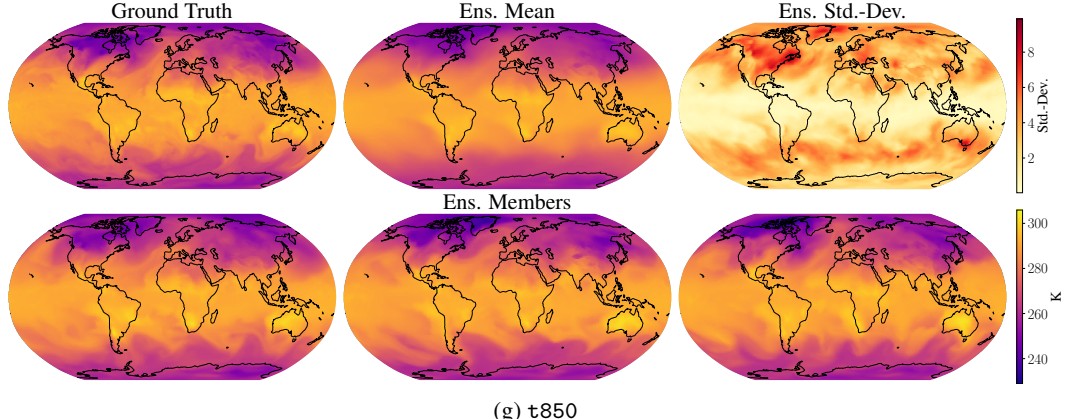

(g) t850

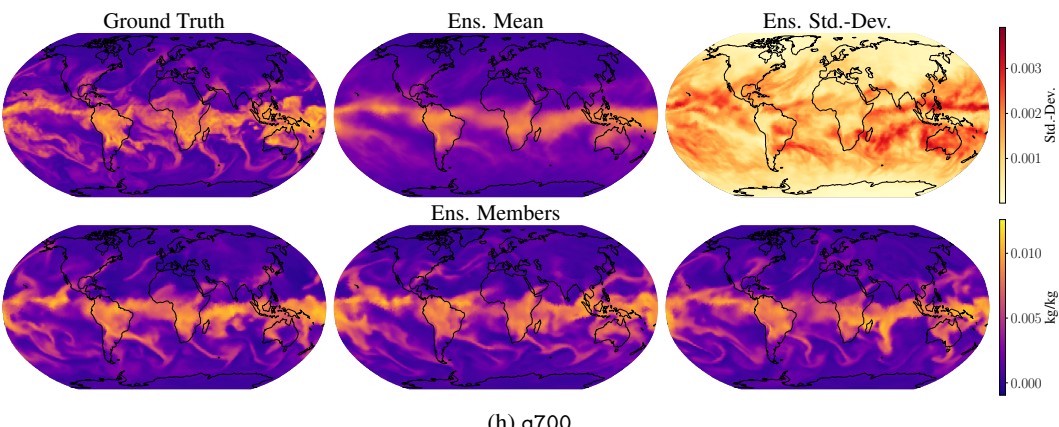

(h) q700

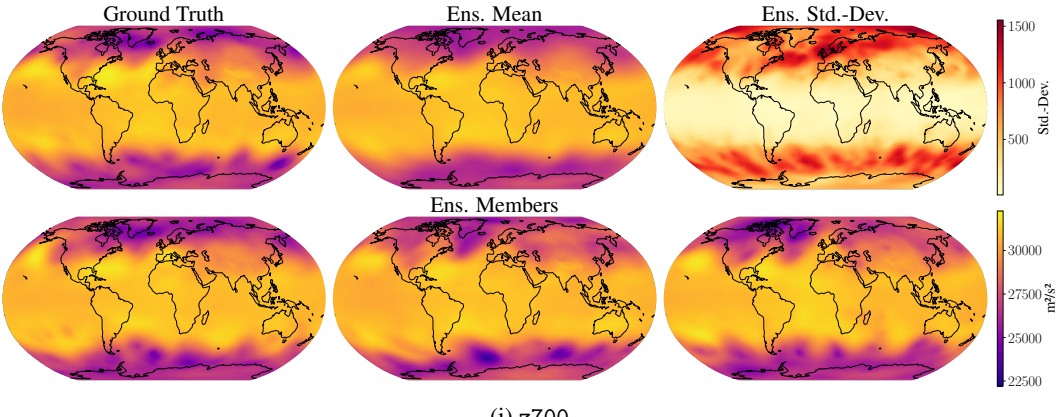

(i) z700

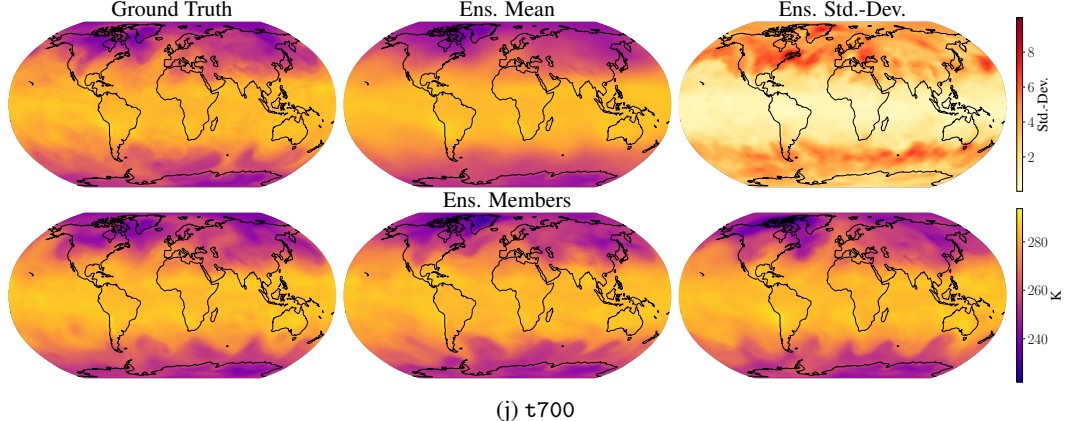

(j) t700

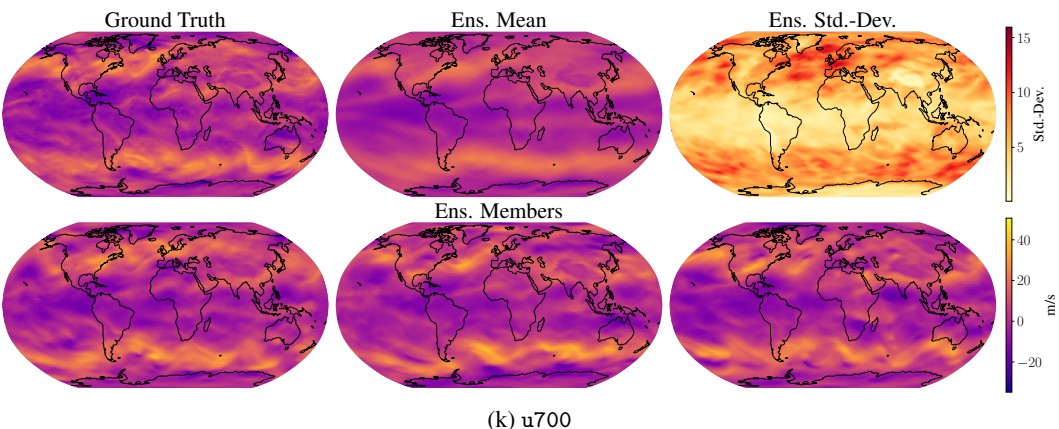

(k) u700

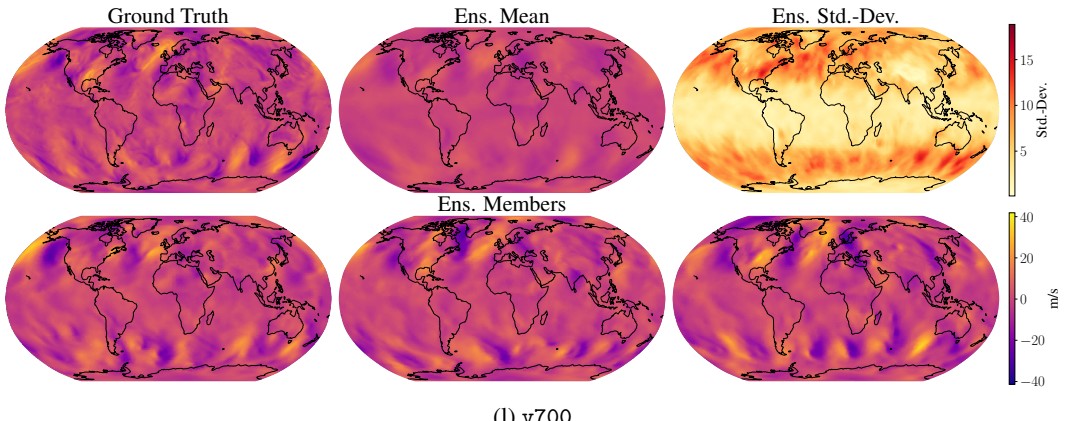

(l) v700

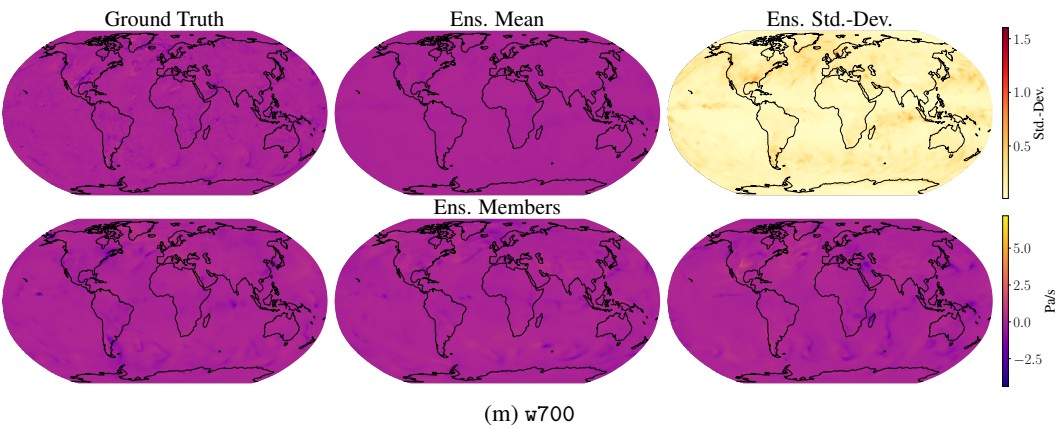

(m) `w700`

Figure 21: Example Graph-EFM global ensemble forecasts at lead time 10 days.

# K    Additional Results: Limited Area Modeling with MEPS Data

In this appendix we show additional results from our experiment with the MEPS data.

## K.1    Metrics

Figures 22 to 24 show metric values for all variables and lead times in the MEPS dataset.

The poor performance of the Graph-EFM (ms) model is noteworthy, both in terms of forecast accuracy and ensemble calibration. This can to a large extent be attributed to the exact training objective (see table 13), particularly a lower weighting $\lambda_{\text{CRPS}}$. Using a lower value for $\lambda_{\text{CRPS}}$ in the multi-scale model was necessary to avoid the artifacts discussed in appendix G. The does however mean that Graph-EFM (ms) does not gain as much of the benefits that come with the CRPS fine-tuning.

## K.2    Example Forecasts

A comparison between example forecasts from different models is given in fig. 25. Note that ensemble members sampled from Graph-EFM show more detailed features than deterministic forecasts.

In fig. 26 we plot example ensemble forecasts from the Graph-EFM model for all variables in the MEPS data. Note that these plots include the boundary area, which is not forecast, as a faded border in each plot. All forecasts are for lead time 57 h. Rolling out a LAM ensemble forecast from Graph-EFM to 57 h is even faster than generating the global ensemble. Using batched sampling on a single GPU, we can produce 100 ensemble members in 140 s (1.4 s per member).

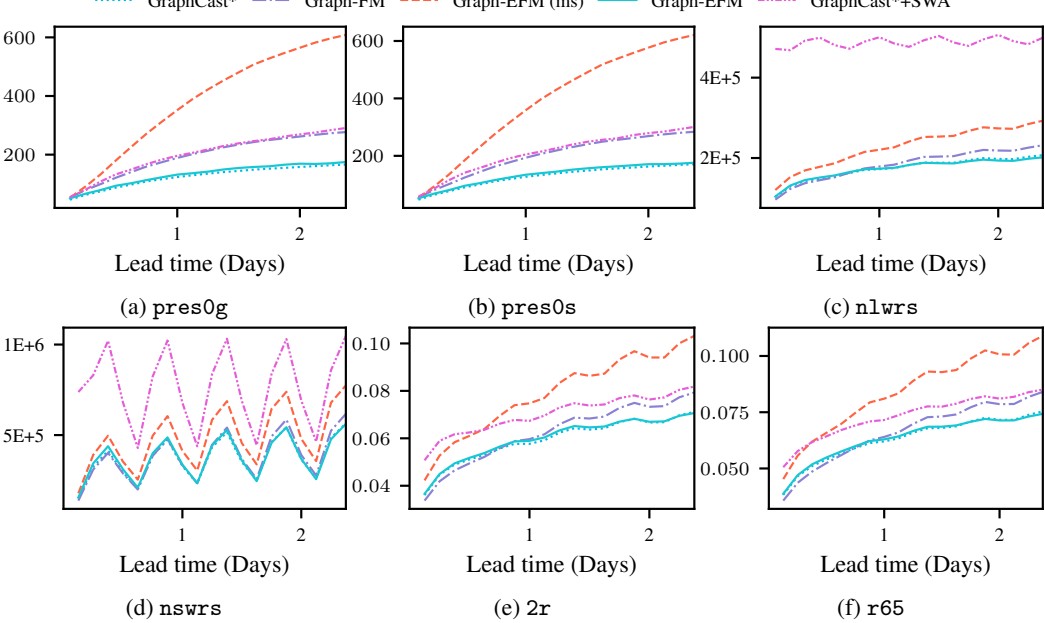

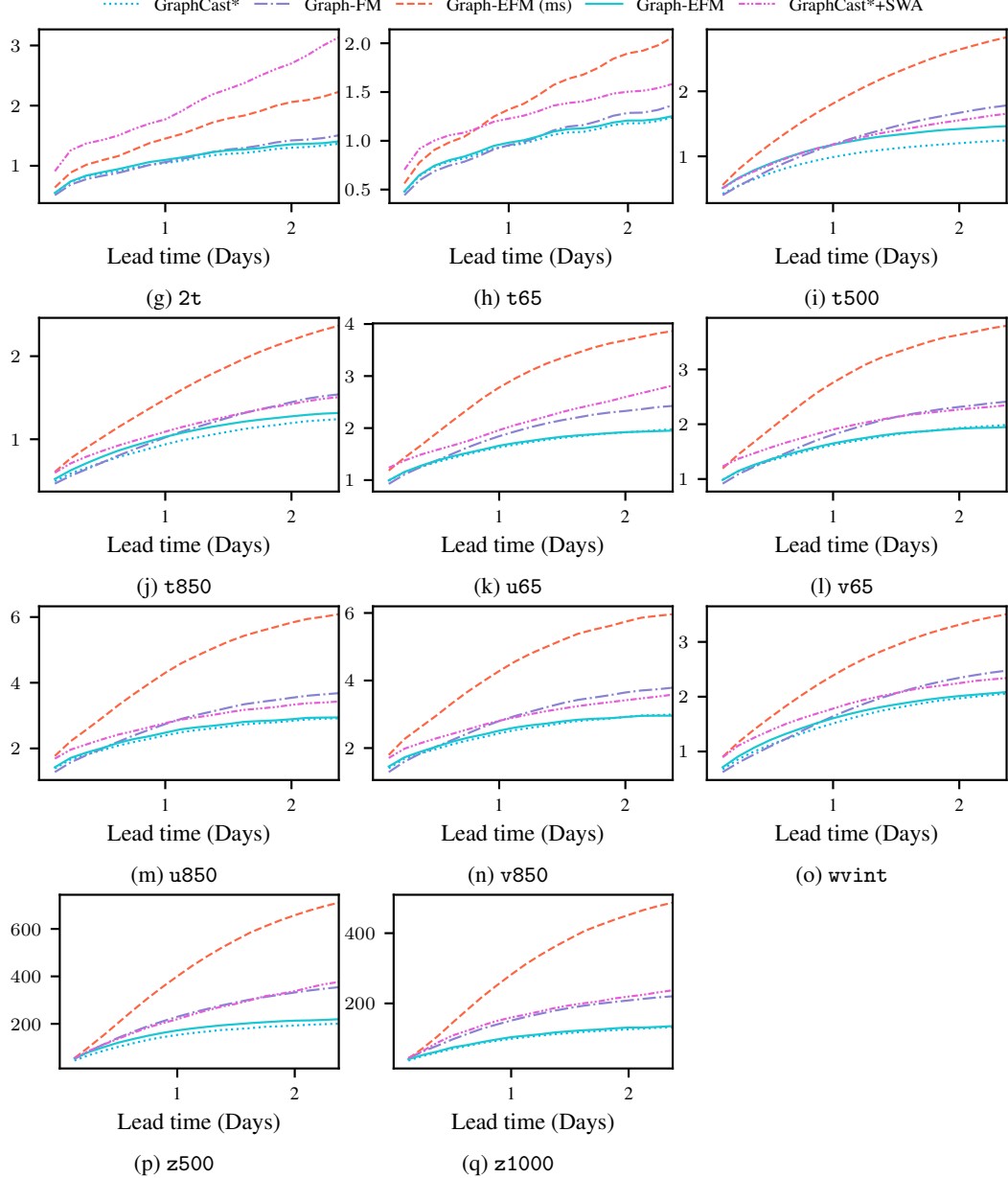

Figure 22: RMSE for MEPS experiment.

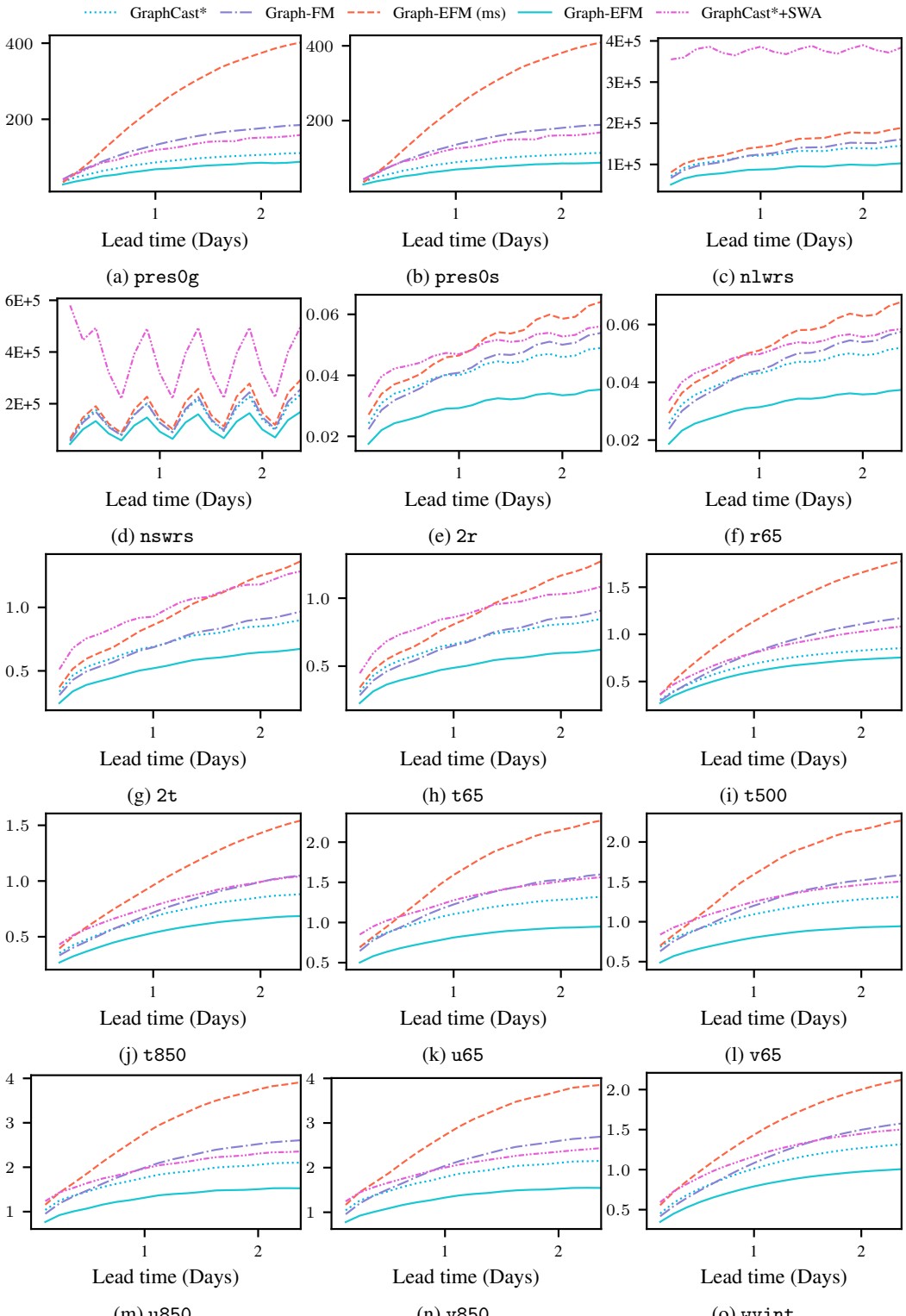

(a) pres0g  (b) pres0s  (c) nlwrs

(d) nswrs  (e) 2r  (f) r65

(g) 2t  (h) t65  (i) t500

(j) t850  (k) u65  (l) v65

(m) u850  (n) v850  (o) wvint

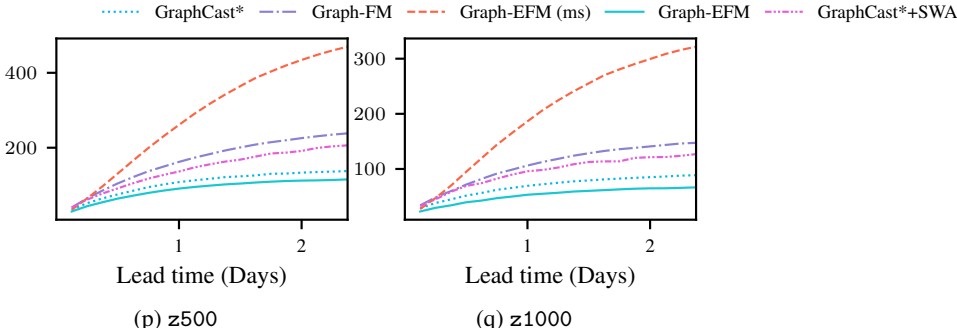

Figure 23: CRPS for MEPS experiment.

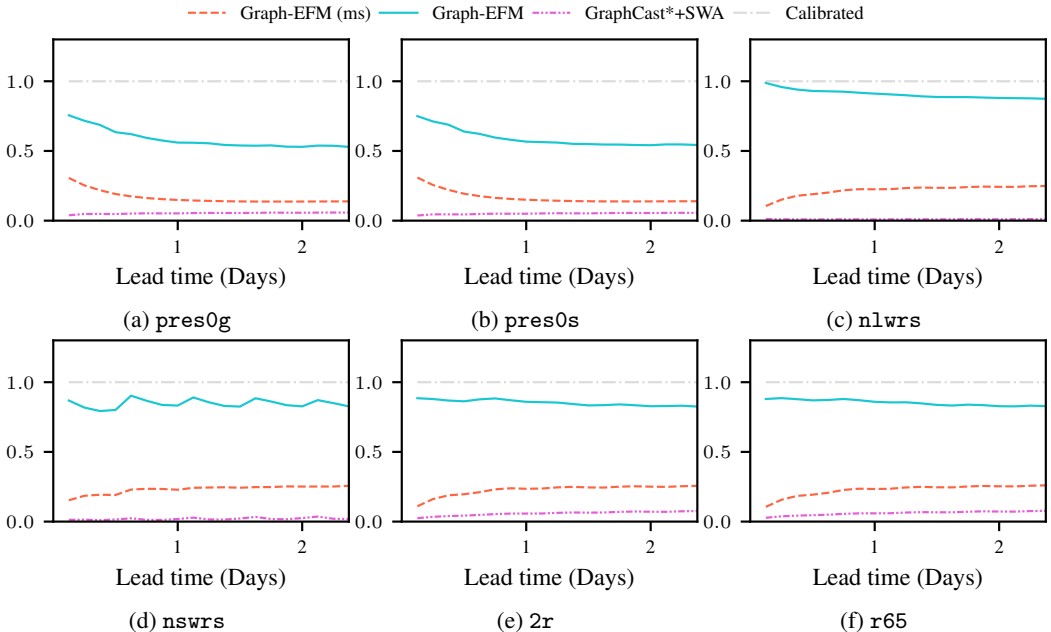

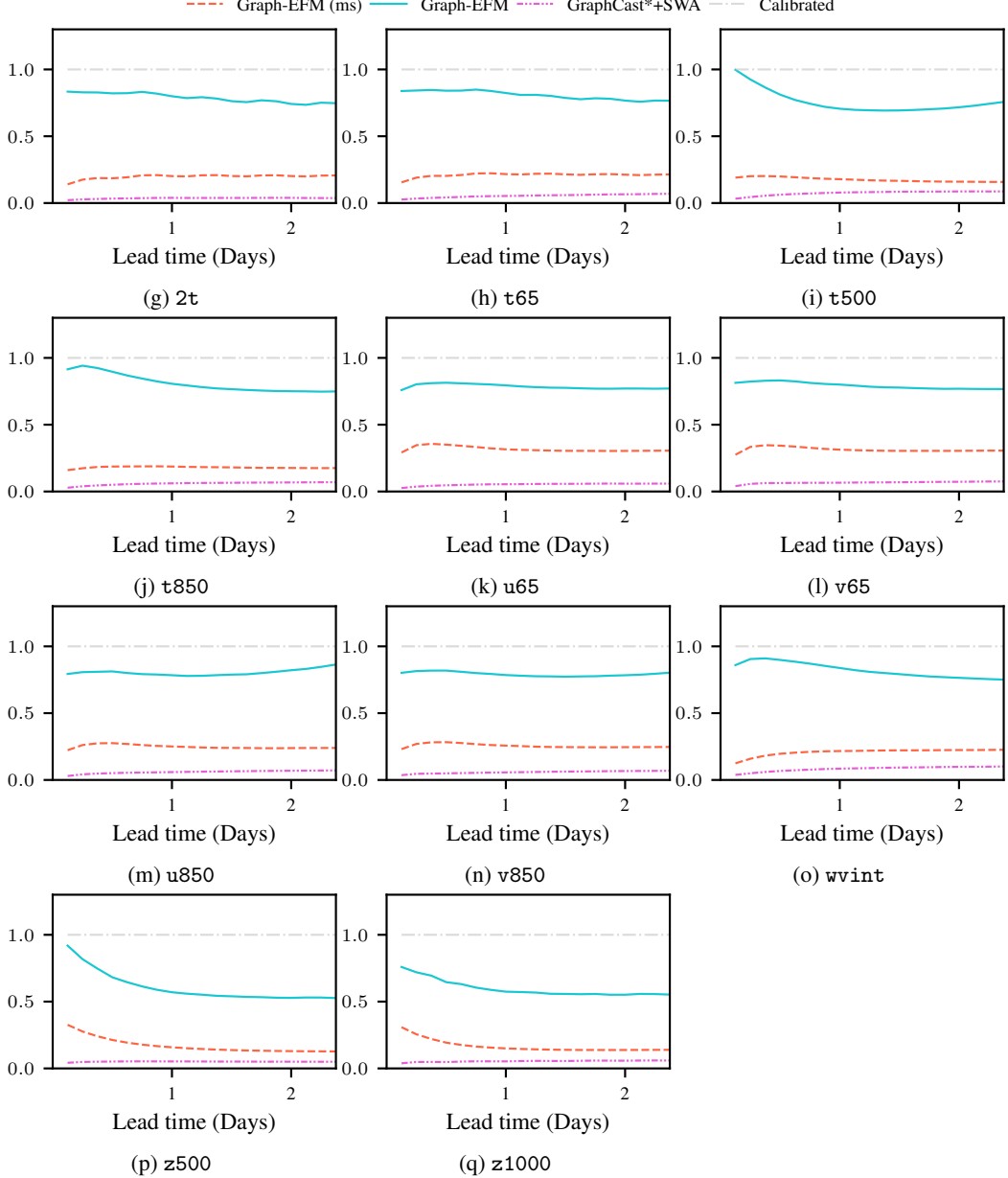

Figure 24: SpSkR for MEPS experiment.

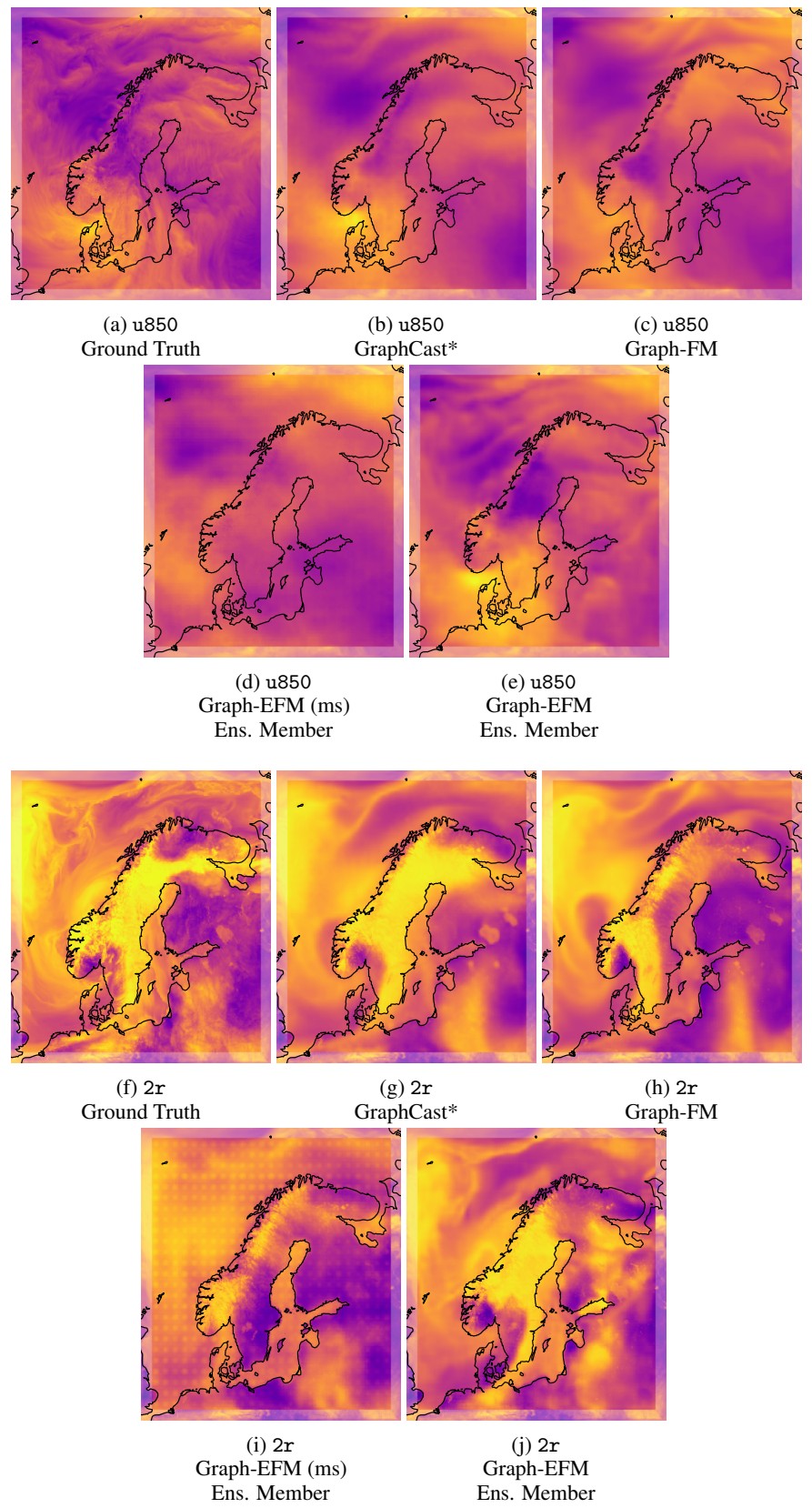

Figure 25: Comparison of LAM forecasts for u-component of wind at 850 hPa (u850) and 2 m relative humidity (2r) at 57 h lead time. For probabilistic models we show sampled ensemble members.

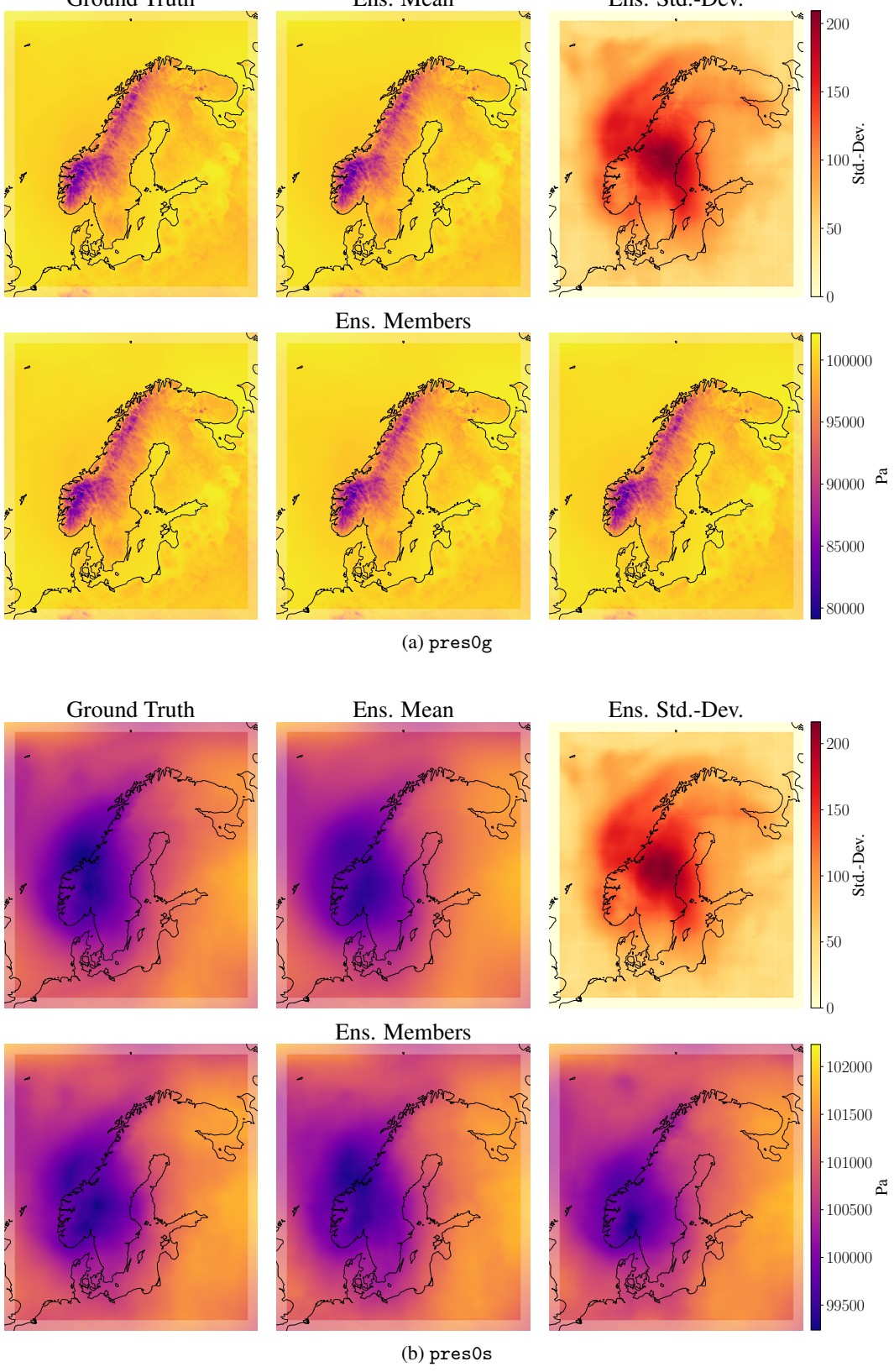

(a) `pres0g`

(b) `pres0s`

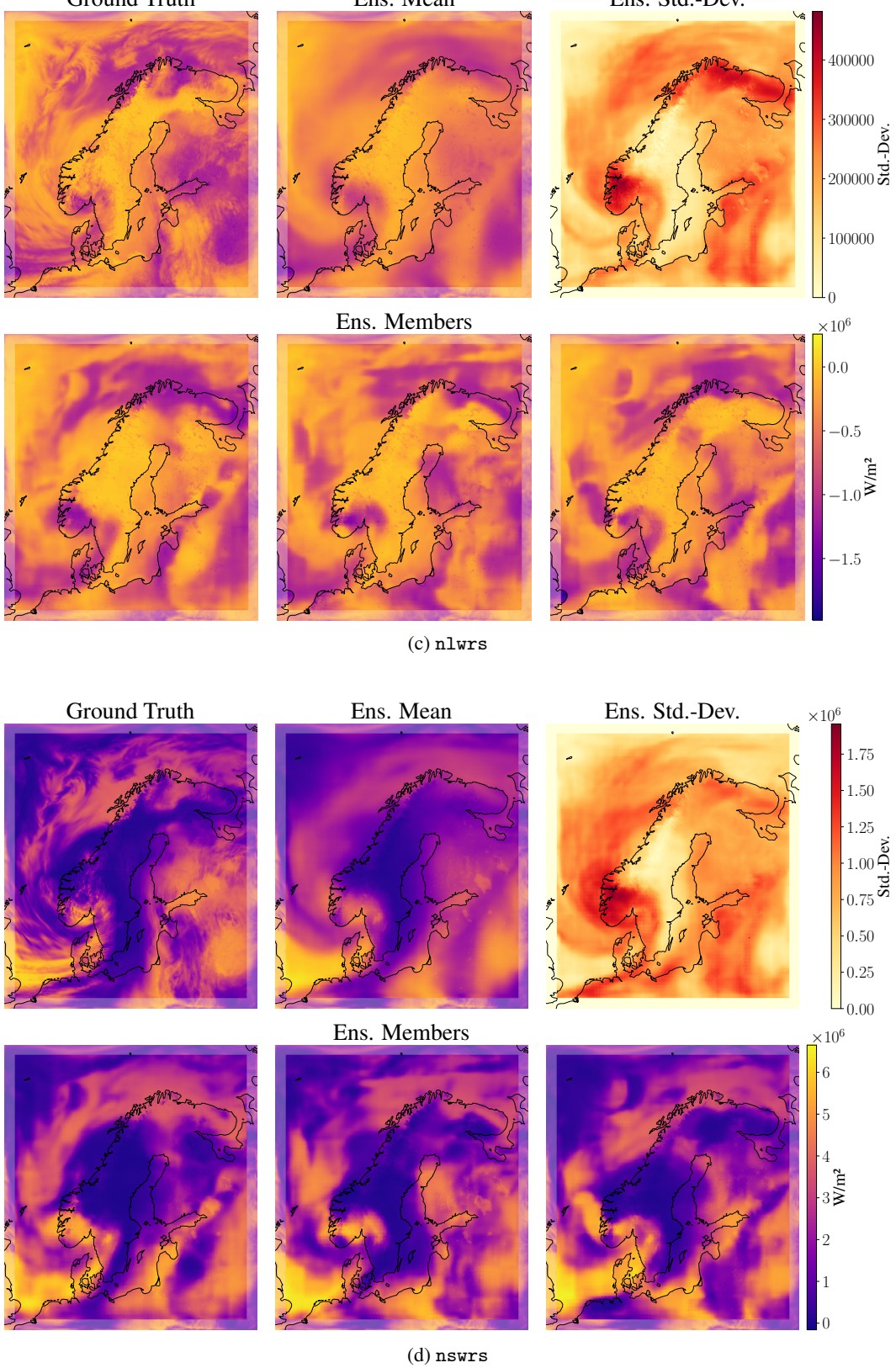

(c) `nlwrs`

(d) `nswrs`

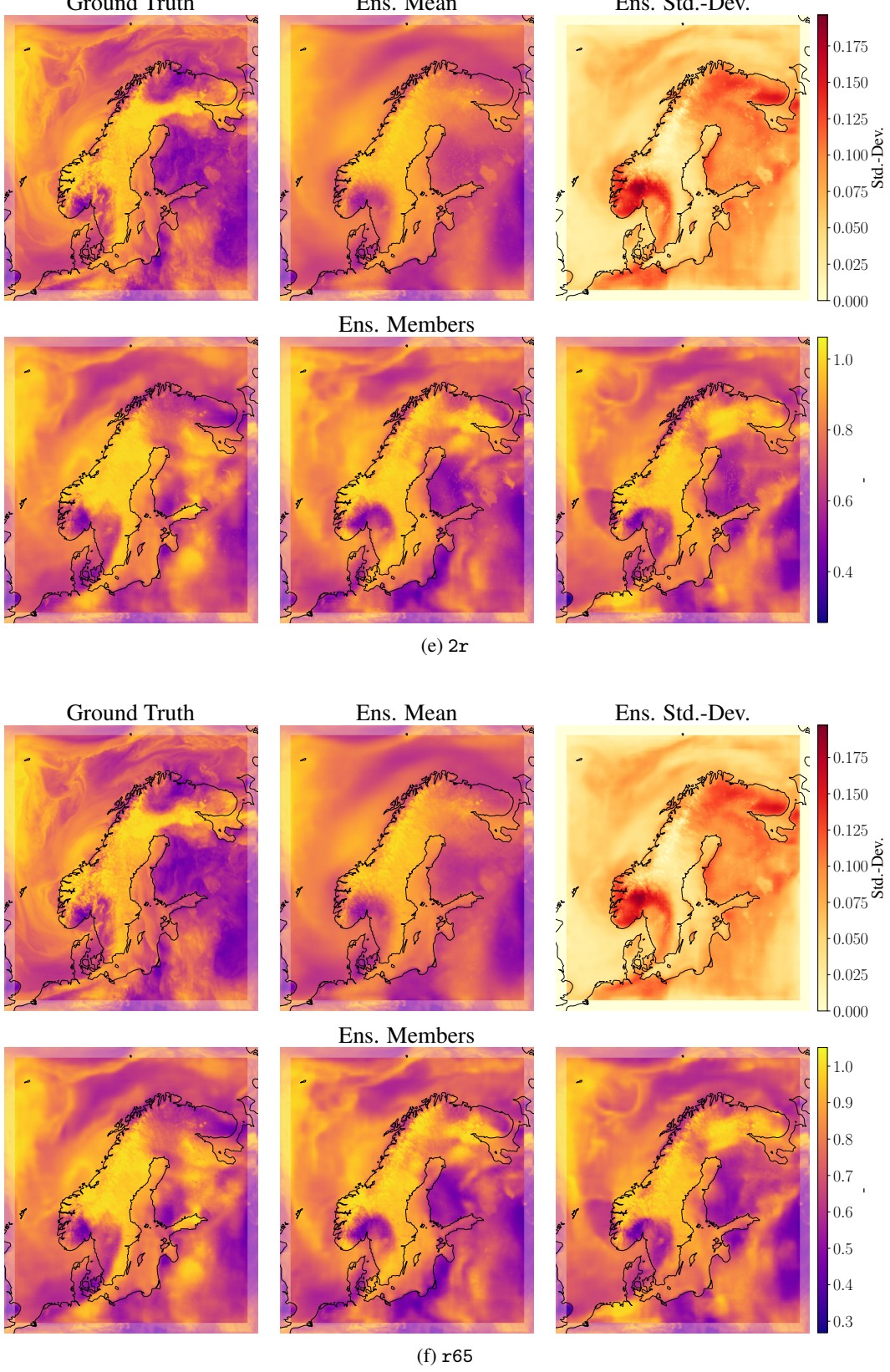

(e) 2r

(f) r65

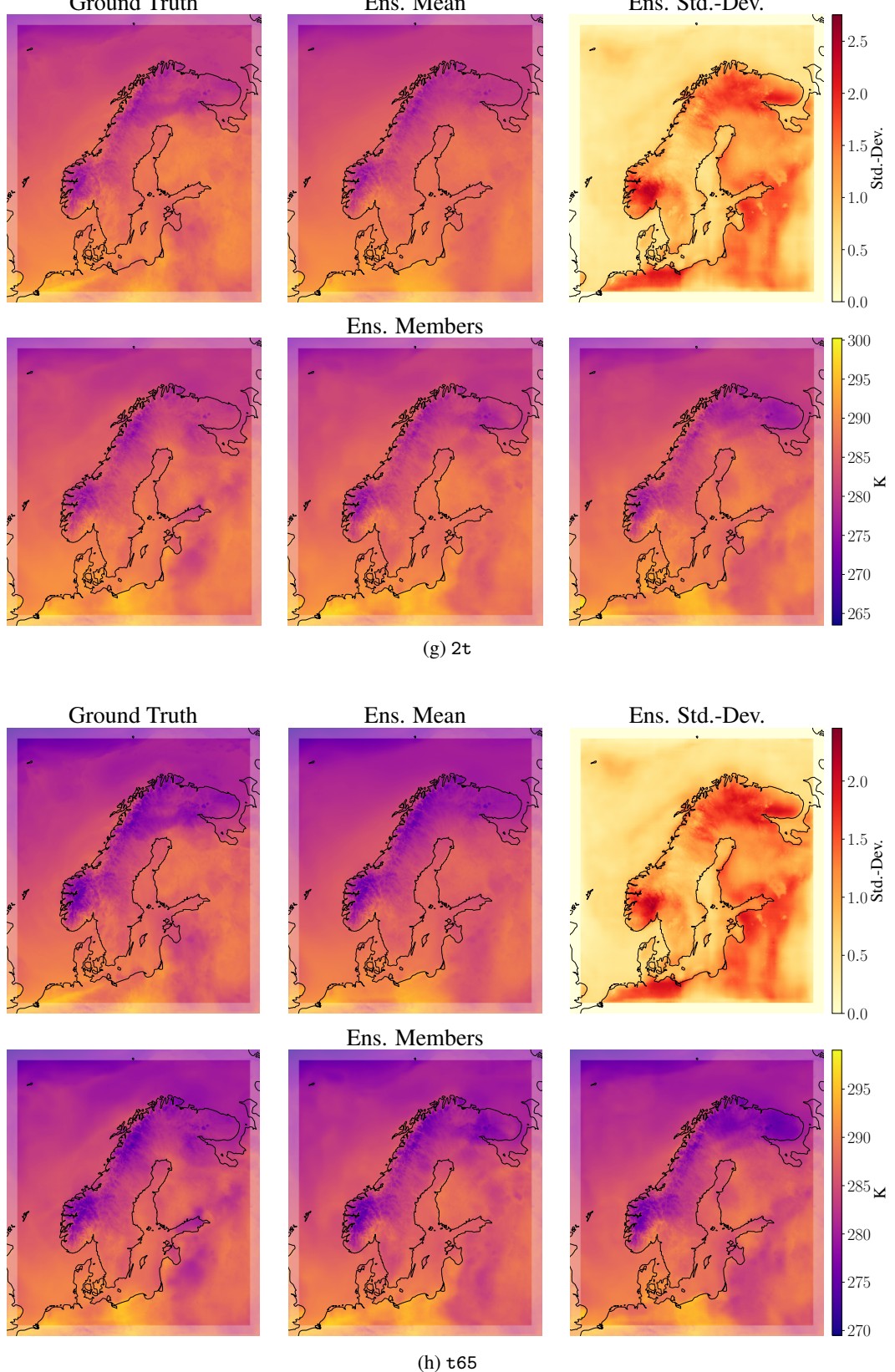

(g) 2t

(h) t65

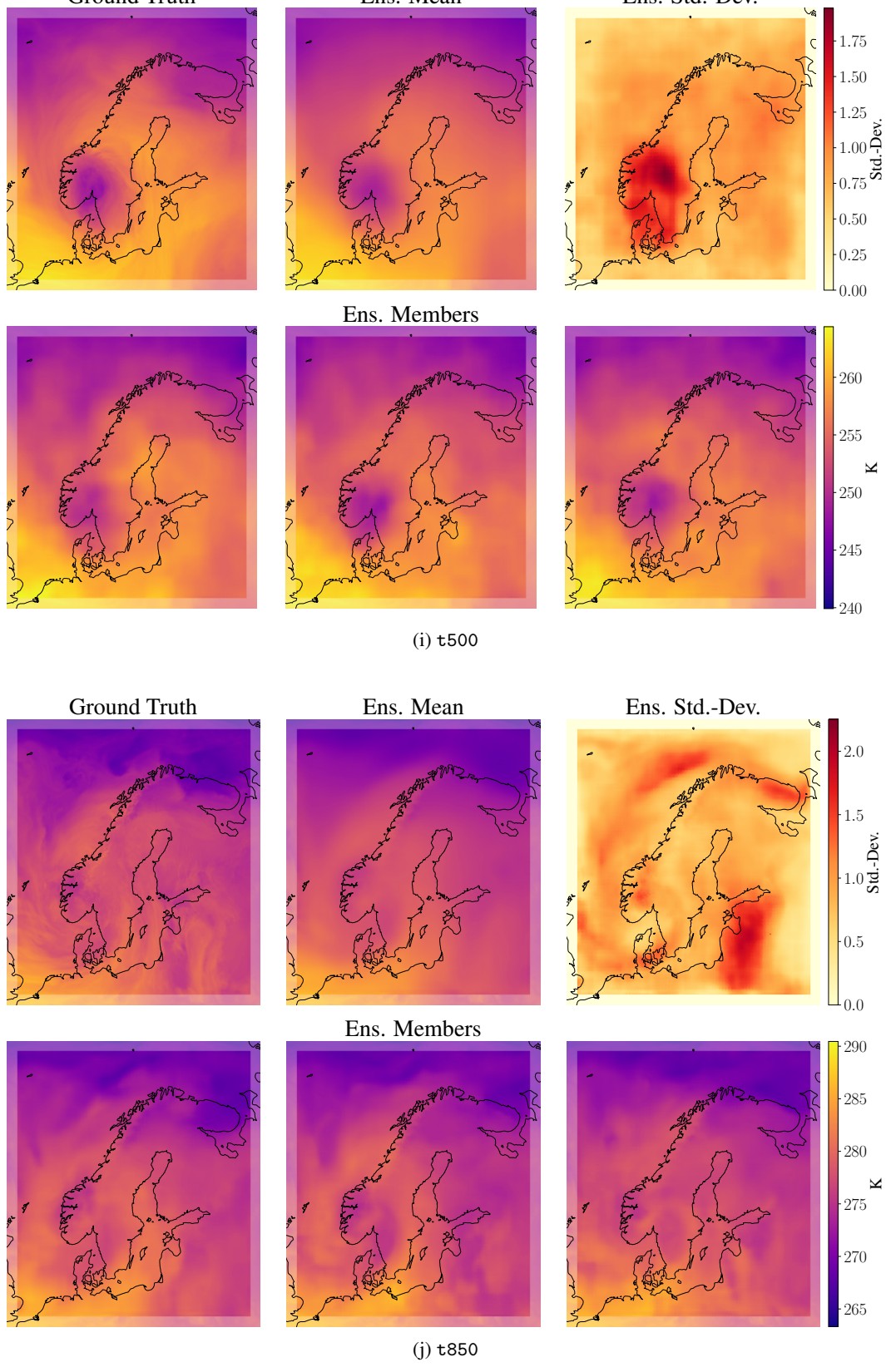

(i) t500

(j) t850

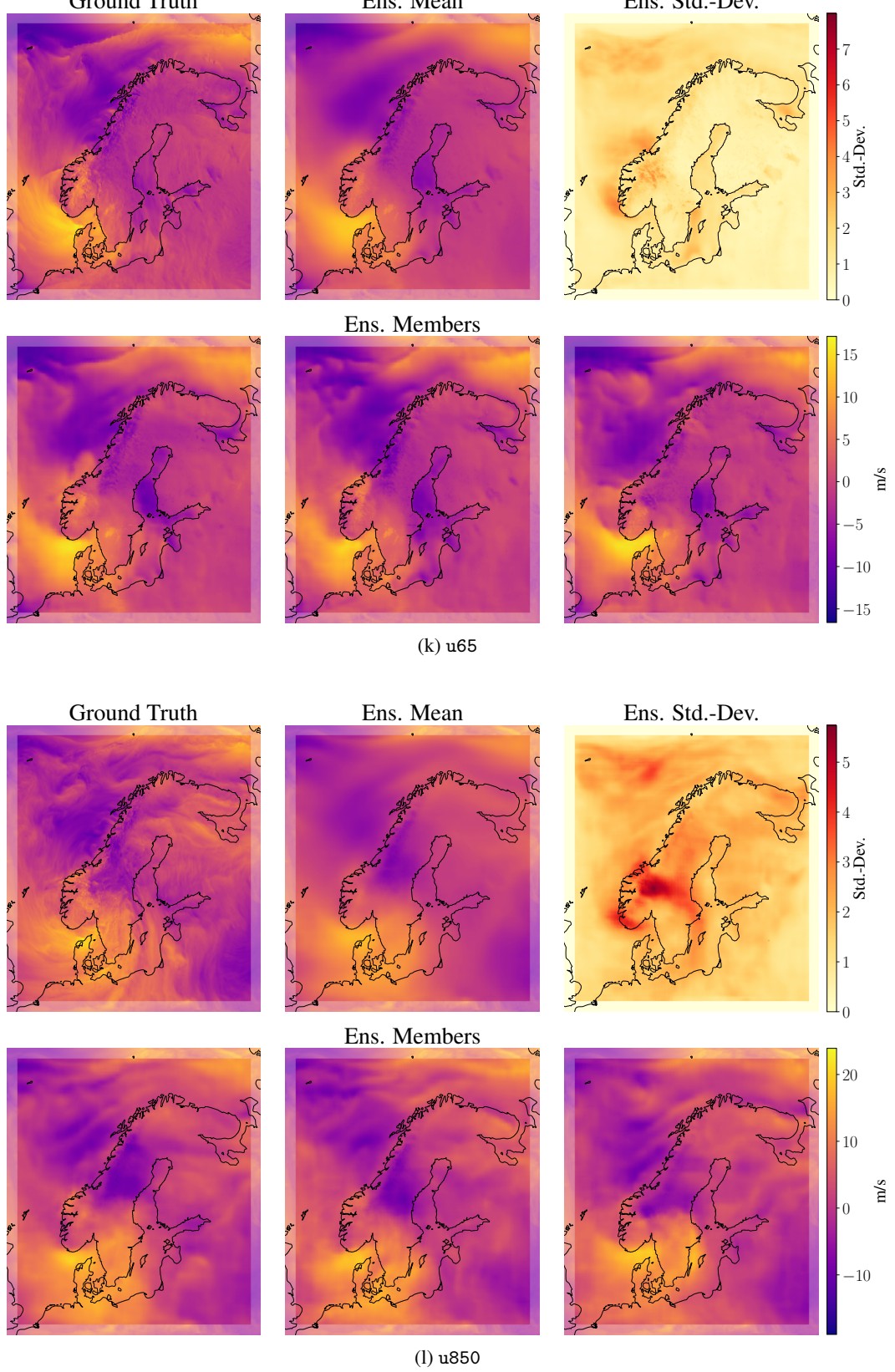

(k) u65

(l) u850

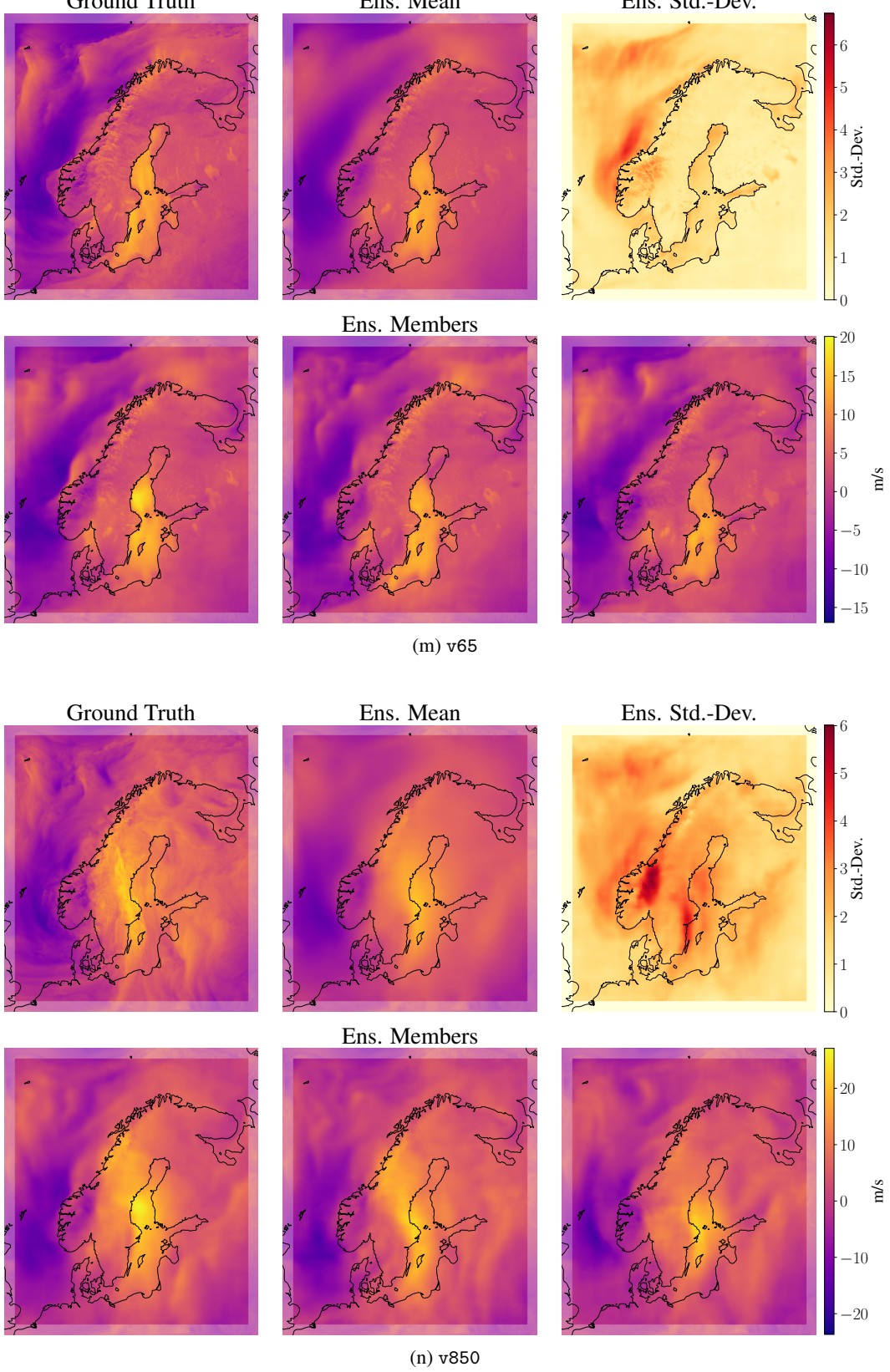

(m) v65

(n) v850

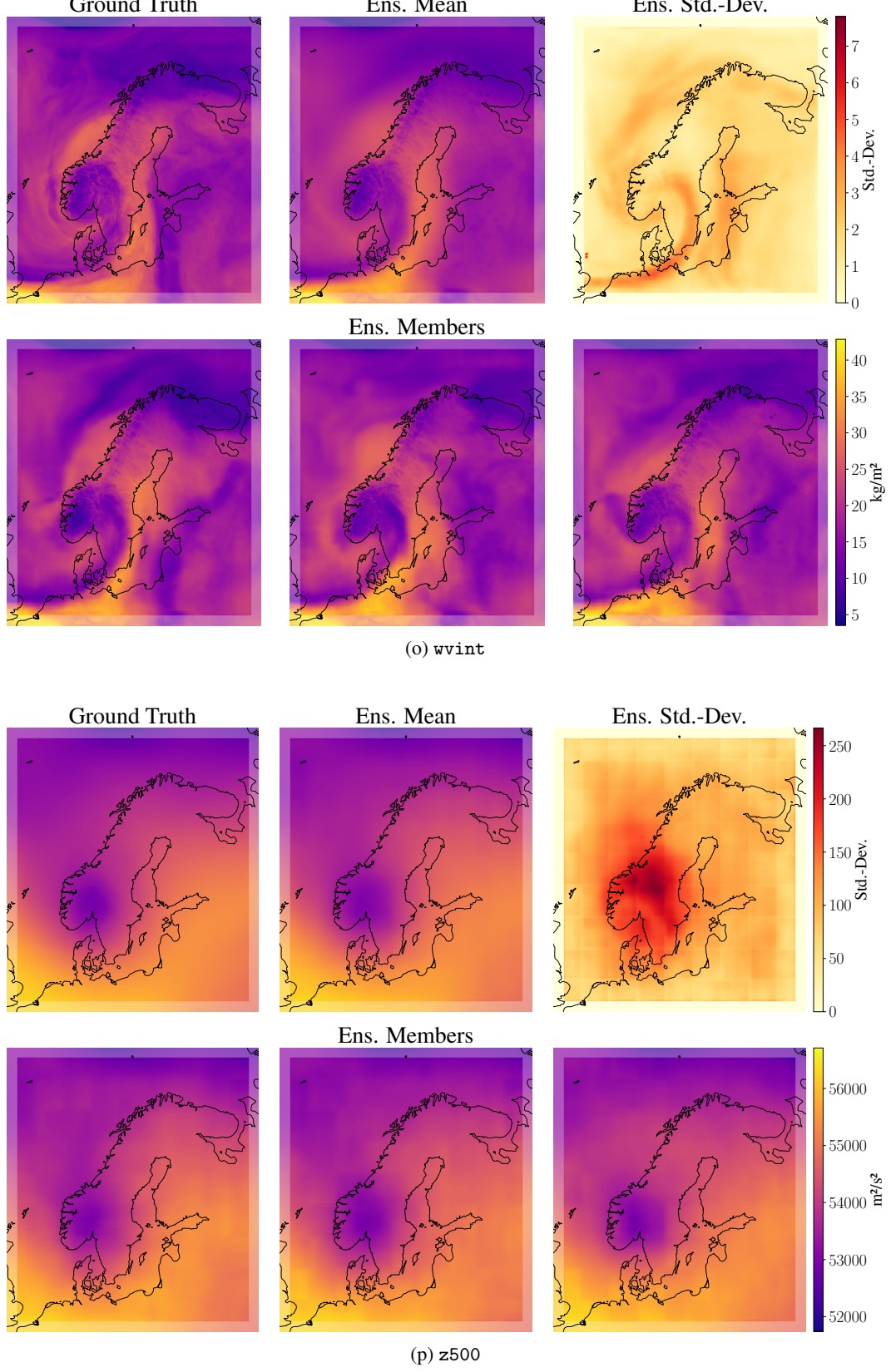

(o) `wvint`

(p) `z500`

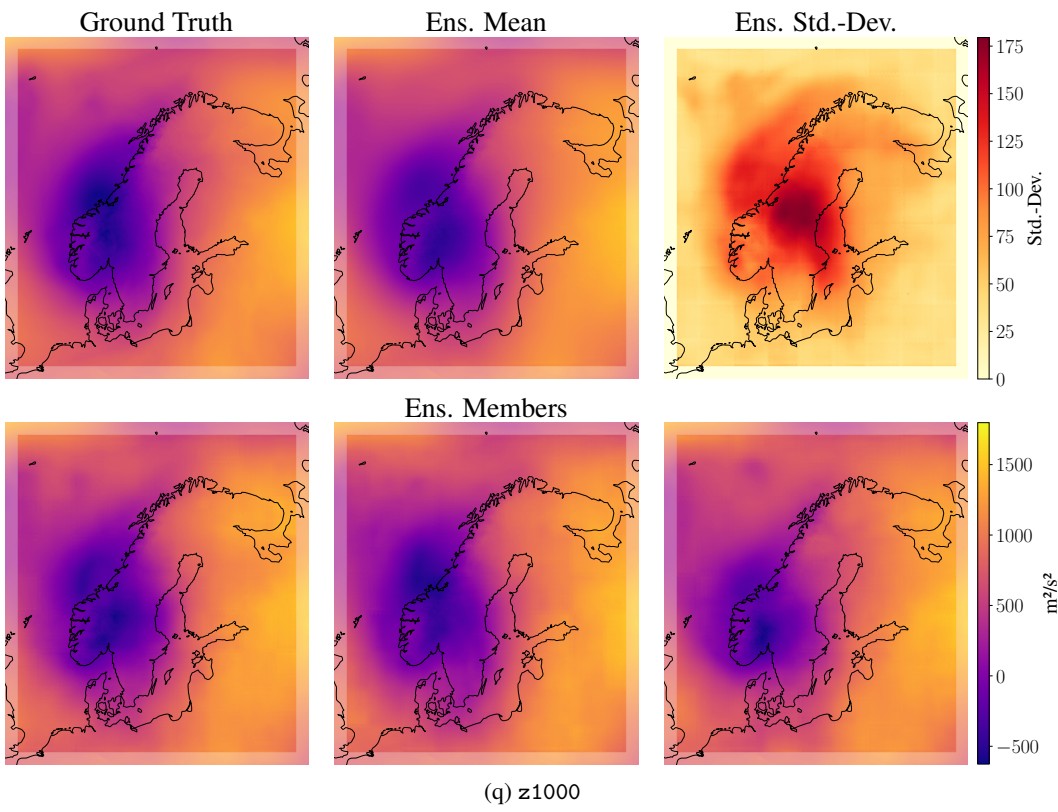

(q) z1000

Figure 26: Example Graph-EFM LAM ensemble forecasts at lead time 57 h.

# L   Additional experiments

## L.1   Comparing Interaction and Propagation Networks in Graph-FM

Given the usefulness of Propagation Networks in Graph-EFM it is reasonable to ask if these could be beneficial to use also in the deterministic Graph-FM model. Also Graph-FM uses a hierarchical mesh graph, so the propagation of information between levels is important also for this model. We test this empirically, by training versions of Graph-FM using Interaction and Propagation networks. As the retention of information is also important for some parts of the architecture, we do not replace all GNNs with Propagation Networks (see appendix C.2). We compare models on both global and LAM forecasting, using the same experimental setups as described in appendices H and I.

**Global forecasting with ERA5**   Figure 27 shows RMSEs for highlighted variables from ERA5 and Graph-FM using Interaction and Propagation Networks. The models have very similar errors for almost all variables. However, for variables in the upper atmosphere the Propagation Network model shows lower errors.

**Limited area modeling with MEPS data**   In fig. 28 we compare RMSEs for Graph-FM models using the different GNN layers on MEPS data. Here we see a greater advantage of the Propagation Networks. We hypothesize that this relates to the boundary forcing. Using Propagation Networks the information from boundary nodes should easier reach the top graph level, where it can faster spread throughout the forecasting area. Motivated by these results we use Propagation Networks in our final Graph-FM architecture, both for global and LAM forecasting.

## L.2   Importance of Latent Map

In Graph-EFM we define the latent map $p\big(Z^t\big|X^{t-2:t-1}, F^t\big)$ using a neural network with explicit dependence on $X^{t-2:t-1}, F^t$. Given that also the predictor takes $X^{t-2:t-1}, F^t$ as inputs, it is not immediately clear that conditioning on this also in the latent map is necessary. Indeed, due to the ability of deep neural networks to well approximate arbitrary functions, the predictor network should internally be able to transform a simple $Z^t \sim \mathcal{N}(0, I)$ to introduce the dependence on $X^{t-2:t-1}, F^t$. This does however assume infinite flexibility in the predictor, which might be far from the situation in practice.

To investigate the importance of a learnable latent map we compare our Graph-EFM model on the MEPS dataset with one where $Z^t$ is sampled from a static distribution $\mathcal{N}(0, I)$. This experiment was carried out using an earlier version of Graph-EFM, with the same architecture but a slightly different training schedule. To save on computations we here only sample 16 ensemble members from each model. RMSE, CRPS and SpSkR are shown in figs. 29 to 31. In terms of RMSE and CRPS there is a clear benefit to letting the distribution over $Z^t$ be a learnable mapping. The SpSkR in fig. 31 shows no clear trends between the models. For practical network architectures the latent map does add flexibility, changing the mean of $Z^t$ that enters the predictor. The learnable latent map should also simplify the inference problem solved when optimizing our variational objective. With the dependence on previous states we can expect a smaller discrepancy between $p\big(Z^t\big|X^{t-2:t-1}, F^t\big)$ and $p\big(Z^t\big|X^{t-2:t-1}, X^t, F^t\big)$, simplifying the optimization of the variational approximation $q\big(Z^t\big|X^{t-2:t-1}, X^t, F^t\big)$. We believe that this is an important aspect of the observed empirical benefit of a using the latent map.

## L.3   Impact of Ensemble Size

We here study how the performance of the model, in terms of metric values, varies when sampling different numbers of ensemble members. To investigate this we ran the evaluation of Graph-EFM with 5–80 members for the global model and 5–100 for the LAM model. Results for a selection of variables are shown in fig. 32 (global, ERA5) and fig. 33 (LAM, MEPS). As expected the RMSE of the ensemble mean decreases when sampling more members. However, already when sampling 20 or 25 members the results are fairly close to the full ensemble. For SpSkR the differences are even smaller. As the CRPS is a property of the distribution of the model forecast, its true value does not depend on the number of samples drawn. In practice we compute CRPS using an unbiased estimator, and the variance of this estimator decreases with ensemble size. When averaged spatially and over

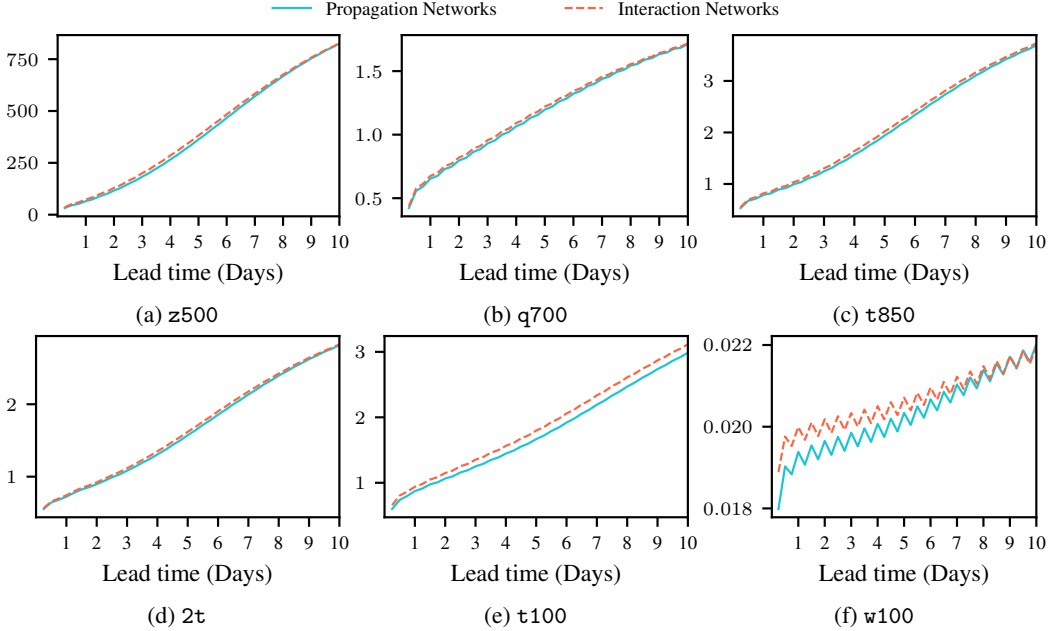

Figure 27: RMSE of Graph-FM models with Propagation and Interaction Networks, evaluated on the ERA5 test set

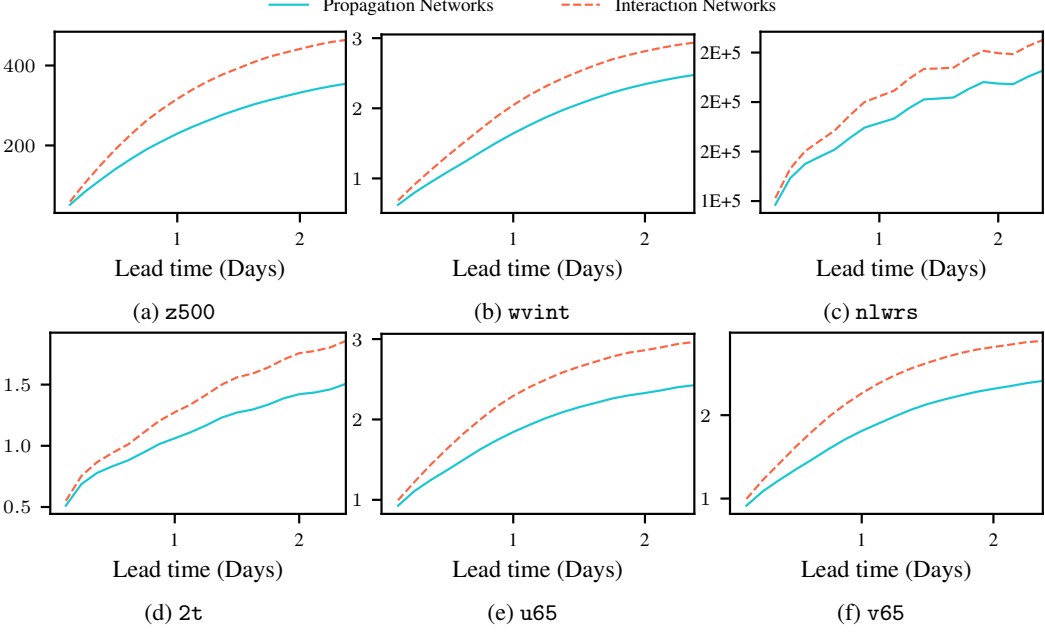

Figure 28: RMSE of Graph-FM models with Propagation and Interaction Networks, evaluated on the MEPS test set.

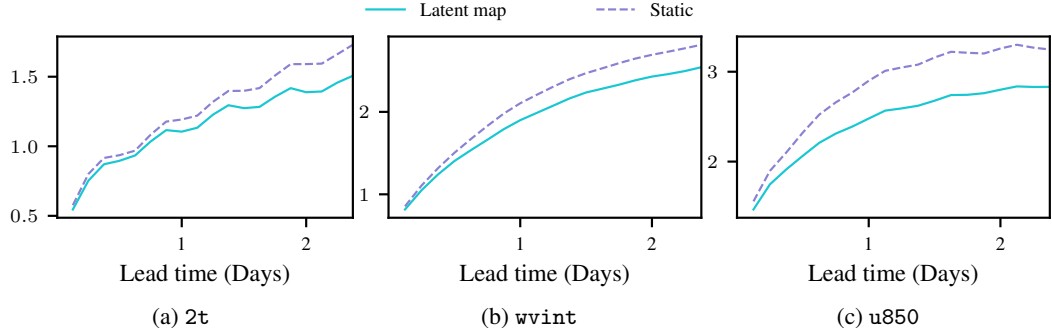

(a) `2t`                (b) `wvint`                (c) `u850`

Figure 29: RMSE for Graph-EFM models with a static distribution for $Z^t$ or a learnable latent map, evaluted on the MEPS validation dataset.

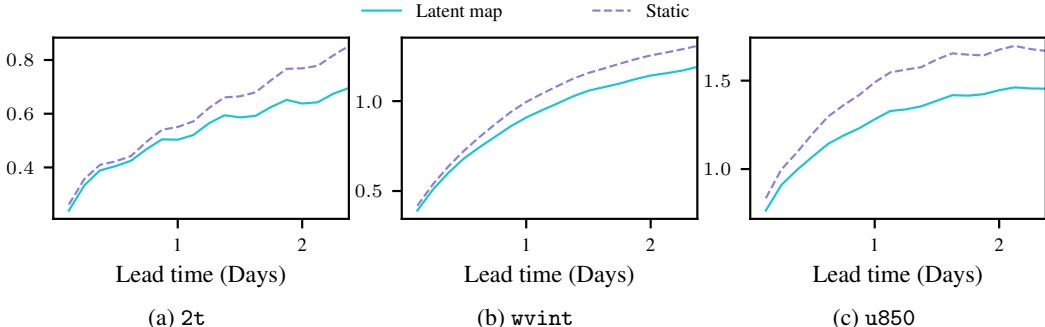

(a) `2t`                (b) `wvint`                (c) `u850`

Figure 30: CRPS for Graph-EFM models with a static distribution for $Z^t$ or a learnable latent map, evaluted on the MEPS validation dataset.

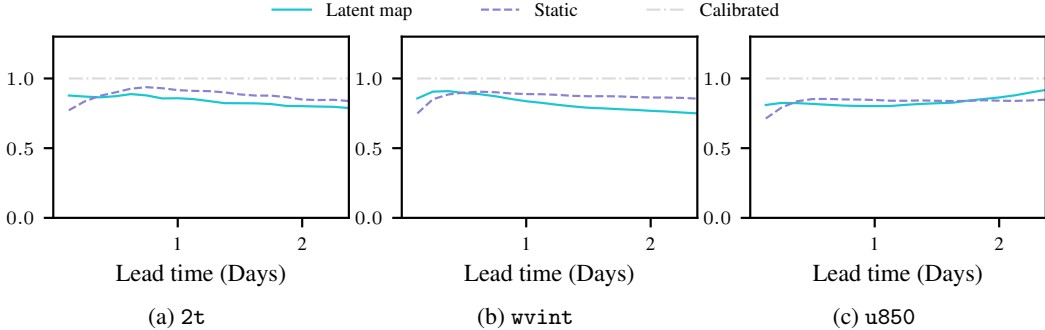

(a) `2t`                (b) `wvint`                (c) `u850`

Figure 31: SpSkR for Graph-EFM models with a static distribution for $Z^t$ or a learnable latent map, evaluted on the MEPS validation dataset.

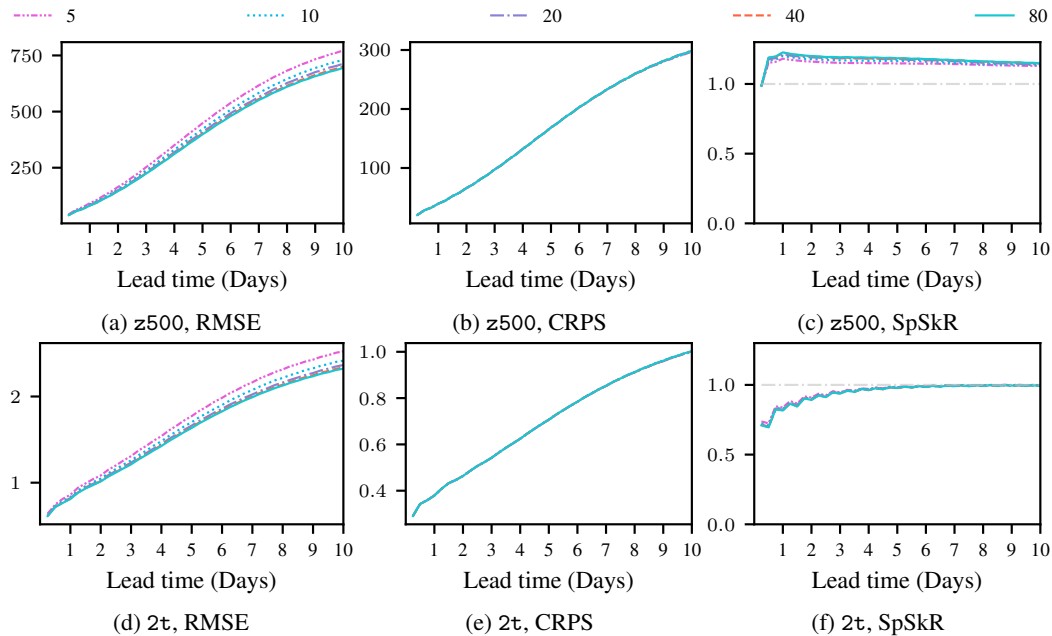

Figure 32: Metric values for z500 and 2t from the global experiment when sampling different numbers of ensemble members from Graph-EFM.

the whole test set we do however not see any difference in CRPS for different ensemble sizes. All these trends hold consistently for all variables in both the ERA5 and MEPS datasets.

In our main experiments in section 5 we use the full 80/100 member ensembles. Given that any improvements to metrics saturate, we would not expect results to meaningfully change from sampling even more members than this. It should however be noted that the motivation for sampling very large ensembles is mainly not to improve on metrics such as these. More important motivations for large ensembles include estimating probabilities of rare events or studying different possible scenarios of extreme weather.

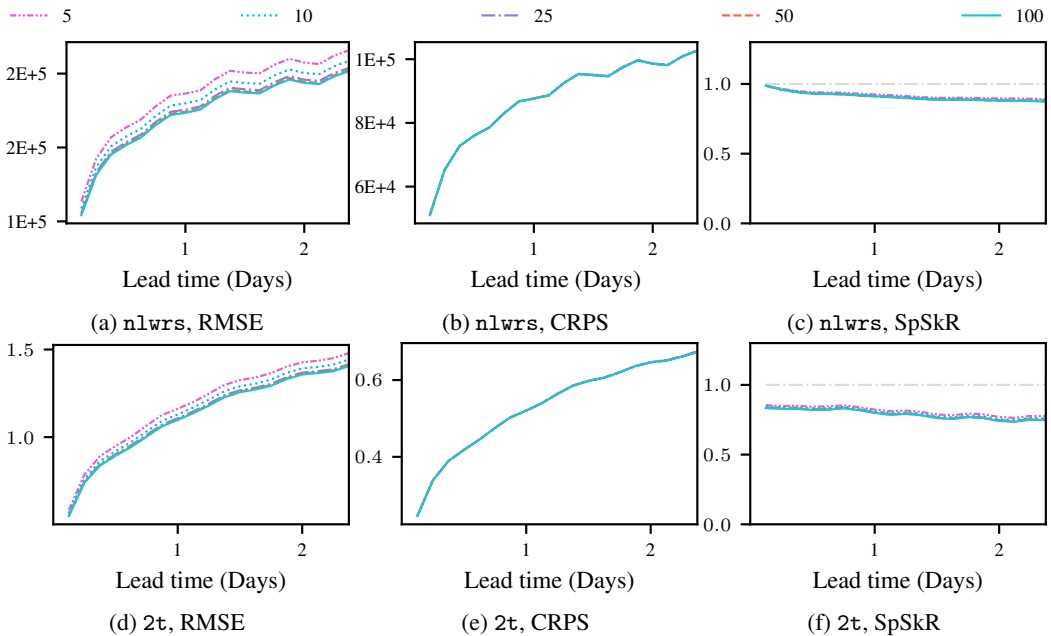

Figure 33: Metric values for `nlwrs` and `2t` from the LAM experiment when sampling different numbers of ensemble members from Graph-EFM.

## Appendix References

[53] J. L. Ba, J. R. Kiros, and G. E. Hinton. Layer normalization. *arXiv preprint arXiv:1607.06450*, 2016.

[54] Z. B. Bouallègue, M. C. A. Clare, L. Magnusson, E. Gascón, M. Maier-Gerber, M. Janoušek, M. Rodwell, F. Pinault, J. S. Dramsch, S. T. K. Lang, B. Raoult, F. Rabier, M. Chevallier, I. Sandu, P. Dueben, M. Chantry, and F. Pappenberger. The rise of data-driven weather forecasting: A first statistical assessment of machine learning-based weather forecasts in an operational-like context. *Bulletin of the American Meteorological Society*, 2024.

[55] A. J. Charlton-Perez, H. F. Dacre, S. Driscoll, S. L. Gray, B. Harvey, N. J. Harvey, K. M. R. Hunt, R. W. Lee, R. Swaminathan, R. Vandaele, and A. Volonté. Do AI models produce better weather forecasts than physics-based models? a quantitative evaluation case study of storm ciarán. *npj Climate and Atmospheric Science*, 2024.

[56] Core writing team, H. Lee, and J. R. (eds.). IPCC, 2023: Climate change 2023: Synthesis report. contribution of working groups I, II and III to the sixth assessment report of the intergovernmental panel on climate change. Technical report, Intergovernmental Panel on Climate Change (IPCC), 2023.

[57] I. Loshchilov and F. Hutter. Decoupled weight decay regularization. In *International Conference on Learning Representations*, 2019.

[58] National Weather Service. Hurricane laura, 2020. URL `https://www.weather.gov/lch/2020Laura`.

[59] P. Ramachandran, B. Zoph, and Q. V. Le. Searching for activation functions. *arXiv preprint arXiv:1710.05941*, 2017.

[60] R. Sims, P. Mercado, W. Krewitt, G. Bhuyan, D. Flynn, H. Holttinen, G. Jannuzzi, S. Khennas, Y. Liu, L. J. Nilsson, J. Ogden, K. Ogimoto, M. O'Malley, H. Outhred, Ø. Ulleberg, and F. v. Hulle. Integration of renewable energy into present and future energy systems. In *IPCC Special Report on Renewable Energy Sources and Climate Change Mitigation*. Cambridge University Press, 2012.

[61] C. K. Sønderby, T. Raiko, L. Maaløe, S. K. Sønderby, and O. Winther. Ladder variational autoencoders. *Advances in neural information processing systems*, 29, 2016.

[62] J. Whitt and S. Gordon. This is the economic cost of extreme weather. In *World Economic Forum Annual Meeting*, 2023. URL `https://www.weforum.org/agenda/2023/01/extreme-weather-economic-cost-wef23/`.

[63] M. Zamo and P. Naveau. Estimation of the continuous ranked probability score with limited information and applications to ensemble weather forecasts. *Mathematical Geosciences*, 2018.

