# OpenReview forum: "Probabilistic Weather Forecasting with Hierarchical Graph Neural Networks"
_NeurIPS.cc/2024/Conference — NeurIPS 2024 spotlight_

### Official Review · Reviewer_PLfK · 2024-07-09

**Soundness:** 3
**Presentation:** 2
**Contribution:** 2
**Rating:** 5
**Confidence:** 4

**Summary:**

This work introduces a VAE variant of GraphCast for global medium-range weather forecasting and a VAE variant of a UNet (that is formulated as a GNN) for limited area modeling over Scandinavia. For this, they adapt GraphCast to have a similar hierarchical structure to UNets, and then treat the coarsest hierarchical layer (the bottleneck) as a latent variable representing the mean of isotropic Gaussians. The ensemble predictions from the model are similarly fast as a single deterministic prediction, achieved through batching. Their calibration can be good for some variables (e.g. global t2m for 10 day lead time has a spread/skill ratio of 0.99), while poorer for others (e.g. local wvint has a spread/skill ratio of 0.57 for 24h lead time).

**Strengths:**

1. The proposed VAE extension to GraphCast is significantly faster than diffusion-based approaches (like the GenCast model).
2. The work does not limit itself to just global weather forecasting, but also presents results for limited area modeling, which is the class of models used by many national weather services.
3. The paper is reasonably well written, keeping a good amount of detail in the main paper, and presenting many additional details in the appendix.

**Weaknesses:**

Major points:
1. Questionable baselines: I am unsure if the chosen baselines are very strong, let me name a few reasons for this:
    - Tab 1 presents performance for GraphCast, e.g. RMSE=387 for z500, 5 day leadtime. However, if I check the headline scores in the WeatherBench 2 (https://sites.research.google/weatherbench/) for GraphCast, i see RMSE=274 for z500, 5 day leadtime, which is significantly higher and beats all models presented in this study.
    - Both Tab 1 and Tab 2 do not include scores for the conventional weather models. I would expect Tab 1 to include IFS & IFS-ENS scores and Tab 2 to include MEPS scores.
    - Since this work introduces a probabilistic weather model, i would expect comparison with other recent works on probabilistic weather models, like the ones cited in this paper (e.g. GenCast).
    - Graph-FM has almost 6x the parameters compared to GraphCast* (Tab 5) - which quite possibly could be the major reason for its improved performance, and not the introduced architectural feature of hierarchical layers.
2. Overlooked connection to UNets: The Graph-FM that was introduced for the LAM setting looks to me as equivalent to a UNet:
    - The input data comes on a regular grid with 236 x 268 pixels. Which is subsequently downsampled using 3x3 windows. Processing at each depth level is done with a locally connected layer (in other words: a local convolutional filter). A semantically simpler description of such a model would be a UNet with 3x3 pooling (learned in this case) and 3x3 conv filters at each stage. Possibly, implementing it as a UNet could also be computationally advantageous, making use of the highly optimized kernels for 2d convolutions and pooling operations, instead of GNN layers that rely on scatter sums.
    - UNets have been previously used for Weather forecasting: e.g. https://agupubs.onlinelibrary.wiley.com/doi/full/10.1029/2018GL080704 & https://agupubs.onlinelibrary.wiley.com/doi/full/10.1029/2020MS002203
3. Proposed VAE implementation physically not meaningful? Your VAE is implemented with a latent variable at the coarsest level. This is supposed to capture epistemic uncertainty related to the forward model (and not due to initial state). However, one may argue for atmospheric models most model uncertainty comes from the subgrid-scale parametrizations and not from the coarse-scale representation of atmospheric dynamics. Hence, to me it seems far more intuitive to introduce the stochasticity at the finest level, representing the small scales. I assume you chose the hierarchical description mostly for computational reasons, but given a lack of physical basis, i would at least expect a more thorough investigation of potential errors introduced by this, e.g. is the ensemble variability too smooth?
4. Missing reference to previously published work? A workshop paper at last years NeurIPS has introduced both the hierarchical GNN and the MEPS dataset https://arxiv.org/abs/2309.17370 , if I am not misstaken. I am not really sure about NeurIPS policy here, but even if this work is a direct extension of the previous work and the previous work is to be considered as non-archival, I still believe you should at least cite the workshop paper.

Minor points:
1. GraphCast + SWAG: This is a baseline with poor performance, that is somewhat arbitrarily picked from many possible approaches to obtain ensemble predictions from neural networks. I see two options here: Either you keep it, but also introduce many other such baselines, to make clear that you did not cherrypick a particularly weak one. Other approaches  that should not be prohibitively expensive to run could e.g. be MC-Dropout or Laplace Approximation. Or, you simply drop it, as is, it does not  add much to the paper.
2. Introduction lacks motivation for LAM: This is an ML conference that you are submitting to. It would probably be good to briefly motivate why doing LAM is even necessary (i.e., why can't we just rely on global models instead)?
3. Extreme Weather evaluation / case study: One key reason for ensemble prediction is capturing the tails, i.e. the extremes. You state in Appendix A that this is out-of-scope for the work. I would argue you are making your life too easy here. Since the presented models are likely not useful unless they display decent performance also for extreme weather, it would be important to evaluate just that. It may be enough for this paper to e.g. study a single extreme event as a case study.

**Questions:**

Why does the original GraphCast paper not report the visual artifacts that you found for Graph-EFM (ms)? Could it be that your models have simply not been trained sufficiently or that there is a bug in your implementation?

Why do you use a fixed variance of the latent variable for the global predictions? It would be interesting to see this ablation.

**Limitations:**

The section on limitations is somewhat short. I believe the key limitation of this work is in the evaluation, i.e. it is unclear to me after reading the work how well the models perform. E.g., are these models robust for longer rollouts or if applied at prediction time further away from training time?
Moreover, the work does transparently communicate a limitation of their multi-scale Graph-EFM: visual artifacts. But, this is strange to me, as it could mean two things: a) the VAE formulation and multiscale edges simply don't work well together or b) since GraphCast did not observe such issues, the presented Graph-EFM (ms) is simply not trained sufficiently or suffers from a buggy implementation. While a) would be an interesting finding, i believe b) can not be ruled out given the presented results in this paper.

---

> ### Author Rebuttal · Authors · 2024-08-06
>
> We thank reviewer PLfK for useful comments. See our response below:
>
> 1. Scores for GraphCast*
>
> As we state clearly when introducing this baseline, this is a version of GraphCast trained on the same 1.5° dataset as the other models. It thus has different scores than the original GraphCast model evaluated in Weatherbench. As we emphasize in the paper, we focus here on a fair comparison of models operating at the same resolution and trained on the same data. In the paper we refer to appendix H for comparisons with other state-of-the-art models, trained on higher resolution data.
>
> 2. Comparison to conventional models
>
> Results for IFS-ENS are included in appendix H. We realize that this might not be mentioned clearly in the main paper and will clarify. As the paper is mainly about probabilistic modeling, we find it more relevant to include IFS-ENS rather than IFS.
>
> Since the limited area modeling experiment is a pure surrogate modeling task, with data coming from MEPS forecasts, it does not make sense to compare to the MEPS model itself.
>
> 3. Comparison to GenCast
>
> See the global author rebuttal.
>
> 4. Graph-FM parameter count
>
> When instantiating models of similar size we use the dimensionality $d_z$ of representation vectors as the main scaling factor for model size. The hierarchical architecture in Graph-FM is what enables more flexibility at the same value of $d_z$, resulting also in more parameters. Note that this flexibility does not introduce a significant increase in computational cost, which would be the case if one for example added 6 times as many layers to GraphCast.
>
> 5. U-Net connection
>
> We disagree that this connection is overlooked and refer the reviewer to the related work section where we clearly note this similarity. It should however be emphasized that our GNN model can not be seen as *equivalent* to a U-net. While some hierarchical structure is shared, the GNN layers can not be reduced to simple convolutions, even in the LAM setting. For example, these feature edge representation updates that have no correspondence in convolutional layers.
>
> We would also like to emphasize that we present a general framework that is applicable to different forecasting regions and graph constructions. The exact similarity to a U-net model will depend on these choices. In the specific LAM case from section 5.2 this connection is stronger (although the models are not equivalent), whereas in the global forecasting setting this connection is more conceptual.
>
> 6. Physical meaningfulness of VAE implementation
>
> We would like to clarify that while the latent variables are associated with the top level of the graph, this does not mean that they only influence the coarsest spatial scales. As the GNN layers of the predictor are applied through the graph hierarchy the randomness in the latent variables can spread to all parts of the graph and introduce variations on different scales. However, explicitly constraining different latent variables to control variation at different spatial scales is an interesting idea for future work.
>
> We have found that introducing the latent variables closer to the grid points in the model can lead to less spatially coherent forecasts. Graph-EFM (ms) can act as an ablation of this, as the latent variables are there directly associated with the nodes of the single level multiscale graph. We found Graph-EFM (ms) to be harder to train and produce worse forecasts.
>
> 7. Reference to workshop paper
>
> By the NeurIPS policy, workshop papers not published in proceedings are not to be counted as publications. Therefore we did not explicitly cite the mentioned workshop paper. However, if it is deemed advisable to do so we would not mind including a reference to it. Perhaps the AC can help clarify the situation.
>
> 8. GraphCast* + SWAG baseline
>
> This baseline was not picked arbitrarily, but as described it was inspired by the usage of SWAG in Graubner et al. The point of this baseline is to include a comparison to multi-model ensembles, which we see can produce too little variability. While there are indeed other ways to achieve this, we wanted to focus on a method that has been proposed in the ML weather prediction literature, rather than choosing a method arbitrarily.
>
> 9. Motivation for LAM
>
> We currently give some motivation for this in section 5.2, but we do agree that this can be expanded on already in the introduction of the paper. We will change this.
>
> 10. Extreme weather case study
>
> We have now included such a case study for Hurricane Laura, that can be found in the global author rebuttal.
>
> 11. Fixed variance of latent variable for global predictions
>
> There might be some misunderstanding here, as the variance of the latent variable (Eq. 4) is fixed for both the global and LAM models. However, the variance $\sigma^2_{\alpha, j}$ of the likelihood term (Eq. 6) is fixed in the global setting while output by the model in LAM. See Appendix C for more discussion about this.
>
> 12. Robustness for longer rollouts
>
> Note that all models are unrolled for much fewer time steps during training than at inference time. In the global case for example, Graph-EFM is unrolled 40 steps for evaluation at 10 days lead time, but only trained by unrolling up to 12 steps. As we focus here on weather forecasting rather than climate modeling, the stability of unrolling the models for months or years is of limited interest in this work.
>
> 13. Visual artefacts in Graph-EFM (ms)
>
> We would like to emphasize that when considering the same setting as in the GraphCast paper (global, deterministic modeling) we do not see any artefacts. The artefacts in Graph-EFM (ms) are mainly observed in the LAM setting (e.g. Figure 6.d) and for the global probabilistic model, none of which were considered in the GraphCast paper. Note however that the final global Graph-EFM (ms) model does not have these artefacts, as the artefacts mentioned as limitations only appear when training with a poor choice of $\lambda_{\text{CRPS}}$.

---

> > ### Comment · Reviewer_PLfK · 2024-08-11
> >
> > Dear Authors,
> >
> > thank you for your effort to respond to the points raised in my review.
> >
> > After reading your rebuttal and the referenced sections in the paper again, I would like to ask you for some further clarifications.
> > I believe the merit of this work mainly consists in an experimental study of cleverly adjusting multiple known concepts from main-stream deep learning to probabilistic weather forecasting. Hence, it is most important to demonstrate two things: 1) that the new architectures are robustly advantageous compared to existing state-of-the-art and 2) that these advantages are relevant for practical use.
> > For computational reasons you choose to train and evaluate on ERA5 at 1.5° resolution, which is arguably not relevant for practical use, and also not the resolution at which current SOTA is trained. You try to mitigate this by re-implementing one of the SOTA methods (GraphCast) to run at 1.5°, and then show at that resolution your proposed changes lead to similar performance deterministically, but can create fast probabilistic ensembles with improved skill. The crux is, your experiments do not demonstrate that these advantages are robust to scaling.
> > Also you argue you are not comparing to probabilistic SOTA (e.g. GenCast), because code is not open source. But given you reimplemented GraphCast, i suspect you should be able to reimplement GenCast also. So while I understand your reason, I do not believe you can shy away so easily from comparing against SOTA probabilistic models (or also: initial condition perturbations, as other reviewers point out).
> > For the LAM models, thank you for clarifying that you evaluate against MEPS forecasts, I overlooked that. While this is of course valid to compare the different neural networks, it does not give an indication how good the emulation is. Now a weather service would arguably only use the emulator if they understood, what they loose in terms of performance compared to the conventional model, and more importantly, if the performance drop is not too large. For this it would be important to have an independent reference. For instance, the cited reference [37] for AROME-MetCoOp evaluated on 154 synoptic weather stations, at such stations, you could compare the performance of your neural network emulators against the original MEPS forecast, and also against the IFS-ENS. It would be important to show that your emulators are at least outperforming the IFS-ENS.
> >
> > Regarding the difference to UNets, could please elaborate what you mean with "While some hierarchical structure is shared, the GNN layers can not be reduced to simple convolutions, even in the LAM setting. For example, these feature edge representation updates that have no correspondence in convolutional layers."?
> > Let me elaborate on why I believe they could be basically the same: On a regular grid, a message passing layer is a nonlinear transformation of local features (both edges and nodes), followed by a (possibly weighted) sum of multiple features per grid cell. If the graph is such that the weighted sum includes features that are e.g. in 2x2 quadratic neighborhoods, then this operation is essentially a kernel. In UNets we learn these kernels. For the nonlinear transformation of local features, you can use 1x1 Conv layers. For downsampling, you just use dilation, for upsampling, linear interpolation followed by 1x1 Conv.
> > There would of course be a fairly simple route to alleviate my concern here: actually training a UNet-based VAE on the MEPS data and compare the performance to the proposed Graph-EFM, but I do understand there is too little time left for the rebuttal to do this. By the way, I consider this point to only influence the assessment of methodological novelty in this paper, but if you can prove that your MEPS emulators are indeed practically useful, such methodological novelty might not even be necessary at all.
> >
> > Regarding your clarification 6, you write "We would like to clarify that while the latent variables are associated with the top level of the graph, this does not mean that they only influence the coarsest spatial scales.", which is clear to me, but was not the point of my concern. Maybe my writing was not clear, let me restate. Because you only draw random variables at the top level, the model lacks fine-scale randomness beyond the top level resolution. This does not mean that the top level random variables can not influence finer scales, but rather that this influence is deterministic. By the way, this touches upon the point of comparing to IC perturbations as a baseline: quite possibly, the uncertainty your model outputs is related to the initial conditions, instead of the forward atmospheric model, so running Graph-FM with perturbed IC and comparing with Graph-EFM with perturbed IC could give insights here. But again, too little time left in the rebuttal..
> >
> > Nonetheless, I will reconsider my rating after the rebuttal period.
> > Thanks!

---

> > > ### Author Response · Authors · 2024-08-13
> > >
> > > Thank you reviewer PLfK for your careful consideration of our response. We are happy to discuss and clarify our points further!
> > >
> > > 1. Regarding the method contribution
> > >
> > > We find it important to point out that we are not just applying off-the-shelf methods to weather forecasting, performing a purely empirical study. As all deep learning papers we build upon established methods, but combine these in novel ways motivated by the application. In the paper we contribute details all the way from training objectives to the internal workings of GNN layers. Hence we think our contribution is also methodological, with a focus on making the method work for weather forecasting.
> > >
> > > 2. Regarding scaling to higher spatial resolutions
> > >
> > > It is true that we limit the study to 1.5° resolution data for computational reasons. As we are sure the reviewer is aware, but we would like to clarify here for completeness, the computational requirements for training models on 0.25° data are substantial. Already now we use 8 high-end GPUs for a week in order to train one model. For comparison, the original training of GraphCast on 0.25° data took 4 weeks on 32 devices. By considering this and our training times for GraphCast* and Graph-EFM, we can get a rough estimate that training a 0.25° version of Graph-EFM would take more than 6 months using the same 8 GPUs.
> > >
> > > In our opinion it is crucial that new ideas, models, and methods can be proposed and evaluated at scales manageable by academic researchers who do not have access to the same computational and engineering resources as a company like DeepMind. This allows for much faster advancements in machine learning methodology for the ML weather prediction area and enables for the best ideas to then be integrated and scaled up in high resolution models. Indeed, we believe that the evaluation that we present in the paper on a lower resolution is indicative of the results that one might expect on a higher resolution, and that these results are sufficient to show that the model that we propose is of interest to the research community.
> > >
> > > 3. Clarification regarding GenCast comparison
> > >
> > > We would like to reiterate that for GraphCast there is code openly available, making it much more feasible to implement and retrain. While the GraphCast* model is implemented in our codebase, we still rely on the GraphCast code for parts of the global model. It is of course not impossible to reimplement and retrain GenCast, but doing this without access to the implementation or detailed documentation would risk getting important details wrong, leading to unfair comparisons.
> > >
> > > 4. Regarding the evaluation of LAM models
> > >
> > > The perfect emulator of a system should be one that gives exactly the same output as the system. In this way it makes sense to evaluate a MEPS emulator by comparing to what the MEPS system outputs (the forecasts). However, we do agree with your point that this is not the full story here. As the MEPS system itself has some error w.r.t. the true weather, the error of the emulator should be considered in this context. The reported error of the emulator (compared to MEPS forecasts) can in this way be seen as an upper bound on how much worse its performance is compared to MEPS (on observations).
> > >
> > > While we do not have data for synoptic stations readily available at the moment to run such an evaluation, one can use the values reported in [37] to put errors into some context. Figure 9a in [37] shows MAE of 2m temperature for different months in 2014 and 2015. The corresponding MAE, when comparing Graph-EFM to the MEPS forecasts in the test set, is 0.61 (and similar for other models). Note that we have reported RMSEs rather than MAEs in the paper, but we have evaluated also the MAE values. An increased error of 0.61 is not negligible, but comparable to the difference between IFS and MEPS over Norway in the figure. Now if one considers trading off the superior MEPS forecasts for a model with a similar error to IFS over the region, but that runs in seconds and can create arbitrary large ensembles, this has some clear usefulness.  This comparison comes with a whole bag of caveats (especially since we are comparing values for different years), but the point is mainly to give some example of how the magnitude of emulation errors relates to model errors on observations.
> > >
> > > (See continuation in answer below)

---

> > > > ### Author Response · Authors · 2024-08-13
> > > >
> > > > 5. Regarding U-net connection
> > > >
> > > > We are happy to elaborate on our answer here, as the rebuttal left us a bit short on space to discuss this point. It is indeed possible that you can put together the components you have listed to construct something that would be basically the same as our LAM model using the specific graph constructed for the MEPS area. To get something *equivalent* to the InteractionNetwork layers would still require quite some careful tailoring of these components, and would not resemble a standard off-the-shelf U-net. Although we do agree that a comparison to an off-the-shelf U-net model (possibly also with a VAE modification) would indeed be interesting, even though we lack the time now.
> > > >
> > > > We again want to stress that this discussion about U-net similarity is fully confined to the specific graph created for the MEPS experiments. This should be interpreted as one example application of a general framework. We think that this aligns with the presentation in the paper. The global model is another example of applying the framework, that is clearly different from a U-net. The specific LAM graph construction is one example of how a LAM graph could be created. There are many options to further explore there, including triangular and irregular graphs, which have no equivalence to a U-net.
> > > >
> > > > 6. Regarding clarification 6 (randomness at different spatial scales)
> > > >
> > > > Thank you for the clarification, we understand your point better now. We base our model on an assumption that the effective dimension of the dynamics is lower than the dimension of the grid. This is modelled by setting the dimension of the latent random variable to be significantly lower than the data. We agree with your point that sampling the randomness at the top level and deterministically mapping this down does limit what kind of variation that can be modeled. However, this is part of our modeling assumption and since the mapping is non-linear and the random variables contain many dimensions of randomness we do not believe it to be a significant limitation. We back up this modeling choice with our experiments using Graph-EFM (ms), where introducing the noise closer to the grid has resulted in a worse model with less realistic forecasts.
> > > >
> > > > We are not entirely sure about how you mean the uncertainty in the model would be related to ICs rather than the dynamics. As the forecast is sampled per time step, the randomness is included in the mapping from one time point to the next, rather than only between time 0 and 1. Uncertainty in initial states corresponds to a distribution $p(X^{-1:0})$ and combining initial state perturbations with a deterministic forecasting model corresponds to sampling from some approximation of this distribution and then applying deterministic dynamics (meaning that $p(X^{1:T} | X^{-1:0}, F^{1:T})$ is a dirac distribution). This is fundamentally different from what we do in Graph-EFM, where we model the distribution $p(X^{1:T} | X^{-1:0}, F^{1:T})$ as a product of random dynamics $p(X^t | X^{t-2:t-1}, F^t)$ at each time point (eq. 2).
> > > >
> > > > But to give some more clarity on this matter we have tried evaluating the models with perturbations added to the initial conditions. Within this short time frame we have only been able to do a simple experiment here, where we use our pre-trained models and add independent Gaussian noise to $X^{-1:0}$. We tune the scale of this noise on the validation set. The results for models with Initial Condition Perturbations (ICPs) are shown in the tables below.
> > > >
> > > > (See continuation below for tables)

---

> > > > > ### Author Response · Authors · 2024-08-13
> > > > >
> > > > > **Global results** (corresponding to Table 1 in paper)
> > > > >
> > > > > |           |                   |       | 24h   |        |       | 57h   |        |
> > > > > |-----------|-------------------|-------|-------|--------|-------|-------|--------|
> > > > > | Variable  | Model             | RMSE  | CRPS  | SpSkR  | RMSE  | CRPS  | SpSkR  |
> > > > > | z500      | Graph-EFM + ICP   | 410   | 193   | 1.34   | 707   | 319   | 1.20   |
> > > > > |           | Graph-FM + ICP    | -     | -     | -      | -     | -     |        |
> > > > > |           | GraphCast* + ICP  | 475   | 244   | 0.66   | 914   | 486   | 0.46   |
> > > > > | 2t        | Graph-EFM + ICP   | 1.66  | 0.73  | 1.07   | 2.36  | 1.03  | 1.06   |
> > > > > |           | Graph-FM + ICP    | -     | -     | -      | -     | -     | -      |
> > > > > |           | GraphCast* + ICP  | 2.14  | 1.11  | 0.31   | 3.48  | 1.82  | 0.35   |
> > > > >
> > > > > (Unfortunately the results for Graph-FM + ICP did not finish in time, and we do not want to delay our answer further.)
> > > > >
> > > > > **LAM results** (corresponding to Table 2 in paper)
> > > > >
> > > > > |           |                   |       | 24h   |        |       | 57h   |        |
> > > > > |-----------|-------------------|-------|-------|--------|-------|-------|--------|
> > > > > | Variable  | Model             | RMSE  | CRPS  | SpSkR  | RMSE  | CRPS  | SpSkR  |
> > > > > | z500      | Graph-EFM + ICP   | 174   | 90    | 0.98   | 222   | 113   | 0.72   |
> > > > > |           | Graph-FM + ICP    | 233   | 124   | 0.73   | 357   | 192   | 0.43   |
> > > > > |           | GraphCast* + ICP  | 156   | 82    | 0.93   | 200   | 109   | 0.42   |
> > > > > | wvint     | Graph-EFM + ICP   | 1.63  | 0.8   | 0.9    | 2.11  | 1.02  | 0.83   |
> > > > > |           | Graph-FM + ICP    | 1.67  | 0.89  | 0.36   | 2.50  | 1.39  | 0.22   |
> > > > > |           | GraphCast* + ICP  | 1.54  | 0.82  | 0.38   | 2.10  | 1.18  | 0.20   |
> > > > >
> > > > > We can note that the ICPs do not result in improved RMSE values for any model, instead making it worse. Visually inspecting the samples (sadly we cannot include figures here) show that the noise tends to stay in the forecast over time, and does not induce physically plausible diversity in the ensemble. The added noise does as expected lead to improved CRPS-values for GraphCast* and Graph-FM, as any added variance up to the model error would improve the CRPS. As the noise creates some spread, the SpSkR also takes values higher than 0. We note however that in these evaluations the SpSkR starts high and rapidly decreases with lead time, as can be expected since all the stochasticity comes from time 0. Adding ICPs to Graph-EFM thus also results in higher SpSkR, even when it was already close to 1. As mentioned this is a simple experiment and there are better ways to perturb ICs. However, we hope that it shows that Graph-EFM (which also uses independent noise, but on the graph) does something more than just use this noise to perturb initial conditions.

---

> > > > > > ### Comment · Reviewer_PLfK · 2024-08-14
> > > > > >
> > > > > > Thanks, this partially addressed my concerns, I am thus adjusting my rating.

---

> > > > > > > ### Author Response · Authors · 2024-08-14
> > > > > > >
> > > > > > > We are happy to hear that our clarifications could address some of your concerns. Thank you again for actively engaging during this discussion period and for much valuable input on the paper!

---

### Official Review · Reviewer_568u · 2024-07-13

**Soundness:** 4
**Presentation:** 4
**Contribution:** 4
**Rating:** 10
**Confidence:** 5

**Summary:**

This paper introduces a new method for predicting weather using advanced deep learning models. The approach, called Graph-EFM, improves accuracy and better handles uncertainties in weather forecasts. It uses a 1.5 degree version of ERA5 and making weather predictions more reliable and useful for real-world applications.

**Strengths:**

The paper's strengths include the innovative use of Graph-EFM for accurate probabilistic weather forecasting, detailed experiments on large datasets, and clear presentation of methods. Graph-EFM significantly enhances uncertainty estimation and forecast reliability adding value to both research and practical weather prediction applications.

**Weaknesses:**

What happened if 0.25 degree ERA5 is used?

**Questions:**

I need to some figures during extreme events, e.g cyclones like Yaku.

Can the model deal with higher resolution data?

Or ERA6 when available?

Or more localised higher resolution data?

Do the results change a lot if it's only trained from 1980 onwards?

**Limitations:**

Not many, this is a fantastic piece of work

---

> ### Author Rebuttal · Authors · 2024-08-06
>
> We thank reviewer 568u for useful comments. See our response below:
>
> 1. I need to some figures during extreme events, e.g cyclones like Yaku.
>
> We have now included such a case study for Hurricane Laura, that can be found in the global author rebuttal.
>
> 2. About higher resolution data and ERA6
>
> There is indeed nothing that technically prevents us from applying these methods to higher resolution data. Some minor adaptations have to be done to the exact graph structures used, but the overall framework is directly applicable. The method should scale well to even higher resolutions, although naturally with an increasing memory requirement. Our choice to focus on 1.5° data is mainly due to the computational needs.
>
> Assuming that ERA6 will follow a similar format as ERA5, although higher resolution, there is no reason to believe our methods would not be applicable also to that dataset.
>
> Regarding higher resolution localized data, note that the MEPS data is exactly an example of this. The 10 km spatial resolution of this data is far higher than the 1.5° ERA5 data for this area (spatial resolution $\approx$ 167 x 76 km) and even higher than 0.25° ERA5 (spatial resolution $\approx$ 28 x 13 km). Also for limited area modeling we expect the method to be readily applicable also to data at even higher resolutions.
>
> 3. Do the results change a lot if it's only trained from 1980 onwards?
>
> While we have not had the time and resources to train such a model now, we would not expect substantial differences in the results from using only data from 1980 onwards. We opted for using as much data as possible, to make sure the model was trained on a set of weather scenarios as diverse as possible. However, in the GraphCast paper it was reported that including data from before 1980 had only minor impact on model performance [1]. Due to variations in climate one might even expect an improvement in performance when considering only more recent data, as the climate in 2020 would be more similar to that of 1980-2017 than that of 1959-1979. Using only more recent years would thus create a smaller shift in the data distribution between training and testing.
>
> [1] R. Lam et al. Learning skillful medium-range global weather forecasting. Science, 2023.

---

### Official Review · Reviewer_LjpL · 2024-07-13

**Soundness:** 3
**Presentation:** 3
**Contribution:** 3
**Rating:** 7
**Confidence:** 3

**Summary:**

The authors propose a graph-based ensemble forecasting model (Graph-EFM) to provide weather prediction with a hirearchical GNN framework. They used a hierarchical mesh graph to handle the challenges of capturing processes unfolding over different spatial scales and modeling the uncertainty in the chaotic system. The Graph-EFM provides a probabilistic weather forecasting with the benefit of capturing forecast uncertainity. The experiment results show the effectiveness and advantages of Graph-EFM compared to other deterministic models.

**Strengths:**

The hierarchical mesh graph provides a reasonable idea to handle different spatial scales for weather forecasting, which could inspire other researchers to handle problems in different domains.
The spatial dependencies are considered and handled within GNN layers.
Using ensumble-based model could capture the uncetainty of weather system.

**Weaknesses:**

In Figure 3, it seems like the selected ensemble members vary a lot, and how close is your forecast to the ground truth. Possibly, explaining a little bit of the underlying meaning of the measures in table 1 & 2 in the paper.

**Questions:**

1. The authors could better explain the underlying meaning of meaures, RMSE, CRPS, and SpSkR, for the results of weather forecast. Basically, I would like to ask authors to show how close their Graph-EFM's forecast is to the ground truth weather.

**Limitations:**

The authors have adequately addressed the limitations.

---

> ### Author Rebuttal · Authors · 2024-08-06
>
> We thank reviewer LjpL for useful comments. See our response below:
>
> 1. In Figure 3, it seems like the selected ensemble members vary a lot, and how close is your forecast to the ground truth?
>
> Note that Figure 3 shows the forecasts for 10 days in the future. At such lead times there is indeed a lot of uncertainty in any forecast that can be produced. In Figure 3 we want to show that the ensemble forecast does indeed capture this variability by producing varying ensemble members that each still represent a realistic weather state. The closest forecast (in terms of RMSE) to the ground truth is instead achieved by the ensemble mean. But as expected, due to the large variability at 10 days lead time, this ensemble mean is blurry and not a realistic weather state. We can however still see that some of the large-scale features present in the ground truth are present also in this prediction.
>
> 2. The authors could better explain the underlying meaning of measures, RMSE, CRPS, and SpSkR, for the results of weather forecast.
>
> We agree that the presentation of these metrics in the main part of the paper is quite short. Due to space restrictions we chose to move the full definitions and descriptions of these metrics to appendix D. We encourage anyone unfamiliar with these metrics in the context of weather forecasting to read also this appendix. To give a short, more high-level explanation of these metrics in the weather forecasting context:
>
> **RMSE** measures the average deviation of the forecast from the ground truth data. For probabilistic models, that produce forecasts as samples from a distribution, we compute the RMSE using a forecast representing the mean of this distribution. This is because RMSE is minimized by predicting the mean.
>
> **CRPS** is a probabilistic metric, and measures how well the ground truth value is captured by the distribution specified by the model. In this case this corresponds to how well the ground truth weather is captured by the probability distribution over possible future weather states specified by the model. One useful interpretation of the CRPS is that it measures the difference between the Cumulative Distribution Function (CDF) of the model distribution and the CDF of a Dirac distribution centered at the ground truth value. One can compute a version of CRPS also for deterministic models, in which case the model distribution also becomes a Dirac distribution, but centered at the predicted value. The CRPS then reduces to the Mean Absolute Error (MAE), and this is indeed what we report for the deterministic models in our paper. CRPS is thus useful as a metric that can compare both deterministic and probabilistic forecasts. See [1] for more about CRPS.
>
> **SpSkR** measures how calibrated the uncertainty of the model is. At SpSkR = 1 the ensemble forecasts are exchangeable with the ground truth, meaning that they exhibit similar levels of variability. This means that we can accurately use the ensemble spread as an indicator for forecast uncertainty. Each forecast will necessarily have some error, and a SpSkR close to 1 indicates that the model expresses the uncertainty about this error correctly.
>
> All the metrics are computed for each variable and lead time separately, but spatially averaged for all grid points.
>
> [1] T. Gneiting and A. E. Raftery. Strictly proper scoring rules, prediction, and estimation. Journal of the American Statistical Association, 2007.

---

> > ### Comment · Reviewer_LjpL · 2024-08-09
> >
> > Thanks for your input!

---

### Official Review · Reviewer_i1dL · 2024-07-13

**Soundness:** 3
**Presentation:** 2
**Contribution:** 3
**Rating:** 6
**Confidence:** 5

**Summary:**

The paper proposes Graph-EFM, a method that combines a hierarchical multi-scale graph neural network with a variational objective for probabilistic weather forecasting. The method performs on par with Graphcast on deterministic metrics with the extra benefit of uncertainty estimation.

**Strengths:**

- The paper is well-written and easy to follow.
- The paper tackles probabilistic weather forecasting, which is an important problem in the field.
- The proposed method is intuitive and makes sense. Overall, generative modeling is a potential direction for probabilistic weather forecasting. People have used GANs and diffusion, so a latent variable model is a natural addition to the literature.
- The performance looks promising, and it is more efficient than existing methods using diffusion.

**Weaknesses:**

- The authors should replace Table 1 with a line graph figure instead, as it allows comparison across different variables and lead times.
- Please see my questions below.

**Questions:**

- How important do you think the architecture is to the performance versus the objective function? The proposed architecture has an intuition similar to UNet, i.e., multi-scale features and the lowest layer can be used to parameterize the hidden variable.
- Diffusion models are considered the best family of models for generative modeling, surpassing GANs and latent variable models for other fields such as computer vision. What is the reason to believe latent variable models are the way to go for probabilistic weather forecasting?
- Why does the paper compare with Graphcast+SWAG but not the perturbed version of Graphcast and Gencast?
- How does the performance vary w.r.t. the number of ensemble samples? Given that sampling from a latent variable is fast, have the authors tried using more ensemble members?
- Is there an explanation why Graphcast is better than Graph-FM, but Graph-EFM is better than Graph-EFM (ms)?
- Why in LAM, the Graphcast architecture is doing better than the proposed architecture?

**Limitations:**

See above.

---

> ### Author Rebuttal · Authors · 2024-08-06
>
> We thank reviewer i1dL for useful comments. See our response below:
>
> 1. The authors should replace Table 1 with a line graph figure instead, as it allows comparison across different variables and lead times.
>
> Given the limited space in the main paper we did not find a way to fit line plots for all metrics, multiple variables and both datasets. We found that shrinking the line plots to make this feasible made them hard to read, so instead opted for Table 1 and keeping all line plots in the appendix. If we have the space in the camera ready version we would be happy to bring back some line plots also to the main paper.
>
> 2. How important do you think the architecture is to the performance versus the objective function?
>
> Both of these contributions are indeed important. The objective function, including the CRPS fine-tuning, is important for learning a calibrated probabilistic model. The hierarchical architecture does also fill an important role in Graph-EFM, spreading out the stochasticity from the latent random variable over the forecast region. Since Graph-EFM (ms) uses the same objective function, but not the hierarchical architecture, it can be viewed as an ablation of this part of the model. The poor performance of Graph-EFM (ms) confirms the importance of both of these components in order to learn a useful probabilistic model.
>
> 3. What is the reason to believe latent variable models are the way to go for probabilistic weather forecasting? (as opposed to diffusion models)
>
> While we believe there is a place also for diffusion models in weather forecasting, one strong reason to explore latent variable models is the difference in sampling speed. Being able to quickly sample large ensembles is a desirable property of probabilistic weather forecasting models. As diffusion models require multiple forward passes through a neural network to produce one sample, this can be a slow process. This issue is aggravated by the common practice of producing forecasts auto-regressively, meaning that each forecast time step must be sampled sequentially. In Graph-EFM this only requires one pass through the network per time step. However, speeding up diffusion model sampling is a very active area of research. We view this as another useful research direction also for weather forecasting, complementary to our approach.
>
> 4. Why does the paper compare with Graphcast+SWAG but not the perturbed version of Graphcast and Gencast?
>
> Regarding GenCast comparison see the global author rebuttal.
>
> We assume here that the perturbed version of GraphCast refers to the one considered in the GenCast paper, where GraphCast is initialized from perturbed initial conditions. We generally consider ensemble forecasting methods that start from multiple (possibly perturbed) initial conditions to be outside of our scope, and thus do not include perturbed GraphCast as a baseline. As outlines in section 3.1 we define our problem setting as modeling the distribution of future weather states conditioned on one specific set of initial conditions. The reason for this is that creating ensembles only based on multiple initial conditions typically limits the ensemble size or requires the use of ad-hoc perturbations without any guarantees of modeling the correct distribution.
>
> 5. How does the performance vary w.r.t. the number of ensemble samples?
>
> See the “global” author rebuttal for an investigation regarding the impact of the ensemble size.
>
> 6. Why in LAM, the Graphcast architecture is doing better than the proposed architecture?
>
> There is some more nuance when comparing GraphCast* and Graph-FM than saying one of these is strictly doing better in any of the experiments. Which out of these two that performs the best depends on what variable and lead time is considered, as can be seen from the full results in Appendix H and I. However, it is true that for longer lead times Graph-FM performs overall worse then GraphCast* in LAM. This is likely related to the structure of the problem in the LAM setting, that features the boundary forcing and other length scales than the global case. We would like to emphasize that we consider the probabilistic Graph-EFM model the main contribution and focus of the paper, and Graph-FM mainly a step in getting there. While further investigation into the possible failure modes of Graph-FM would be interesting, we prefer this to not be the focus of this work.
>
> 7. Is there an explanation why Graphcast is better than Graph-FM, but Graph-EFM is better than Graph-EFM (ms)?
>
> As noted above, the difference between Graphcast* and Graph-FM has some nuance. Still, what should be remembered here is that the graph partly fills a different purpose in the deterministic and probabilistic models. While in the deterministic model it is used to spread information from the input and produce the best possible forecast for the next step, in the probabilistic models the graph has the key role of augmenting the sampled latent variables into a coherent realization of the weather state. In particular, as we associate the latent variable with the mesh nodes, the stochasticity enters in different ways when using the multiscale and hierarchical graphs. Since the multiscale graph does not offer any further dimensionality reduction than mapping from the grid to the mesh, it also means that the latent variable $Z^t$ will have a higher total dimensionality ($|\mathcal{V}_1| \times d_z$) in Graph-EFM (ms) than when using the hierarchical graph in Graph-EFM ($|\mathcal{V}_L| \times d_z$). For these reasons one should not assume that any performance differences between using the multiscale or hierarchical graphs in deterministic models straightforwardly translates to the probabilistic case.

---

> > ### Comment · Reviewer_i1dL · 2024-08-08
> >
> > Thank you for your answering my questions. I would still love to see a comparison of Graph-EFM with perturbed Graphcast. Regardless of the formulation of the problem, IC perturbations are always a common technique to achieve uncertainty estimation, and have been used in numerical methods, hybrid ML methods like NeuralGCM, and deep learning. Comparing with such a simple baseline will help shed light on the performance gain of Graph-EFM.

---

> > > ### Author Response · Authors · 2024-08-09
> > >
> > > We agree that this could be interesting to investigate in more detail. However, we believe that it should in that case also involve a version of Graph-EFM with initial condition perturbation, to understand the effect of different sources of randomness on the final uncertainty estimates and to ensure that we compare "apples and apples". Note that there is nothing that prevents us from combining the latent variable formulation of Graph-EFM with initial condition perturbation, but in our view these mechanisms conceptually correspond to different sources of uncertainty: uncertainty about the initial conditions, and uncertainty in the modelled dynamics even in the (hypothetical) case of perfectly known initial conditions. A version of Graph-EFM with initial condition perturbations should additionally be trained with the perturbations present in order to learn the correct distribution. We will consider a more in-depth investigation into this matter for future work.

---

### Author Rebuttal · Authors · 2024-08-06

We thank all reviewers for valuable comments and questions that we are sure will improve the overall quality of our paper. We have responded to the points raised by each reviewer separately, but also include this general rebuttal with a few points that we think could be relevant to all.

### Extreme Weather Case Study

Multiple reviewers requested a case study on ensemble forecasting for extreme weather events. We agree that this would be a valuable addition to the paper, as this is one important motivation for the work.

We include here a case study for Hurricane Laura, that will be added to the paper. In August of 2020 Laura developed in the Atlantic Ocean and reached Hurricane levels in the Gulf of Mexico, eventually making landfall in Louisiana and causing major damages [1]. We study here forecasts for 2020-08-27T12 UTC, which is 6 hours after the Hurricane hit land. All forecasts are for this exact time point, but initialized a varying number of days before. We here run 50 member ensemble forecasts using Graph-EFM and deterministic forecasts using the GraphCast* and Graph-FM models. Figure 1 in the attached pdf shows 10 m wind speeds in ERA5 and the forecasts. For Graph-EFM we plot both forecasts from randomly sampled ensemble members and a cherry-picked best member that was deemed to most closely match ERA5. Note that the 1.5° resolution that we work with makes determining similarity or exact positions somewhat challenging.

We see in Figure 1 that at 7 days lead time there is great uncertainty, and the deterministic models do not show the hurricane at all. In the ensemble forecast from Graph-EFM there exists however already members indicating the possibility of a Hurricane making landfall a week ahead (see for example the cherry-picked “Best member”). While the ensemble includes many possible scenarios, a total of 7 members show the development of a hurricane in the area. Having information of such possible scenarios a long time ahead allows for planning and readying disaster response efforts that might be needed. Note that discovering these scenarios is only possible through an ensemble forecast, as the deterministic models do not indicate such an event. At 5 days lead time all models are indicating the development of a hurricane. While the deterministic models do a good job here, they are indicating a landfall location slightly too far eastward. The ensemble members from Graph-EFM however show a range of different positions for the hurricane, indicating the uncertainty in the landfall location. At 3 days ahead 42 out of 50 ensemble members show the hurricane making landfall, but still with some uncertainty about the exact location. At 1 day lead time all models give an accurate forecast of the position of the hurricane. Apart from position, it is also interesting to consider how the models capture the intensity in terms of wind speeds. At 1 day ahead the deterministic models are somewhat underestimating the wind speed. The Graph-EFM ensemble shows a range of possible values, indicating the uncertainty in the exact wind intensity. Overall this study exemplifies how a large ensemble forecast from a machine learning model can be used to discover possible extreme weather scenarios at long lead times and uncover the uncertainties associated with them.

[1] Hurricane Laura 2020. National Weather Service. https://www.weather.gov/lch/2020Laura

### Impact of Ensemble Size on Results

As pointed out by reviewer i1dL it would be interesting to know how the performance of the model, in terms of metric values, varies when sampling different number of ensemble members. To investigate this we ran the evaluation of Graph-EFM with 5-80 members for the global data and 5-100 for the limited area model.

Results for a selection of variables and metrics are shown in Figure 2 in the attached pdf. As expected the RMSE of the ensemble mean decreases when sampling more members. However, already when sampling 20 or 25 members the results are fairly close to the full ensemble. For SpSkR the differences are even smaller. As the CRPS is a property of the distribution of the model forecast its true value does not depend on the number of samples drawn. In practice we compute CRPS using an unbiased estimator, and the variance of this estimator decreases with ensemble size. When averaged spatially and over the whole test set we do however not see any difference in CRPS for different ensemble sizes. All these trends hold consistently for all variables in both the global and limited area datasets. We intend to add the results of this investigation to the paper.

Given that any improvements to metrics saturate, we would not expect the results to meaningfully change from sampling more than 80/100 members. It should however be noted that the motivation for sampling very large ensembles is mainly not to improve on metrics such as these. More important motivations for large ensembles include estimating probabilities of rare events or studying different possible scenarios of extreme weather.

### Comparison to GenCast

The reason for not comparing against GenCast is the lack of any openly available code, pre-trained models or produced forecasts (at the time of writing). Such a comparison would require reimplementing and retraining a GenCast model from scratch, which we deem outside the scope of this work. Note that doing this for GraphCast was much more feasible, as there is code openly available and the model is more similar to ours.

---

### Decision · Program_Chairs · 2024-09-25

**Decision:**

Accept (spotlight)

**Comment:**

The authors propose an autoencoder version of GraphCast and UNet for probabilistic global medium-range weather forecasting.

- **Reviews**: While the paper received (very) high scores from some reviewers, some reviewers were initially more critical.
- **Rebuttal**: The authors successfully addressed the concerns of the more critical reviewers by discussing computational challenges and justifying their approach through experiments.
- **Decision**: I recommend acceptance.